# Evaluating the strength of the land–atmosphere moisture feedback in earth system models using satellite observations

Paul A. Levine[1], James T. Randerson[1], Sean C. Swenson[2], and David M. Lawrence[2]

[1]Department of Earth System Science, University of California, Irvine, CA 92697, USA
[2]Climate and Global Dynamics Division, National Center for Atmospheric Research, Boulder, CO 80305, USA

*Correspondence to*: Paul A. Levine (plevine@uci.edu)

**Abstract.** The relationship between terrestrial water storage (TWS) and atmospheric processes has important implications for predictability of climatic extremes and projection of future climate change. In places where moisture availability limits evapotranspiration (ET), variability in TWS has the potential to influence surface energy fluxes and atmospheric conditions.
Where atmospheric conditions, in turn, influence moisture availability, a full feedback loop exists. Here we developed a novel approach for measuring the strength of both components of this feedback loop, i.e., the forcing of the atmosphere by variability in TWS and the response of TWS to atmospheric variability, using satellite observations of TWS, precipitation, solar radiation, and vapor pressure deficit during 2002–2014. Our approach defines metrics to quantify the relationship between TWS anomalies and climate globally on a seasonal to interannual time scale. Metrics derived from the satellite data
were used to evaluate the strength of the feedback loop in 38 members of the Community Earth System Model (CESM) Large Ensemble (LENS) and in six models that contributed simulations to Phase 5 of the Coupled Model Intercomparison Project (CMIP5). We found that both forcing and response limbs of the feedback loop in LENS were stronger than in the satellite observations in tropical and temperate regions. Feedbacks in the selected CMIP5 models were not as strong as those found in LENS, but were still generally stronger than those estimated from the satellite measurements. Consistent with
previous studies conducted across different spatial and temporal scales, our analysis suggests that models may overestimate the strength of the feedbacks between the land surface and the atmosphere. We describe several possible mechanisms that may contribute to this bias, and discuss pathways through which models may overestimate ET or overestimate the sensitivity of ET to TWS.

## 1 Introduction

Land–atmosphere feedbacks can result from the coupling of the terrestrial moisture state with temperature, precipitation, or radiation (Betts et al., 2014; Findell and Eltahir, 1997; Guillod et al., 2015; Koster et al., 2004). Land–atmosphere coupling occurs when terrestrial moisture anomalies influence the partitioning of surface energy between latent and sensible heat fluxes that, in turn, influence the development of the planetary boundary layer (PBL) (Seneviratne et al., 2010). Temperature coupling generally leads to a positive feedback, with wetter soil contributing to a higher evaporative fraction (EF, the ratio of

the latent heat flux to the sum of the sensible and latent heat fluxes), a lower surface temperature, and decreased evaporative demand (Hirschi et al., 2011; Miralles et al., 2012). Precipitation coupling can lead to both positive and negative feedbacks, as the influence of EF on the development of the PBL can serve to either enhance or suppress cloud formation and precipitation (Findell and Eltahir, 2003; Guillod et al., 2015). Cloud radiative coupling can likewise lead to positive or negative feedbacks as insolation and evaporative demand, as a function of cloud cover, are either enhanced or suppressed (Betts, 2009; Cheruy et al., 2014). Temperature, precipitation, and radiation feedbacks each stem from coupling between terrestrial moisture and evapotranspiration (ET), which occurs most strongly in conditions of intermediate moisture availability (Seneviratne et al., 2010).

Evidence of these feedbacks has been observed in both in situ and remotely sensed data (Eltahir, 1998; Findell and Eltahir, 1997; Guillod et al., 2014, 2015).  Some observational analyses have found land–atmosphere feedback strength to be relatively weak compared to the influence of large-scale atmospheric forcing (Alfieri et al., 2008; Phillips and Klein, 2014). Other observational studies have highlighted the role of these feedback mechanisms in the initiation and exacerbation of climatic extremes such as droughts and heat waves (Hirschi et al., 2011; Miralles et al., 2014; Whan et al., 2015).

Large-scale land–atmosphere coupling in general circulation models has been demonstrated by a series of experiments from the Global Land Atmosphere Coupling Experiment (GLACE) project (Guo et al., 2006; Koster et al., 2004, 2006). The GLACE efforts found that coupled climate models differed greatly in the extent to which soil moisture variations affect precipitation and surface air temperature, but models generally agreed on the spatial distribution of relative coupling strength, with "hotspots" of strong coupling during boreal summer found in the central United States, northern Amazonia, the Sahel, western Eurasia, and northern India. These hotspots were found in regions of intermediate soil wetness, which is consistent with the understanding that strong land–atmosphere coupling occurs under conditions in which terrestrial moisture availability limits ET (Seneviratne et al., 2010). GLACE efforts also showed that correct soil moisture initialization improves seasonal forecast skill of temperature and, to a lesser extent, precipitation, particularly in cases with a large initial soil moisture anomaly (Koster et al., 2010, 2011).

Additional studies have considered land–atmosphere feedbacks in the coupled earth system models (ESMs) used by the Intergovernmental Panel on Climate Change (IPCC) (Dirmeyer et al., 2013; Notaro, 2008; Seneviratne et al., 2006, 2013). Notaro (2008) was able to confirm the boreal summer GLACE hotspots, as well as identify several additional austral summer hotspots, in the models used for the IPCC Fourth Assessment Report (AR4). Analysis of long-term projections from the Fifth Phase of the Coupled Model Intercomparison Project (CMIP5) indicated an increased control of land surface moisture on boundary layer conditions with climate change (Dirmeyer et al., 2013). The GLACE Coupled Model Intercomparison Project (GLACE–CMIP5) experiment found that modeled coupling strength plays an important role in

simulated response to global warming, with greater warming evident in more strongly coupled models due to interactions between soil moisture, temperature, and precipitation (Berg et al., 2015; May et al., 2015; Seneviratne et al., 2013).

Despite the importance of land–atmosphere coupling in both short-term predictability of climatic extremes and long-term uncertainty in climate change, validation efforts have suggested that climate models may not be correctly representing the strength, and in some cases even the sign, of these feedbacks (Ferguson et al., 2012; Hirschi et al., 2014). The metrics developed for GLACE are based on model experiments with no direct observational equivalents. However, correlation-based metrics that do enable direct comparison with observations suggest that models may overestimate land–atmosphere coupling strength (Dirmeyer et al., 2006a). Zeng et al. (2010) found that version 3 of the Community Climate System Model (CCSM3) showed a higher coupling strength than reanalysis or observational data. Mei and Wang (2012) found that coupling strength was reduced when the Community Atmosphere Model (the land surface component of CCSM3) was updated from version 3 (CAM3) to version 4 (CAM4), though the coupling strength of the updated version was still stronger than observations and reanalysis.

The Local Land–Atmosphere Coupling (LoCo) Project has focused on developing a suite of metrics for diagnosing land–atmosphere coupling strength in observations and models. LoCo metrics consider both the influence of soil moisture on EF, and the influence of EF on diurnal-scale boundary layer development (Santanello et al., 2009). Ferguson et al. (2012) used the LoCo approach to compare global remote sensing data sets of soil moisture, EF, and lifting condensation level with several land surface models and reanalyses. They found that even though the models were able to simulate the correct spatial pattern of stronger coupling in moist–arid transitional regions, the models tended to simulate a stronger influence of soil moisture on surface turbulent fluxes than what was observed in the satellite data. Guillod et al. (2014) used a combination of flux tower, remote sensing, and reanalysis data sets to demonstrate that measured strength of coupling between EF and precipitation depends greatly on the data source and scale, and that strong coupling apparent in a previous analysis (Findell et al., 2011) was not consistent with the observations.

While many of the previously mentioned studies have confirmed the long-standing suspicion that models may overestimate coupling strength relative to observations, more recent work has indicated that observations and models may not even agree on the sign of the precipitation feedback. Taylor et al. (2012) performed a spatial analysis of the relationship between soil moisture and afternoon precipitation using data from remote sensing, reanalysis, and coupled models. They found evidence of a negative feedback in the remote sensing observations, with afternoon rain being more likely over regions of drier soil, as opposed to the positive feedback that was apparent in the models. Guillod et al. (2015) addressed these findings by replicating the spatial analysis and complementing it with a temporal analysis. They found a negative spatial feedback, consistent with the one found by Taylor et al. (2012), but a positive temporal feedback, with afternoon precipitation at a given location being more likely after mornings of relatively moist soil.

These studies highlight the need for continued efforts toward evaluating the coupling strength of models relative to observations using a wide array of data sources at a range of spatiotemporal scales. Apparent coupling strength depends greatly on the spatial scales of analysis (Hohenegger et al., 2009), indicating that observations at the scale of flux towers should not be expected to yield the same coupling strength as those at the scale of global climate models (Guillod et al., 2014). Consistency between the spatial scale of observations and models is greatly assisted by earth observation satellites that have been continuously monitoring several relevant land surface and atmospheric variables over multiple years (Teixeira et al., 2014). Measured or modeled coupling strength will also depend on the time scales in question (Guillod et al., 2015), and while the LoCo efforts have improved the understanding of synoptic and diurnal scale mechanisms, there in an additional need to examine these processes on seasonal to interannual time periods.

Here we introduce a set of metrics for measuring the strength of land-atmosphere interactions on seasonal timescales by combining satellite remote sensing datasets of terrestrial water storage, precipitation, shortwave radiation, and surface atmospheric temperature and water vapor during 2002–2014.  These new metrics complement previous studies and are unique in several ways. In particular, we designed our metrics to consider interannual variability of entire seasons in order to complement the temporal resolution of LoCo metrics, which focus on day-to-day variability within one or more seasons. Land–atmosphere coupling on seasonal timescales has been shown to be essential in enabling tropical forests to survive during the dry season in the Amazon (Lee et al., 2005) and as a mechanism enabling seasonal forecasts of fire risk (Chen et al., 2013, 2016).

Until recently, studies using remote sensing data to look for evidence of land–atmosphere coupling relied on products that provide information about surface soil moisture (Ferguson et al., 2012; Taylor et al., 2012). Consideration of root-zone soil moisture has recently been accomplished only indirectly via data-assimilated estimates (Guillod et al., 2015). The inability to directly consider root-zone soil moisture has been suggested as an explanation for the relatively weak coupling observed using remote sensing data (Hirschi et al., 2014). In order to include root-zone soil moisture, as well as other sources of moisture available across entire seasons, here we analyzed remote sensing data of the entire terrestrial water storage (TWS) column.

The metrics introduced here were specifically designed to use the monthly TWS anomaly (TWSA) anomaly product from the Gravity Recovery and Climate Experiment (GRACE) mission (Landerer and Swenson, 2012; Wahr et al., 2004). The GRACE TWSA product integrates soil moisture at all layers along with surface, canopy, snow/ice, and aquifer storage, as each of these components represents a potential source of moisture for fulfilling evaporative demand. For example, in areas where agricultural ecosystems are important, diversion of lake and river water resources and withdrawal from aquifers may contribute to irrigation fluxes and thus ET. Furthermore, surface storage of liquid water and snow represent sources of water

that are available for and potentially limiting to ET. Under these conditions, month-to-month TWS anomalies capture portions of the terrestrial water cycle that soil moisture alone may not.

Previous studies have largely focused on land surface moisture availability as a forcing mechanism on the atmosphere, as
this relationship has important implications for seasonal predictability as well as the projection of the frequency and severity of climatic extremes. However, the land surface response to the atmosphere is governed by many of the same processes through which terrestrial moisture availability forces atmospheric conditions, and it determines the conditions that drive subsequent land surface forcing. It is therefore critical to assess the response of land surface moisture to atmospheric conditions, as an accurate representation of these processes is essential for generating the correct terrestrial moisture
variability that will go on to influence the atmosphere. As far as we can tell, this response limb of the land surface feedback loop has not been systematically integrated with existing analyses of land–atmosphere coupling strength.

Our globally applicable approach used the annual cycle of TWS drawdown and recharge to isolate the months of the year during which the land surface loses moisture, which we refer to as the drawdown interval (Figure 1a). We selected this
interval because past work has shown that the land surface's influence on the atmosphere is most prevalent during summer in the northern hemisphere (Cheruy et al., 2014; Phillips and Klein, 2014) and during the dry season in tropical forests (Harper et al., 2013; Lorenz and Pitman, 2014). This approach allowed us to investigate land surface coupling at a global scale, and to extend metrics developed in previous work for pre-defined monthly intervals corresponding to boreal summer (e.g., Guo and Dirmeyer, 2013; Koster et al., 2006) to be applicable to any seasonality.

In our analysis, separate metrics were calculated to consider the influence of TWS at the onset of the drawdown interval on atmospheric conditions in subsequent months, and simultaneously, the influence of atmospheric conditions during the drawdown interval on terrestrial water storage at the end of the season. We refer to these two relationships as the forcing and response limbs, respectively, of the fully coupled feedback loop between the land surface and the atmosphere (Figure 1b).
We estimated the strength of these feedbacks during 2002–2014 using GRACE and other satellite remote sensing data (Table 1). We then used the satellite observations to evaluate the strength of these feedbacks in the Community Earth System Model (CESM) Large Ensemble (LENS) (Kay et al., 2014) and in several models that contributed simulations to CMIP5 (Table 2).

## 2 Methods

### 2.1 Remote sensing data

We obtained Level-3 TWSA data from GRACE using the University of Texas at Austin Center for Space Research (CSR) spherical harmonic solutions (Swenson, 2012). Global land data at a 1° resolution were scaled using the coefficients provided by Landerer and Swenson (2012). The study period was limited to September 2002 through November 2014, in

order to minimize temporal gaps. GRACE data during the study period included eight non-consecutive and two consecutive missing months, which were filled using linear interpolation. At each grid cell, the TWSA time series was decomposed into linear trend, seasonal cycle, and interannual variability components using ordinary least squares regression. This decomposition allowed us to estimate a mean annual cycle at each grid cell with minimal influence of any long-term trend.

Level-3 near-surface temperature and relative humidity were obtained globally at a monthly, 1° resolution from the ascending (daytime) orbit of the Atmospheric Infrared Sounder (AIRS) platform (Susskind et al., 2014). Vapor pressure deficit (VPD) was calculated from the AIRS data using the August–Roche–Magnus approximation to the Clausius–Clapeyron relation (Lawrence, 2005). Precipitation (PPT) data were obtained from the Global Precipitation Climatology

Project (GPCP), a merged satellite and gauge-based data set (Huffman et al., 2009) at a daily, 1° resolution and then integrated monthly. Downwelling shortwave radiation (SW↓) was obtained globally at a monthly, 1° resolution from the Clouds and the Earth's Radiant Energy System (CERES) Energy Balanced And Filled (EBAF) Surface product (Loeb et al., 2009). More information describing the remote sensing and reanalysis data products used in our analysis is summarized in Table 1.

**2.2 Drawdown interval**

As a first step, we used the mean annual cycle from GRACE to determine the months of the maximum and minimum TWS anomalies in order to define the drawdown interval at each 1° land grid cell (Figure 2). Northern hemisphere middle and high latitudes exhibited a drawdown interval beginning in the spring (MAM) and ending in the late summer or fall (ASO), reflecting the timing of the boreal summer growing season. At lower latitudes, the North American, African, and Asian

monsoons were evident, with Mexico, India, and the Sahel showing a drawdown interval beginning in September, after the monsoonal precipitation has peaked, and ending the following spring after the winter dry season. The onset of the drawdown interval reversed abruptly at the equator in Africa and Asia, with the drawdown interval reflecting a winter dry season in the austral low latitudes transitioning to a summer growing season in the austral midlatitudes. Within the months of our study period, the portion of land grid cells that experience 11, 12, and 13 complete drawdown intervals are 9.4%, 90.5%, and 0.1%

respectively.

**2.3 Coupling metrics**

Existing literature generally defines "land–atmosphere coupling" as the extent to which atmospheric conditions are forced by the land surface state, and would use the term "atmosphere–land coupling" to refer to the land surface response to atmospheric drivers (Seneviratne et al., 2010). In this study, we develop what we refer to as "coupling metrics" to indicate

the strength of both limbs of the fully coupled land–atmosphere feedback loop. We use the terms "forcing" and "response" to indicate whether we are considering the forcing of the atmosphere by the land surface, or the response of the land surface to the atmosphere (Figure 1b).

We defined our forcing metric as the Pearson product-moment correlation coefficient between the TWS anomaly at the onset of the drawdown interval ($TWSA_{max}$) and the surface atmospheric conditions during the drawdown interval ($ATM_{di}$). In our analysis, we selected 3 variables to represent the atmospheric state: VPD, SW↓, and PPT. These atmospheric variables were averaged during the drawdown interval, including during the months of climatological maximum and minimum TWSA. We chose these variables because they represent various aspects of evaporative supply (PPT) and demand (VPD and SW↓).

Similarly, we defined our response metric as the correlation coefficient between atmospheric conditions during the drawdown interval ($ATM_{di}$) and the land surface state at the end of the drawdown interval ($TWSA_{min}$). Although most previous diagnoses of land–atmosphere coupling has focused on the forcing limb, we argue the response limb is equally important as a metric for model evaluation. Specifically, if variability in the balance between evaporative supply and demand does not lead to the correct TWS variability, then the incorrect TWS response will feed back into subsequent forcing on the atmosphere.

We note that these metrics do not provide distinctive information for measuring the strength of land–atmosphere coupling or the land surface response. While the metrics include the influence of direct land-atmosphere interactions, they are also potentially influenced by atmospheric and soil moisture persistence, as well as remote forcing from sea surface temperatures (SSTs) (Orlowsky and Seneviratne, 2010; Mei and Wang, 2011). Nevertheless, these metrics may still serve as useful benchmarks against which to evaluate the ability to ESMs to reproduce the proper relationships based on the combination of these factors.

Here we note that our evaluation of both the forcing and response metrics will follow a nomenclature that considers strong coupling as acting in the direction of an overall positive feedback loop. In regions with a strong positive feedback, higher than average TWS would be followed by lower than average VPD, as more available water is able to fulfill evaporative demand. Therefore, strong TWS forcing on VPD would be associated with a negative correlation coefficient. Higher VPD during the drawdown interval would increase evaporative demand, potentially leading to a more negative TWS anomaly, therefore a strong response of the land surface to VPD would also be associated with a negative correlation coefficient.

Because the partitioning of surface fluxes can, depending on the spatiotemporal scale, cause a change of either sign to cloudiness and precipitation (Taylor et al., 2012; Guillod et al., 2015), correlation coefficients of either sign could indicate strong land surface forcing on PPT and SW↓. However, the response metrics would be expected to show greater consistency. Higher PPT during the drawdown interval would be expected to increase TWS (positive correlation), while higher SW↓ would increase evaporative demand, thereby decreasing TWS (negative correlation). Therefore, to maintain consistent nomenclature based on evaluating the strength of a positive moisture feedback, we consider strong coupling in both the

forcing and response metrics to be associated with a positive correlation in the case of PPT and a negative correlation in the case of SW↓.

**2.4 Community Earth System Model Large Ensemble**

We used the metrics described above to evaluate feedback strength in the Community Earth System Model (CESM) Large Ensemble (LENS). LENS comprises an ensemble of 38 fully coupled runs in which air temperature initial conditions are perturbed slightly (by an amount less than round-off error) to reveal the internal variability inherent within the coupled climate model. LENS has demonstrated that the uncertainty in climate projections due to internal climate variability inherent in CESM is comparable to the ranges of output within the entire CMIP5 experiment (Kay et al., 2014). LENS uses version 1 of CESM (CESM1) with version 5 of the Community Atmosphere Model (CAM5) and version 4 of the Community Land Model (CLM4) at a horizontal resolution of 1°. The ensemble run follows protocols from the CMIP5 experiment, with historical radiative forcing for the 20[th] century and representative concentration pathway 8.5 (RCP8.5) forcing for the 21[st] century.

The LENS data were chosen as a starting point for feedback evaluation for two reasons. First, the availability of a TWS variable in these simulations enabled a direct comparison with metrics derived using data from GRACE. The TWS field in CLM4 included water from surface and canopy storage, snow and ice, soil moisture, and a dynamic aquifer, in addition to river water storage terms from the coupled River Transport Module (RTM). The coupling of CLM4 with RTM has been shown to be important for simulating both the annual cycle and interannual variability of TWS in comparison with GRACE (Kim et al., 2009).

Second, the ensemble allowed us to test the importance of internal model variability for the diagnosis of feedback strength. Because the complete satellite record was relatively short (containing no more than 12 drawdown intervals at any location), comparison with an equivalent single time series of model output could be influenced by a model's internal decadal-scale variability (Kay et al., 2014). Analyzing the full ensemble from LENS enabled us to assess the sensitivity of our forcing and response metrics to this variability. We extracted from each ensemble member the equivalent months of the satellite record (September 2002 through November 2014), with data prior to December 2005 coming from the historical runs, and data from January 2006 onward coming from the RCP8.5 simulations.

**2.5 Assessment of uncertainty**

To assess the sensitivity of our metrics to observational uncertainty, we used a Monte Carlo sampling approach. For each of the 38 members of LENS, we calculated coupling metrics ten times with random noise added to both TWSA and atmospheric variable time series at each grid cell. The noise was randomly generated from a Gaussian distribution with a mean of zero and a standard deviation equal to 25% of the standard deviation of the original data. Comparing these results

with those from the unaltered data provided some indication of how much our coupling metrics are degraded by random noise as an approximation of observational uncertainty.

In addition, to assess how our analysis may be influenced by uncertainty due to the selection of satellite data, we substituted data from the European Centre For Medium-range Weather Forecasting (ECMWF) Interim Reanalysis (ERA-Interim) (Dee et al., 2011) in place of AIRS, GPCP, and CERES-derived variables. We only used atmospheric reanalysis data for this sensitivity analysis, as these data benefit from assimilation of observations, while we continued to use GRACE for TWSA. Comparing results from this GRACE–reanalysis hybrid to those using only satellite data provided a general indication of how sensitive our coupling metrics were to the data source.

## 2.6 CMIP5 analysis

To extend our analysis to models that did not output an explicit TWS field, we compared accumulated residuals of precipitation, evapotranspiration, and total runoff (surface and subsurface) with the explicit TWS variable in the LENS simulations. We also compared coupling metrics calculated from LENS using accumulated residuals with those calculated from the explicit TWS field. After we determined that the accumulated residuals of the water balance represented much of the variability in the explicit TWS variable and yielded coupling metrics with similar distributions within LENS, we calculated equivalent metrics for several model simulations in the CMIP5 archive (Table 2). We selected the CMIP5 models that were similar to LENS (CESM1-CAM5 and CESM1-BGC) as well as the models that participated in the GLACE–CMIP5 experiment (Seneviratne et al., 2013) for which each necessary output field was available (CCSM4, GFDL-ESM2M, GFDL-ESM2G, IPSL-CM5A-MR, and IPSL-CM5A-LR).

## 3 Results

### 3.1 Drawdown interval and interannual variability

A comparison of the months of maximum and minimum terrestrial water storage as determined by climatologies of GRACE and the LENS ensemble mean indicated that the model largely reproduces the timing of TWSA seasonality evident in the satellite observations (Figure 2). Geographic patterns of seasonality were consistent between the model and observations, though a phase shift in the drawdown interval is apparent in eastern Canada and central Eurasia where LENS had a one-month early bias for both the maximum and minimum TWSA, in southeast North America where the onset of the modeled drawdown interval was slightly later than the observations, and in parts of east Asia and Australia where the modeled drawdown interval ended earlier than in the observations. However, despite capturing generally correct timing, the model exhibited higher interannual variability of $TWSA_{max}$ and $TWSA_{min}$ across the 11-12 drawdown intervals compared with the satellite data (Figure 3) particularly in the southern United States, southern South America, central and eastern Africa,

southern Asia, and eastern Australia. One possible explanation for this is the presence of multi-year trends in aquifer storage in CLM that are not consistent with GRACE (Swenson and Lawrence, 2015).

A comparison of the interannual variability of atmospheric variables across multiple drawdown intervals between the model and satellite data showed various degrees of consistency (Figure 4). The magnitude and geographic pattern of $VPD_{di}$ was generally consistent, though LENS showed greater interannual variability than AIRS in central and western North America, South America, northern and southern Africa, and southern Asia. In the case of $PPT_{di}$, LENS showed less interannual variability than GPCP in Southeast North America and much of South America, but the two were largely consistent elsewhere. $SW{\downarrow}_{di}$ was the least consistent between the model and satellite data, as LENS showed greater interannual variability than CERES in southern North America, northern Eurasia, most of Africa, and most of Australasia.

Comparing both the timing of TWS dynamics and the interannual variability of TWS and the atmospheric variables between the observations and model output provides context for interpreting the correlation-based metrics we present next. Although there are some inconsistencies, as noted above, the model largely reproduced the same patterns evident in the remote sensing data. In many regions, the interannual variability in model output was similar to the observed variability, indicating that CESM was able to simulate reasonably well the baseline properties (timing and variability) that influence feedback dynamics.

### 3.2 Evaluating feedbacks for a single model simulation

The forcing metric for VPD derived from GRACE and AIRS showed regions of strong coupling, in which $TWSA_{max}$ was negatively correlated with $VPD_{di}$, in the northern Great Plains, northern South America, southern Africa, southern and western India, north central Eurasia, and northern Australia (Figure 5a). Regions with strong positive correlation were much less common, and were largely confined to areas of very low GRACE-derived $TWSA_{max}$ variability (Figure 3a). Positive correlations are unlikely to reflect direct land–atmosphere coupling. Instead, they demonstrate how remote SST forcing can, depending on persistence and time delays with atmospheric responses, lead to apparent negative relationships such as those demonstrated by Wei et al. (2008). In comparison with the satellite data, the VPD forcing metrics from the first ensemble member of LENS (Figure 5c) showed much stronger coupling in the southern and eastern Amazon, and marginally stronger coupling strength across many regions in temperate Asia.

The response metrics for VPD showed much stronger coupling than the forcing metric in both the satellite data and the model (Figure 5b,d). Satellite data yielded negative correlation coefficients nearly everywhere, with positive correlations found only in arid regions of low TWS variability. Particularly strong response metrics were found in eastern North America, northern South America, western Eurasia, the Sahel, India, and eastern Australia. The first ensemble member from LENS showed widespread negative correlations, and did not show the positive correlations found in the satellite data.

Response coupling in LENS was much more spatially homogeneous than in the satellite data, though northern South America and western Eurasia still showed stronger coupling than elsewhere.

Many of the areas that showed a strong forcing metric for VPD also showed a relatively strong forcing metric for PPT, though the PPT forcing metric was overall weaker than that for VPD (Figure 6a). The response metric for PPT was generally positive, indicating that for much of the globe, a more positive $TWSA_{min}$ was associated with higher precipitation rates (Figure 6b). Both the forcing and response metrics were somewhat stronger in the LENS member relative to those evident in the satellite data (Figure 6c,d).

The forcing metrics for SW↓ showed a mixture of positive and negative correlations, indicating that higher $TWSA_{max}$ was either positively or negatively coupled with shortwave radiation (Figure 7a). This finding is consistent with both positive and negative coupling between cloud cover and terrestrial moisture observed over shorter time scales (Taylor et al., 2012; Guillod et al., 2015). The response metrics for SW↓ were generally negative, indicating that greater seasonal shortwave radiation was associated with more negative $TWSA_{min}$ (stronger coupling), with West Africa being a notable exception (Figure 7b). The LENS member showed generally stronger coupling in both the forcing and response metrics for SW↓ (Figure 7c,d).

### 3.3 Evaluating the CESM Large Ensemble

In temperate and tropical regions, forcing metrics were generally stronger in LENS (more positive correlations for PPT, more negative for VPD and SW↓) than in the satellite and reanalysis data, indicating a stronger land surface forcing of the surface atmospheric state in the model than in the observations (Figure 8). In boreal regions, forcing metrics were much weaker (closer to zero) than at lower latitudes in both the satellite data and in LENS, indicating very little relationship between $TWSA_{max}$ and $ATM_{di}$. This is consistent with high levels of climate variability in many high latitude regions driven by the Arctic Oscillation, the North Atlantic Oscillation, and other dynamical modes (Cohen and Barlow, 2005). Furthermore, at high latitudes, ET is generally energy limited rather than moisture limited, which would lead to weak forcing metrics as moisture availability would not strongly influence atmospheric conditions.

Response metrics were also generally higher in LENS than in both the satellite and reanalysis data (Figure 9). Noticeable exceptions were the VPD and PPT response metrics in the tropics, which were close to the satellite observations, and the boreal SW↓ and tropical PPT response metrics, which were close to the reanalysis estimates. Despite the internal variability evident within the model ensemble, and the difference between metrics as measured by the satellite data compared with the reanalysis data, the general pattern indicated that modeled response metrics were higher than those from observations and reanalysis.

### 3.4 Analysis of uncertainty

The internal variability across the ensemble of simulations in LENS yielded a distribution of forcing and response metrics with a spread on the same order of magnitude as the difference between modeled and satellite-derived zonal averages. The distribution of coupling metrics from LENS revealed the sensitivity of the relationships to decadal climate variability given the relatively short TWS time series. Comparing this distribution with the spread between the purely satellite-derived metrics and GRACE-reanalysis hybrid indicated the sensitivity of our metrics to the choice of data source. The differences between satellite and reanalysis metrics were generally higher in the tropics, particularly for VPD and SW↓, and in mid-latitude VPD for both forcing and response variables. Elsewhere, the differences were generally similar to or less than the differences between the observationally constrained zonal averages and the LENS distributions.

Comparing the original LENS forcing and response metrics with those calculated after adding random noise to LENS (Figures S1 and S2) provided an estimate of the metrics' sensitivity to observational uncertainty. Adding random noise with 25% of the standard deviation of the original data to the model time series of TWSA and atmospheric variables at each grid cell does degrade the metrics slightly, causing areal averages to be closer to zero, but the differences are relatively small compared to the differences between observed and modeled averages as well as the spread of the ensemble itself. This sensitivity analysis provided evidence that observational errors likely have a relatively small impact on the quality of our satellite-derived metrics.

### 3.5 Evaluating CMIP5 models

Comparison of the explicit TWS field from LENS with the accumulating residuals of the surface water budget (Figure S3), as well as the forcing and response metrics calculated using both (Figures S4 and S5), indicated that the alternative formulation provides an acceptable substitute when an explicit TWS field is not available from an ESM. More specifically, it suggests that water storage in rivers, lakes, and other parts of the terrestrial hydrologic system that are downstream from grid cell-level runoff did not significantly degrade the set of metrics evaluated here. Some degradation of the forcing metrics for PPT was apparent in the middle and low latitudes, but the remaining metrics are not highly sensitive to TWS formulation. This suggests that metrics calculated for CMIP5 output using accumulating residuals could be reasonably and effectively compared with the metrics derived from LENS and the observations (Figure 10).

As with LENS, the metrics derived from CMIP5 output indicated generally stronger coupling metrics than the observations for both the forcing and response limbs. Exceptions include the VPD response metric in the tropics, the boreal PPT and SW↓ forcing metrics, and the midlatitude SW↓ response metrics. The spread between various models was generally greater than the spread within any single model with a multi-member ensemble. Of the four models that use CLM4 for the land surface, the two that use CAM5 for the atmosphere (LENS and CESM1-CAM5) were clustered close together, and exhibited

generally the strongest forcing and response metrics. The two that use CAM4 (CCSM4 and CESM1-BGC) were close to each other, but with lower metrics in both forcing and response than the CAM5 models. The two GFDL models were both within the general ensemble range in the metrics for both VPD and PPT, but GFDL-ESM2M was an extreme outlier in both forcing and response metrics for SW↓.

Comparison of CMIP5 and LENS models indicated a mostly positive relationship between forcing and response metrics in temperate and tropical latitude bands. In boreal latitudes, there was little distinction between the forcing metrics of the different models, all of which were close to zero, though there were some clear differences within the response metrics. In temperate and tropical latitudes, models that showed the strongest forcing metrics generally also showed the strongest response metrics for a given variable. This relationship suggests that analysis of the response limb of the feedback loop is important for understanding how conditions are set up for subsequent forcing via land–atmosphere coupling.

## 4 Discussion

### 4.1 Benchmarking models with observed coupling metrics

The metrics developed here from satellite observations provide a means for evaluating land–atmosphere feedback strength on seasonal to interannual timescales in coupled ESMs. The use of correlation coefficients in this study does not enable a direct assessment of whether the relationships are directly causal, as correlation between atmospheric and terrestrial conditions could result from atmospheric persistence and remote forcing from SSTs (Orlowsky and Seneviratne, 2010; Mei and Wang, 2011). Nonetheless, the satellite-derived metrics provide a meaningful constraint against which coupled models can be benchmarked, as these models need to correctly represent the combined effects of persistence, remote SST forcing, and land–atmosphere coupling.

The forcing metrics, by indicating the relationship between antecedent TWS and subsequent atmospheric characteristics, provide observational constraints to complement previous research in large-scale land–atmosphere coupling in global models (e.g., Guo and Dirmeyer, 2013; Koster et al., 2006; Seneviratne et al., 2013). Observed forcing metrics were found to be strong in some of the regions of intermediate wetness in which ET is limited by terrestrial moisture availability, in addition to some regions in the moist tropics in which ET is generally considered to be energy-limited. Recent observational analyses by (Hilker et al., 2014) demonstrate that at least in the Amazon, deep rooting zone water supplies can become seasonally depleted, leading to a stronger land–atmosphere coupling. This is consistent with findings that deep rooted plants vertically redistribute soil water to shallower layers, allowing higher levels of evapotranspiration to be sustained during the dry season (Lee et al., 2005). It is also consistent with recent work demonstrating that TWSAs can be used as predictors for fire season severity in the Amazon (Chen et al., 2013).

The inclusion of response metrics in our analysis allows the full feedback loop to be considered by recognizing the two-way dependence between the land surface and the atmosphere. The generally higher correlation coefficients in observed response metrics indicates the importance of the land surface response in priming the system for subsequent forcing on the atmosphere. For example, if the TWS response is too strongly coupled to the atmosphere, a small change in atmospheric conditions could yield an unrealistically large change in TWS. The unrealistically large TWS anomaly, in turn, would have the potential to impart a larger land surface forcing of the atmosphere in subsequent time steps. That models and ensemble members with high forcing metrics were also generally found to have high response metrics (Figure 10) highlights the need to consider this.

Both the forcing and response metrics as calculated from the output of the ESMs analyzed in the current study indicated generally stronger coupling compared with those derived from the satellite observations. There are exceptions to this pattern, but it holds generally true, particularly across middle and lower latitudes, and particularly in the LENS data. This is consistent with previous studies conducted at finer temporal resolutions (Ferguson et al., 2012) and across more limited spatial domains (Hirschi et al., 2011). As described below, there are several possible explanations as to why models may simulate a stronger feedback than is observed in the satellite record.

### 4.2 Possible explanations for enhanced feedback strength in models

One set of possible explanations for the stronger coupling metrics in models relative to observations involves models overestimating the amount of water available for ET during the drawdown interval. The land surface influence on the atmosphere requires water to be a limiting factor to ET but not limiting enough to prevent it altogether. Under more moisture-limited conditions, a drawdown interval may experience multiple shorter time periods during which ET is inhibited due to insufficient water, and the terrestrial moisture state exerts no control over flux partitioning. These periods of insufficient moisture would tend to reduce the overall feedback strength integrated across the duration of the drawdown interval. Model shortcomings that make water too readily available for ET could reduce the amount of time spent in a periods of insufficient moisture during the drawdown interval, thereby unrealistically strengthening the longer-term feedback. We note that the opposite could take place under near-saturated conditions if a model overestimates the amount of time in which ET is energy-limited, but we would not expect these conditions to be as prevalent during the drawdown interval that was the time period of focus in our analysis.

ESMs are known to simulate unrealistically homogeneous rainfall intensity, with overestimates of drizzle and underestimates of large infrequent events (Dai, 2006). Infrequent high-intensity rainfall events would yield much more runoff from saturated soil, which would lead to a weaker connection between the land and atmosphere than frequent low-intensity drizzle. If a model simulates too much drizzle, precipitation could lead to too much storage, which would cause a model to overestimate the response metrics. Too much storage also could allow water to be too readily available for ET, causing an overestimate of

the forcing metrics. Contributions from drizzle could be offset if insufficient rainfall intensity does not allow high enough throughfall or soil moisture recharge. The issue of rainfall intensity is related to issues of convective parameterization (described below), and may be addressed in future versions of ESMs through atmospheric superparameterization, in which a model's convective parameterization is replaced with embedded cloud resolving models (Kooperman et al., 2016).

A misrepresentation of either the amount of bare soil or of bare soil processes also could lead to overestimates of the amount of water available for ET and thereby coupling strength. Current land surface schemes of ESMs are based on the "big leaf" model paradigm, which could lead to overestimates of ET if runoff and groundwater recharge are underestimated as a consequence of too small of a bare soil fraction. In addition, even if bare soil fraction were correct, overestimates of ET due to an incomplete representation of surface resistance of bare soil, as found in CLM by Swenson and Lawrence (2014), would amplify positive feedbacks.

Additional explanations for why models may overestimate feedback strength include the parameterization of convection in the PBL or stomatal conductance responses to soil moisture. Previous work using a regional climate model (RCM) with a higher spatial resolution have determined that convective parameterizations are as important as spatial resolution in the simulation of precipitation coupling (Hohenegger et al., 2009). Taylor et al. (2013) similarly found parameterized convection in an RCM yielding a positive coupling in contrast to the negative coupling found in both observations and model runs with explicitly simulated convection. If negative coupling mechanisms are present in reality but absent from models, this could contribute to an overestimate of coupling metrics and underrepresentation of negative feedbacks in models. Similarly, the diversity of stomatal conductance parameterizations in CMIP5 ESMs is relatively low (Medlyn et al., 2011; Swann et al., 2016, in press), and if stomatal apertures close too rapidly in response to an initial deficit in terrestrial water storage, transpiration–humidity feedbacks may be intensified in an unrealistic manner.

One factor that could contribute toward stronger coupling metrics in models relative to observations is the effect of observational uncertainty combined with a relatively short time series. Adding random error to one or more variables in a correlation analysis will reduce the correlation coefficient, and this degradation has been shown to be sensitive to the length of data sets used to establish metrics of land–atmosphere coupling (Findell et al., 2015). Given the relatively short time series available for the current analysis, the correlation coefficients derived from remote sensing data may be reduced due to observational uncertainty, unlike those derived from internally consistent models. We found that adding random noise to LENS at 25% of the standard deviation of the original data caused some degradation of our area-averaged coupling metrics, but only by a small amount relative to the difference between LENS and the observations (Figures S1 and S2). We chose 25% as a qualitative upper bound on likely uncertainties introduced from random observational error within the TWSA and atmospheric variable time series. This highlights the need for developing more quantitative error estimates in remote sensing

and reanalysis products. More generally, this sensitivity analysis suggests that our coupling metrics, when averaged across large areas, may be useful in identifying robust data-model differences.

Another possible explanation stems from the fact that our coupling metrics include covariability due to atmospheric persistence and remote forcing by SST (Orlowsky and Seneviratne, 2010; Mei and Wang, 2011) alongside the direct influence of land-atmosphere interactions. For this reason, we caution that overestimates of coupling metrics do not imply that the land–atmosphere feedback is necessarily stronger, but could be due to an overestimate of SST-driven correlations between the land surface and the atmosphere. Wei et al. (2008) demonstrated that negative correlations between soil moisture and subsequent precipitation can be explained by precipitation persistence combined with negative temporal autocorrelation of precipitation associated with intra-seasonal modes such as the Madden-Julian Oscillation (MJO). Poor representation of the MJO period in CMIP5 models leads to unrealistic patterns of precipitation persistence (Hung et al, 2013). If models are failing to capture MJO-driven negative correlations, this could lead to overly strong positive correlations relative to observations. However, this would depend on the length of the drawdown interval relative to persistence time and the period of intra-seasonal modes.

**4.3 Uncertainties and future applications**

The current study demonstrates the utility of the coupling metrics presented here, but conclusions are limited by the time span of the satellite record. While LENS enables the internal variability of these relationships to be investigated within the model, it is unclear how much natural climate variability affects these relationships in reality on timescales longer than the satellite record. Furthermore, we acknowledge that observational error over an insufficiently long time series could reduce the apparent strength of correlations (Findell et al., 2015). Therefore, the utility of the coupling metrics we present will increase alongside the length of the time series available from remote sensing platforms. This emphasizes the importance of the GRACE follow-on mission (Flechtner et al., 2014) and the need for continuity in the record between missions.

Furthermore, incorporating additional remote sensing products can reduce uncertainties inherent in the satellite-derived data sets. We presented metrics derived using ERA-Interim in place of AIRS, GPCP, and CERES in order to qualitatively illustrate this uncertainty. We found a non-negligible amount of uncertainty in both forcing and response metrics due to inconsistencies between the remote sensing and reanalysis products. Future work will address these uncertainties by incorporating additional observations and observationally constrained data sets such as those from the Global Soil Wetness Project (Dirmeyer et al., 2006b) and the Global Land Data Assimilation System (Rodell et al., 2004). In addition, as increasingly long time series of data become available from the Soil Moisture-Ocean Salinity (Mecklenburg et al., 2012) and Soil Moisture Active Passive (Panciera et al., 2014) missions, the metrics developed here can be applied to those data sets as well, which will elucidate the importance of surface soil moisture relative to the total TWS column in these interactions.

Finally, the issue of causality and the possibility that correlations result primarily from atmospheric persistence and remote forcing from SST rather than land-atmosphere interactions may be addressed using sensitivity experiments similar to those of GLACE and GLACE-CMIP. While the previous experiments have tested the importance of soil moisture interaction with the atmosphere, additional experiments could expand upon these methods by treating SST variability similar to terrestrial soil moisture availability. Such experiments could determine the relative importance of remote SST forcing, including the effect of atmospheric persistence, and local land–atmosphere coupling in explaining correlations between TWS and atmospheric conditions.

As these sources of uncertainty are diminished, the coupling metrics introduced here may be used to assess whether improvements to model biogeophysics and parameterizations yield relationships that are more consistent with observations. CMIP5 models are known to have a high ET bias (Mueller and Seneviratne, 2014), which could be due in part to the explanations proposed as possible reasons for overestimated coupling metrics in models. As data become available from Phase 6 of the Coupled Model Intercomparison Project (CMIP6), these metrics could provide an assessment of whether improvements to ET processes in models also improves the relationship between the land surface and the atmosphere.

**5 Conclusion**

We have developed a new approach for measuring the strength of the two-way feedback relationships between TWS and the atmosphere. This approach was designed specifically to take advantage of TWSA data from the GRACE mission, along with concurrently collected remote sensing and reanalysis data sets of atmospheric variables, in a manner that could then be applied to earth system models. The coupling metrics described here quantify the relationships between both antecedent TWS and subsequent atmospheric conditions, as well as antecedent atmospheric conditions and subsequent TWS.

Regions of strong forcing, in which the TWSA at the beginning of the drawdown interval was related to the subsequent atmospheric state, coincided with the semi-arid zones previously found to be hot spots of land atmosphere coupling, as well as some new tropical zones that may have moisture limited ET regimes. Regions of strong response metrics, in which the TWSA at the end of the drawdown interval is related to the atmosphere, are much more widespread. Modeled coupling metrics are generally found to be stronger than those observed in the satellite record. If this discrepancy is due to models overestimating the two-way feedback between the land surface and the atmosphere, this could lead to models incorrectly projecting future warming trends and climatic extremes (e.g., Hirschi et al., 2011; Seneviratne et al., 2013; Cheruy et al., 2014; Miralles et al., 2014).

The results of this study are consistent with previous studies at smaller temporal scales indicating land–atmosphere coupling strength may be stronger in models than in observations. There are several possible mechanisms that may contribute to the

overestimation of land–atmosphere coupling in models, and future studies may incorporate the metrics introduced here to assess the role of these mechanisms. These metrics will become increasingly useful as the temporal coverage of the remote sensing record grows longer and additional missions come online.

**Acknowledgements.** We received funding support from the United States Department of Energy Office of Science Biogeochemistry Feedbacks Scientific Focus Area and the Climate Change Prediction Program, Cooperative Agreement (DE-FC03-97ER62402/A010). The CESM Large Ensemble Community Project was supported by the National Science Foundation (NSF) with supercomputing resources provided by the Climate Simulation Laboratory at NCAR's Computational and Information Systems Laboratory (CISL). We also acknowledge the World Climate Research
Programme's Working Group on Coupled Modelling, which is responsible for CMIP, and we thank the climate modeling groups (listed in Table 2 of this paper) for producing and making available their model output. For CMIP the U.S. Department of Energy's Program for Climate Model Diagnosis and Intercomparison provides coordinating support and led development of software infrastructure in partnership with the Global Organization for Earth System Science Portals.

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

**Table 1:** Remote sensing and reanalysis products used for analysis.

| Variable | Abbr. | Data product | Spatial resolution | Temporal resolution | Reference |
|---|---|---|---|---|---|
| Terrestrial water storage | TWS | GRACE Tellus RL05.1 | 1° | monthly | Landerer and Swenson (2012) |
| Vapor pressure deficit | VPD | AIRS AIR3XSTM v6 | 1° | monthly | Susskind et al. (2014) |
| Precipitation | PPT | GPCP v2.1 | 1 ° | daily | Huffman et al. (2009) |
| Downwelling shortwave radiation | SW↓ | CERES EBAF Ed2.8 | 1° | monthly | Loeb et al. (2009) |

**Table 2:** CMIP5 models used for analysis.

| Model acronym | Atmospheric model | Land surface model | Horizontal resolution | Ens. size | Reference(s) |
|---|---|---|---|---|---|
| CCSM4 | National Center for Atmospheric Research (NCAR) Community Atmospheric Model version 4 (CAM4) | Community Land Model (CLM4) | 288 x 192 | 6 | Lawrence et al. (2011); Neale et al. (2013) |
| CESM1-CAM5 | NCAR Community Atmosphere Model version 5 (CAM5) | CLM4 | 288 x 192 | 3 | Lawrence et al. (2011); Meehl et al. (2013) |
| CESM1-BGC | NCAR CAM4 with biogeochemistry | CLM4 | 288 x 192 | 1 | Lawrence et al. (2011); Lindsay et al. (2014); Neale et al. (2013) |
| IPSL-CM5A-LR | Laboratoire de Météorologie Dynamique atmospheric model (LMDZ5A) | Organizing Carbon and Hydrology in Dynamic Ecosystems (ORCHIDEE) | 96 x 96 | 3 | Cheruy et al. (2013); Dufresne et al. (2013); Hourdin et al. (2013) |
| GFDL-ESM2G | Geophysical Fluid Dynamics Laboratory (GFDL) Earth System Model 2 (ESM2) | Land Model 3.0 (LM3.0) | 144 x 90 | 1 | Dunne et al. (2012); Shevliakova et al. (2009) |
| GFDL-ESM2G | GFDL ESM2 | LM3.0 | 144 x 90 | 1 | Dunne et al. (2012); Shevliakova et al. (2009) |

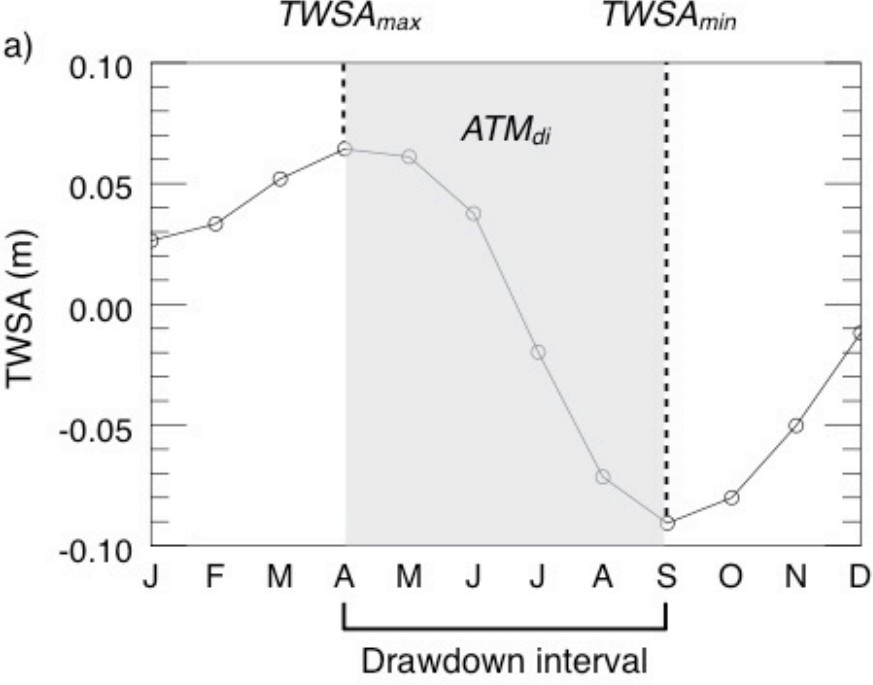

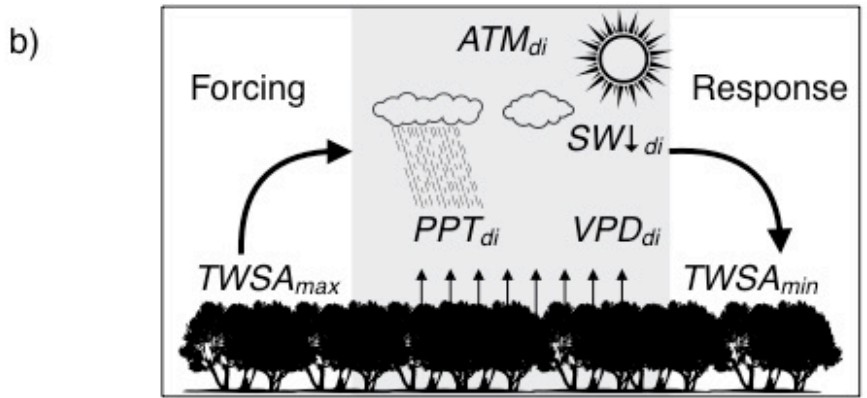

**Figure 1:** Conceptual description of coupling metrics: a) Example TWSA climatology from a typical mid-latitude location in central North America (38° N, 92° W) illustrating the definition of the drawdown interval as the months from the maximum TWSA through the minimum TWSA. $TWSA_{max}$ and $TWSA_{min}$ are the TWSA values (in units of water height) during the maximum and minimum months respectively, and $ATM_{di}$ is the atmospheric variable of interest averaged across the months of the drawdown interval. b) Representation of the interactions between TWS and atmospheric component, demonstrating the forcing limb of the feedback loop, in which $TWSA_{max}$ forces subsequent atmospheric conditions, as well as the response limb, in which $TWSA_{min}$ responds to the atmospheric state during the drawdown interval.

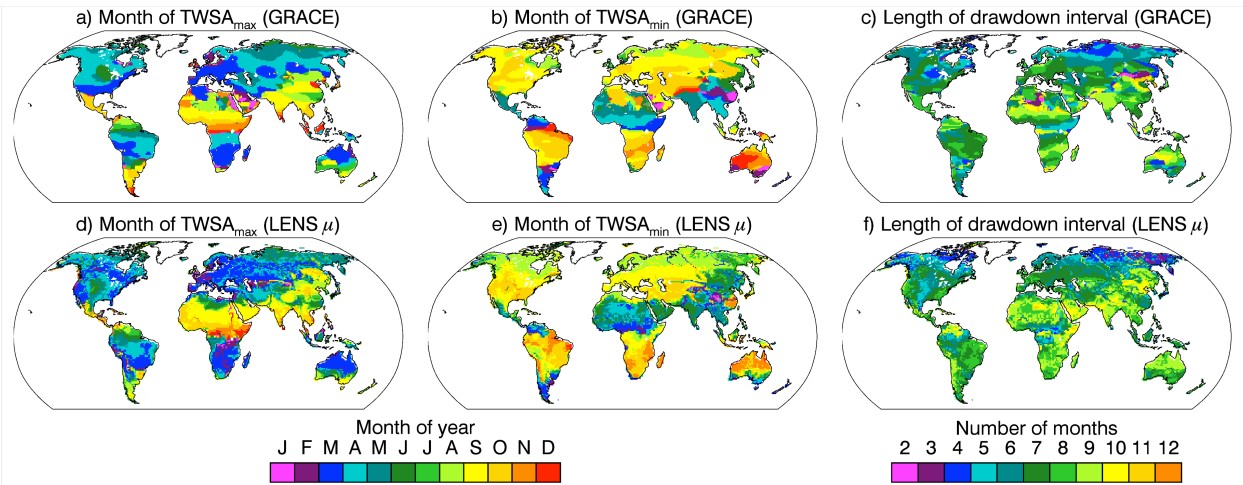

**Figure 2:** Month of maximum and minimum TWSA and the length of the drawdown interval from GRACE (a–c) and the LENS ensemble mean (d–f). Months of maximum and minimum were based on the climatology of detrended TWSA over the 146 months in the GRACE record.

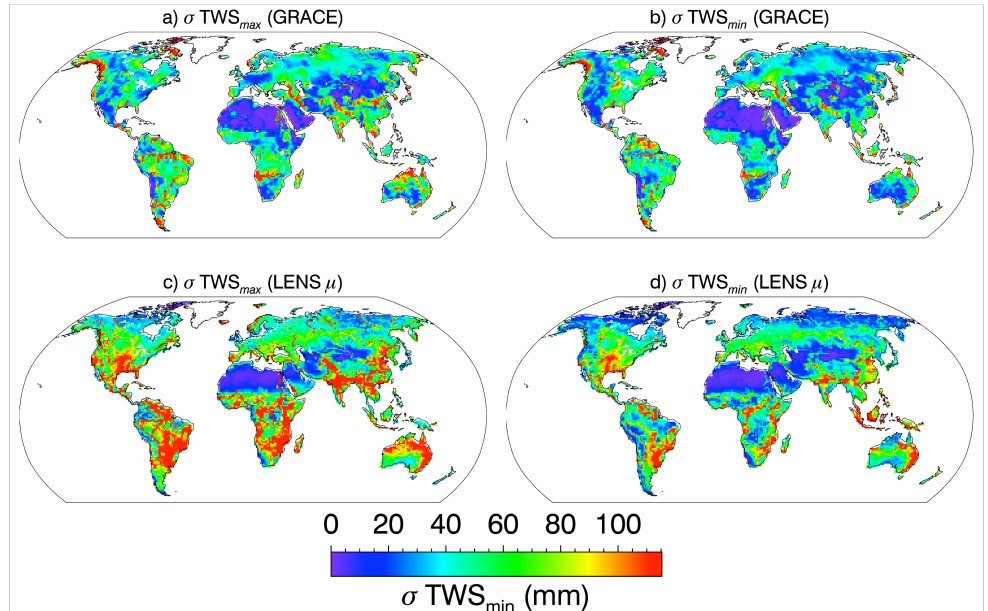

**Figure 3:** Interannual variability (standard deviation) of $TWSA_{max}$ and $TWSA_{min}$ from GRACE (a and b) and the LENS ensemble mean (c and d).

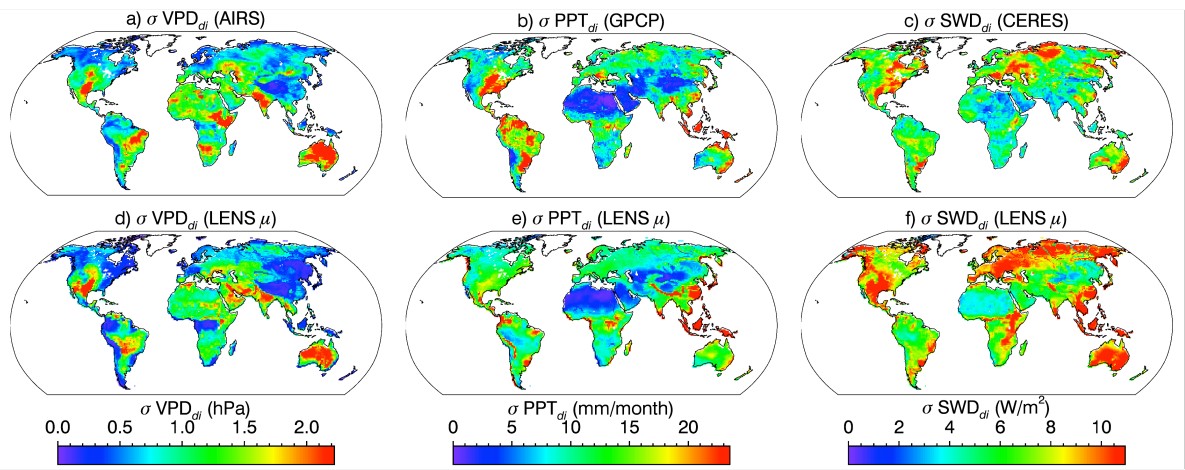

**Figure 4:** Interannual variability (standard deviation) of $VPD_{di}$ from AIRS (a), $PPT_{di}$ from GPCP (b), $SW\downarrow_{di}$ from CERES (c) and the equivalent quantities from the LENS ensemble mean (d–f).

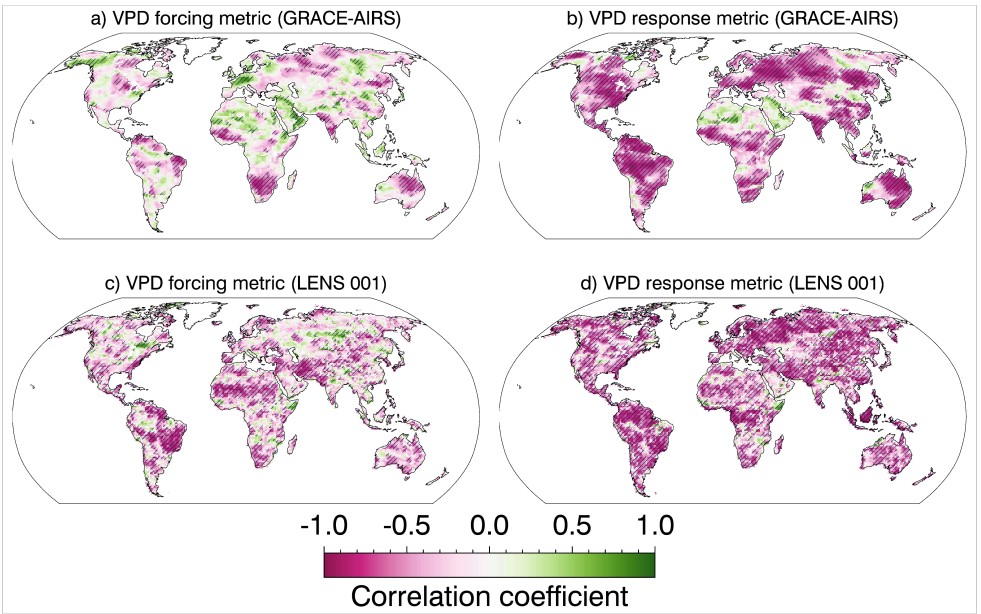

5    **Figure 5:** Forcing and response metrics for VPD from GRACE/AIRS (a and b) and LENS ensemble member 001 (c and d). Crosshatching indicates a correlation coefficient that is statistically significant at $p \leq 0.05$ (one-tailed Student's t-test).

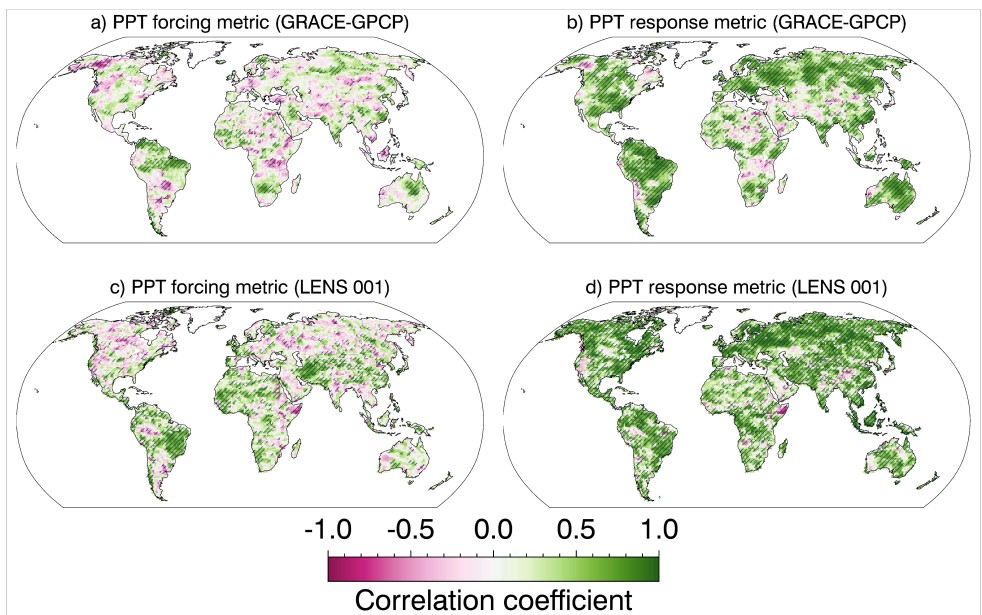

**Figure 6:** Forcing and response metrics for PPT from GRACE/GPCP (a and b) and LENS ensemble member 001 (c and d). Crosshatching indicates a correlation coefficient that is statistically significant at $p \leq 0.05$ (one-tailed Student's t-test).

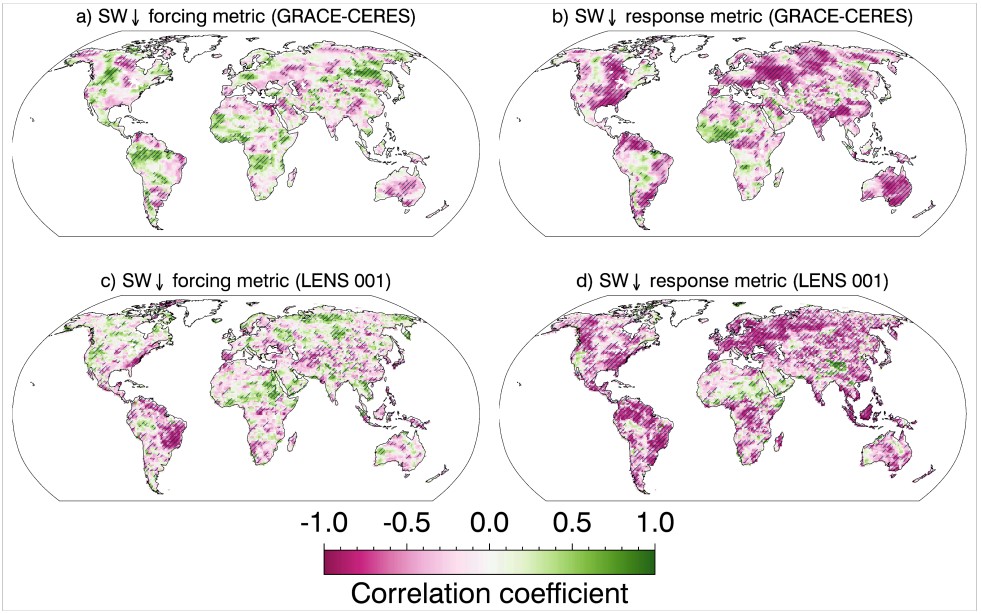

5   **Figure 7:** Forcing and response metrics for SW↓ from GRACE/CERES (a and b) and LENS ensemble member 001 (c and d). Crosshatching indicates a correlation coefficient that is statistically significant at $p \leq 0.05$ (one-tailed Student's t-test).

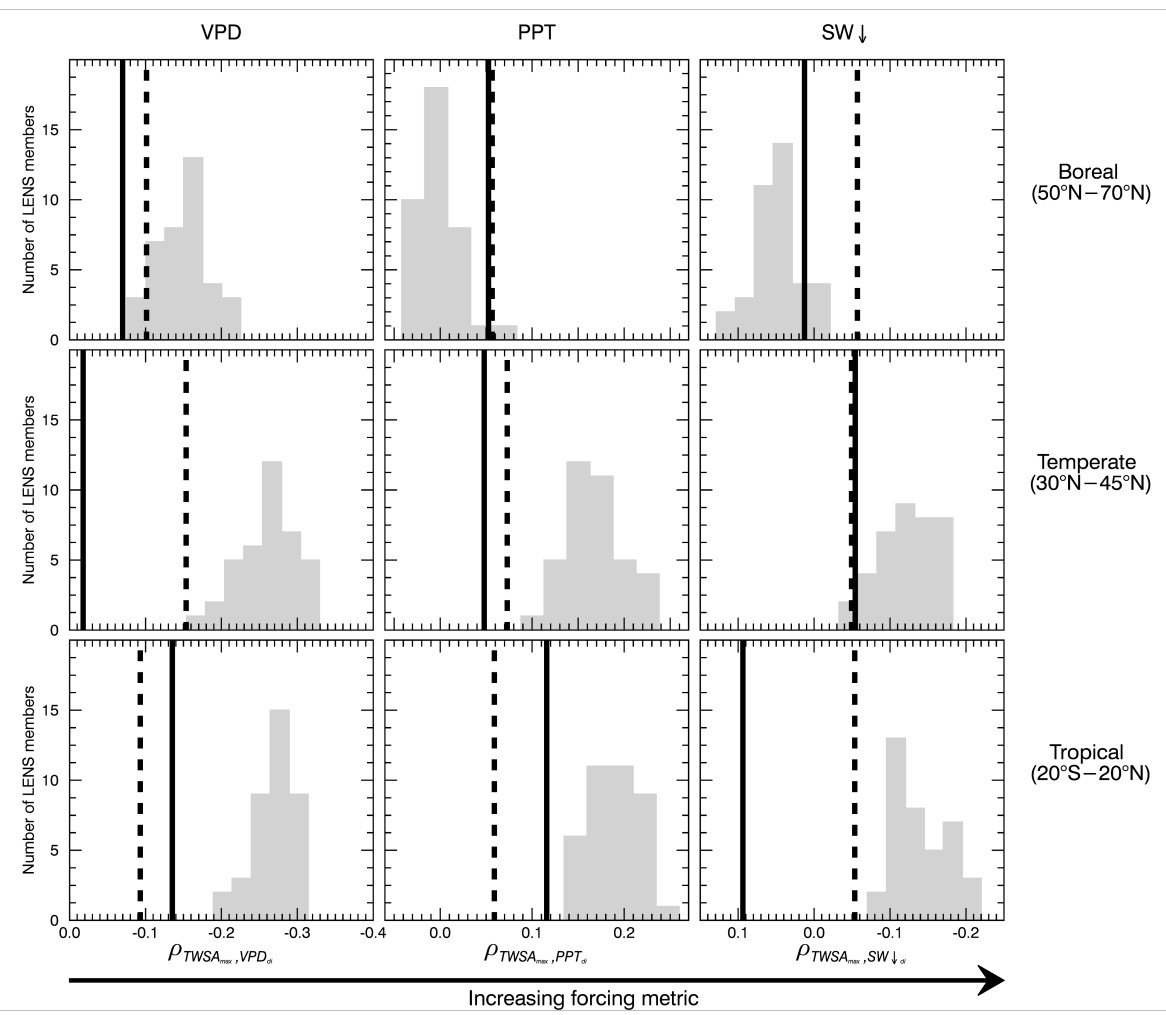

**Figure 8:** Ensemble histogram of forcing metrics from the 38 simulations in LENS (grey bars) compared to satellite observations from GRACE/AIRS/GPCP/CERES (solid black line) and the alternate set of observations from GRACE and ERA-Interim (dashed black line), averaged across land regions within different latitude bands.

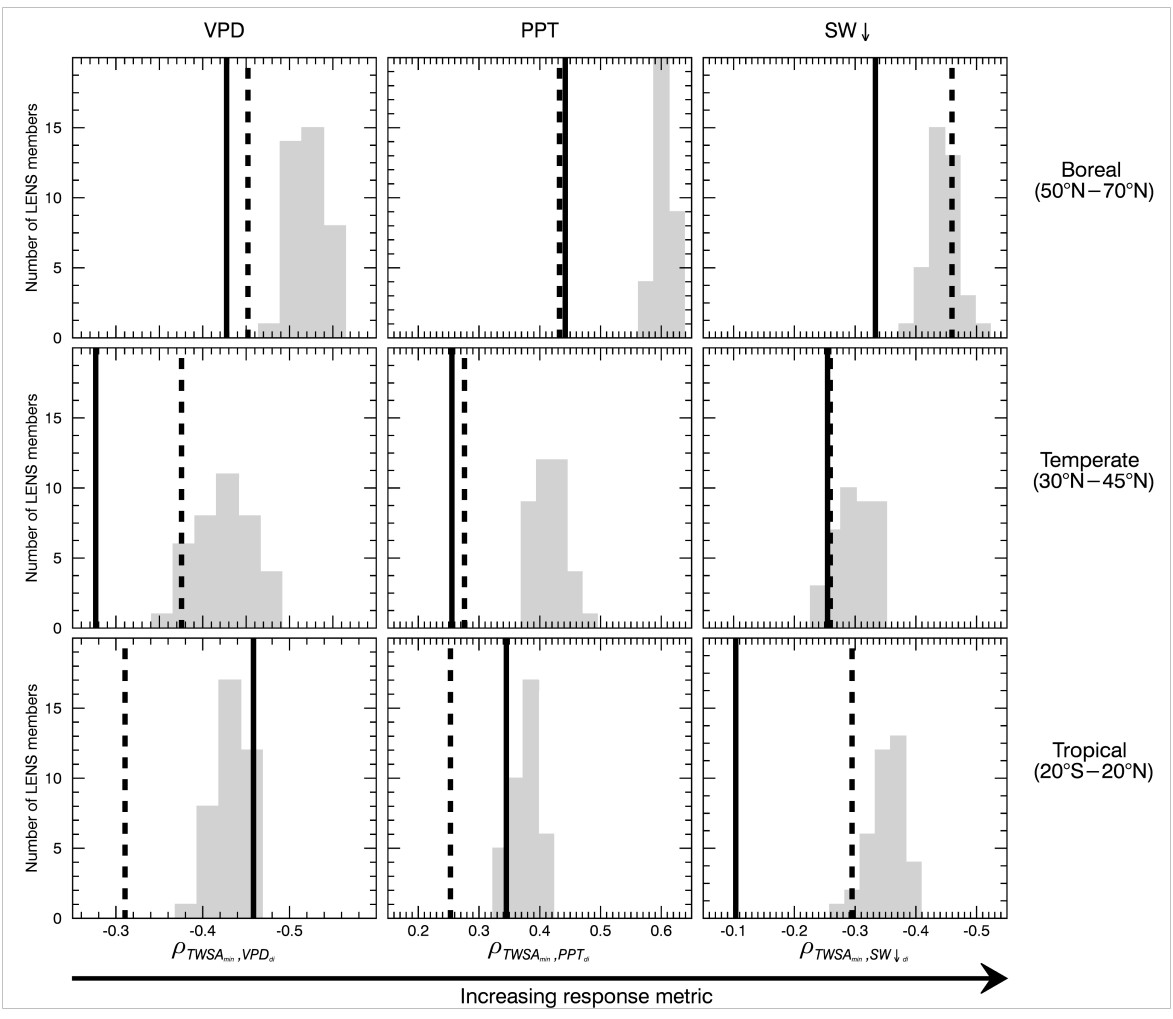

**Figure 9:** Ensemble histogram of response metrics from the 38 simulation in LENS (grey bars) compared to satellite observations from GRACE/AIRS/GPCP/CERES (solid black line) and the alternate set of observations from GRACE and ERA-Interim (dashed black line), averaged across land regions within different latitude bands.

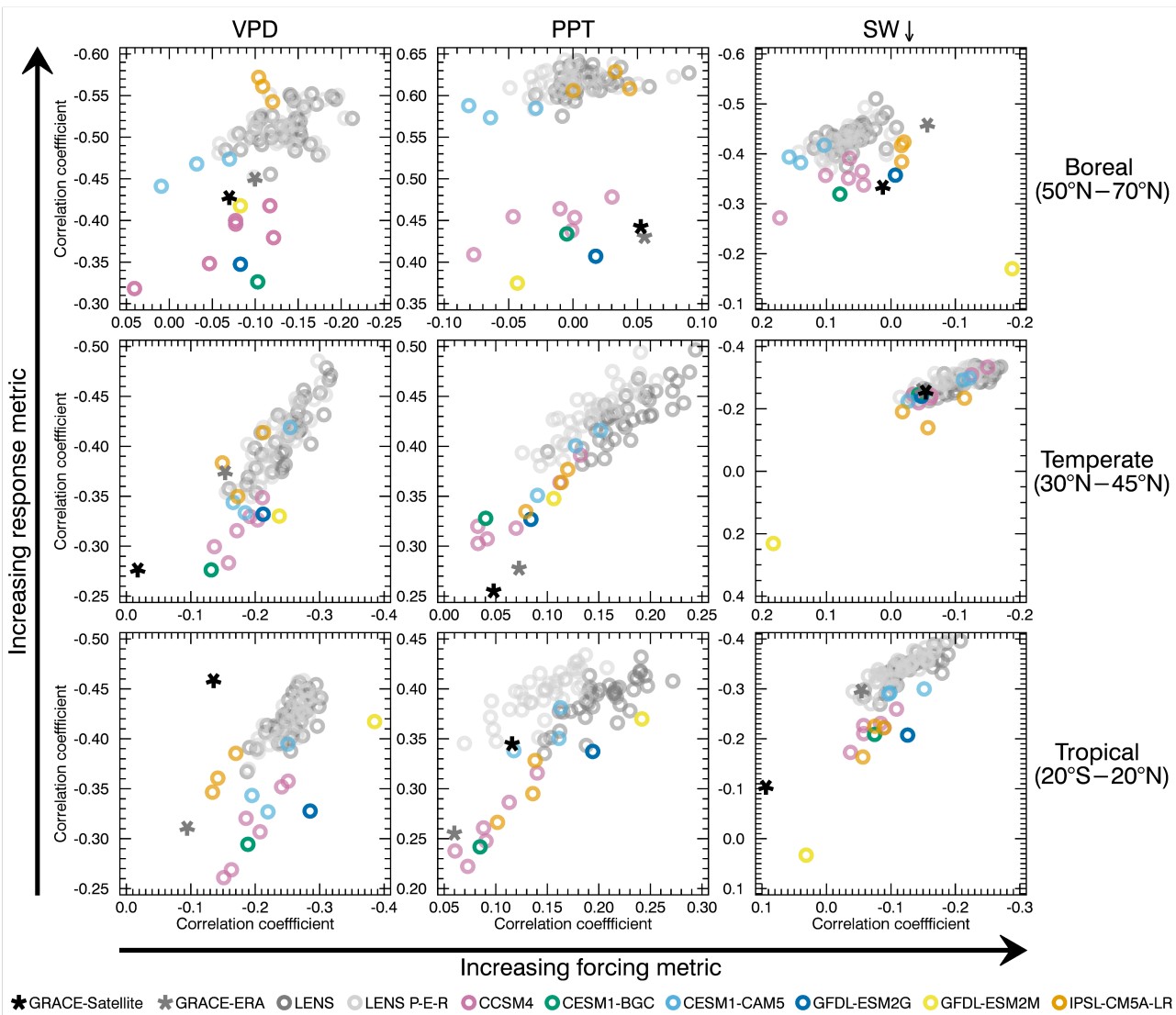

**Figure 10:** Scatter plots of forcing and response metrics for LENS and CMIP5 models with observations, averaged across land regions within different latitude bands. For LENS, we show metrics calculated using the explicit TWS output (darker grey) and TWSA estimates from the accumulated residuals of the surface water balance (lighter grey). For CMIP5 models, we calculated metrics using TWSA estimates from the accumulated residuals of the surface water balance.