# Peer review of "Evaluating the strength of the land-atmosphere moisture feedback in earth system models using satellite observations"

_Hydrology and Earth System Sciences, 2016_

## Referee Comment (RC1) · T. Ford (Referee) · 17 Jun 2016

General Comments

This manuscript evaluates the strength in the two-way relationship between total water storage (TWS) and atmospheric conditions within a suite of satellite remote sensing, reanalysis, and climate model systems. The correlation method for evaluating the coupling strength is not novel, but the concurrent assessment of both response and feedback "limbs" is. Further, comparison of the response and feedback strengths with model and satellite datasets provides additional evidence of the dichotomy in the apparent strength (and in some documented cases, the sign) of soil moisture-atmosphere coupling between model and observational frameworks. As the authors allude to,

these model system issues can influence estimates of future warming/drying trends. I enjoyed reading this very well written manuscript. The results and implications are international in scope, and it will make a worthy contribution to the land-atmosphere interaction body of knowledge. The outstanding issues I outline below are mostly minor, and the manuscript will be suitable for publication once they are addressed/clarified appropriately.

Specific Comments

Methods: Page 5, line 16: "with temporal gaps filled using linear interpolation." More detail is needed here. How many months in the data record are filled? What was the typical time interval of missing data; for example, were the "gaps" filled predominantly just 1 month or multiple consecutive missing months of data? Is a linear interpolation reasonable for temporal gap-filling GRACE data?

Page 5, lines 20-22: "...the use of TWS data allows us to include surface storage, canopy storage... all of which may be sources of moisture that are potentially limiting factors for ET". This is just one specific instance of discussing TWS for land-atmosphere analysis, but in general I think the paper would be better served with a more complete discussion of the advantages and limitations of using TWS for this purpose. For example, is the groundwater component of TWS a limiting factor for ET if the rooting depth does not reach the water table? In many temperate agrarian or grassland landscapes this is the case. How do you think these issues could potentially affect the results, or more specifically do you think your findings would be different if you just used (e.g.,) surficial soil moisture or root-zone soil moisture?

Results: Page 10, lines 23-26: you find consistently weaker forcing relationships in boreal regions and attribute this to "high levels of climate variability in many high latitude regions" because of AO, NAO, and the like. However, do you think the predominantly energy-limited evaporative regime of many boreal regions contributes to limiting the feedback connection between soil moisture and atmospheric conditions? I would not

expect the forcing "limb" to match the strength of the "response" limb in such conditions where evaporative fraction is more a function of incoming radiation than soil moisture availability.

Page 11, lines 28-29: "Furthermore, the well-understood physical mechanisms allow causality to be inferred even when not directly demonstrated." The argument in the previous sentence is well-taken, but I disagree with the premise of this statement, particularly inferring causality in the forcing "limb" in the absence of discussion/analysis of possible confounding effects. At the very least I would like to see some assessment or acknowledgement of the role of atmospheric persistence as a confounding factor when quantifying the forcing limb. For example, is a strong forcing limb caused by the physical constraint of soil moisture on energy partitioning and modification of boundary layer dynamics and thermodynamics (as suggested by the authors), or is it the result of large-scale atmospheric circulation and persistence of synoptic-scale patterns that modify precipitation and atmospheric demand throughout the duration of the TWS draw down season? A correlation coefficient cannot adequately address the question of large-scale atmospheric persistence vs. soil moisture feedback, and indeed this is beyond the scope of this manuscript. However, this does also mean that one cannot infer causality and mechanistic connections between soil moisture and VPD/PPT/SW based on the evidence provided here.

Page 12, lines 14-16: "That models and ensemble members with high forcing metrics were also found to have high response metric... highlights the importance of the land surface response in priming the system for subsequent forcing on the atmosphere..." I don't understand how the land surface response "primes the system" for subsequent forcing when your analysis (and Figure 2) suggest the forcing occurs prior to the response. What am I missing? Can you please expound?

Conclusion: Page 16, lines 13-14: "...which suggests that some of these models may have difficulty properly predicting warming trends and climatic extremes." You include an excellent discussion of the potential links between model overestimation of land

surface forcing and warming trends, founded in the body of literature. However, you do not explicitly quantify this linkage in this manuscript. The ability of models to "properly" predict warming trends and climatic extremes is not evaluated here, so this statement should probably be removed.

Technical Corrections:

Page 11: I'm going to guess the section 4 title should be "Discussion" and not "Introduction"

Page 12, line 21: parentheses should only be around the publication year - Guo et al., (2006)...

---

## Referee Comment (RC2) · Anonymous Referee #2 · 21 Jun 2016

General comments:

This paper describes a new methodology for the analysis of land-atmosphere coupling that focuses on the seasonal time scale and that addresses both the widely-studied impact of the land surface on the atmosphere and the somewhat less often considered (in this context) sensitivity of the land surface to atmospheric forcing. This methodology is applied to remote-sensing data of terrestrial water storage (TWS), the CESM Large Ensemble (LENS) and a few CMIP5 models.

Although many methodologies to quantify land-atmosphere coupling already exist, the focus on seasonal time scales and the investigation of both the forcing and feedback loop of land-atmosphere coupling make this methodology valuable. The use of LENS

is also valuable and in particular is useful to consider potential impacts of internal variability on the metrics.

The manuscript is very well written and very clear. The authors have done a good job at keeping the text relatively short and straightforward despite substantial analyses. Most of my comments below are minor and I think the manuscript can be published after considering these.

Specific comments:

- Page 4, line 16-17: "Until recently, studies using remote sensing data to look for evidence of land–atmosphere coupling relied on data products that provide information about surface soil moisture." There are some exceptions and some remote-sensing products also estimate soil moisture in the whole root zone (for instance, I recall that Guillod et al. (2015) used such estimates).

- Section 3.3: These results on the role of internal variability are interesting. Another source of discrepancy between a single model run and observations can be observational error combined with short record (e.g., Findell et al., 2015), which may artificially decrease the coupling found in observations. Together, this effect and internal variability probably tell us something about the data length required to robustly assess land-atmosphere coupling.

- Page 8, line 24 and Page 11, line 5: Supplementary figures references could be more specific. I think that the first mention on page 8 refers to Fig. S3 while the second mention (page 11) refers to Figs. S1 and S2. The order of the supplementary figures could also be adapted so they are cited in sequence.

- Page 13, line 20-29: The authors write that overestimating ET would lead to excessive land-atmosphere coupling. This is a bit confusing: if the coupling is strongest in transitional regions between wet and dry climate, why would a too high ET in an already wet region lead to an increase in coupling? I agree that model errors with respect to the distribution of water between storage reservoirs can be an issue; however I do not think that this necessarily leads to stronger coupling, but maybe I am missing something here.

Technical corrections:

- Page 11, line 23: I assume that "Introduction" should in fact be "Discussion".

- Page 12, line 22: "(Guo et al., 2006) explain..." should be "Guo et al. (2006) explain...".

- Page 13, line 13: "and underestimation of and remote SST forcing" should be "and underestimation of remote SST forcing" (?).

- Page 16, line 11: "Regions of strong coupling metrics": I think the authors mean "Regions of strong response metrics" here.

References:

- Findell, K. L., et al., 2015. Data Length Requirements for Observational Estimates of Land-Atmosphere Coupling Strength. J. Hydrometeorol. 16, 1615–1635.

---

## Referee Comment (RC3) · Anonymous Referee #3 · 23 Jun 2016

In this study, the authors aim to assess the feedbacks between terrestrial water storage (TWS) and the atmosphere in remote sensing observations (using GRACE data for TWS), and they aim to use the estimate they obtain to evaluate climate model simulations from the CESM large-ensemble and from (a subset of) CMIP5 models. They use correlations at the interannual time-scale, with lead-lags of a few months between TWS and atmospheric variables, to assess both how the atmosphere (during months of TWS decrease) influences the minimum annual TWS (the forcing response), and how TWS (at time of maximum annual TWS) feeds back on the atmosphere in the following months (the feedback response). Their main result include some characterization of these relationships in model and observations, which show that the CESM model, and

most CMIP5 models, appear to overestimate both forcing and feedback relationships. Some discussion is then proposed on the possible reasons for that.

Although I like the idea of trying to use GRACE observations to derive a large scale "land-atmosphere moisture coupling" metrics to compare to models, I see a number of issues with the study as it stands, that I think warrant some major revision.

1) My first main comment has to do with the metric used. First, the brevity of the time period of analysis is obviously an issue. The metrics are essentially interannual correlations over 13 years, i.e., correlations over 13 data points. That's very short. I am not sure I have seen many studies looking at interannual variability over 13 years only, in particular in terms of land-atmosphere studies. Accordingly, results for the feedback metrics appear very noisy. First, I believe a discussion of field significance is warranted here: patches of apparently significant values may still be random in that context (e.g., see Livezey and Chen (1983)). Second, note that recent research underscores the need for long-record datasets to establish land-atmosphere coupling, that coupling metrics require more data than single-variable simple statistics (e.g., mean and variance) to be robustly estimated, and finally that, unlike single-variable statistics, coupling metrics are actually degraded by observational uncertainty (Findell et al. 2015). The latter point, in particular, is in my view a much likelier explanation for the weaker correlations found here in observations – between uncertain observation datasets that are independent of each other - compared to correlations computed with model outputs, which are by definition perfectly consistent with each other. The authors touch on the issue of observational uncertainty by computing correlations with the ERA reanalysis, but I don't think enough is made of that. So, given the brevity and uncertainty of observations, even without consideration of any other issues (but, see below), I am really uncomfortable with the approach proposed by this paper, which is to consider the observational estimate as a benchmark for model evaluation. Personally, I think an approach where observations and model results are used together to try to infer the 'real' coupling would make more sense here. But, this would lead to a very

different paper. Overall, if the authors are going to go on with their approach, I would recommend much more caution in how things are presented, including in the title of the study and the conclusions.

2) Another issue with the metrics involves the definition of the (feedback) metric. The way it is defined, it is looking at the impact of TWS at the end or peak of the rainy season on climate in the subsequent months. The authors indicate as much, and say they want to consider, in the Tropics, the impact of late-rainy season TWS on dry-season climate. I see two issues with that. First, in my view, while that may useful in the deep Tropics where the dry season is short, this approach is problematic in monsoon regions, or regions of the Tropics that have a well defined rainy season (i.e., outside of the deep Tropics): basically, after the rainy season, there is not much rain to look at any more. For instance, over the Sahel, what the authors are computing is the impact of September TWS on precipitation over September-May. But it doesn't rain much over that time period. In my view, it would be much more interesting to look at the impact of end-of-dry season TWS on the subsequent rainy season to see if, in these regions, available land moisture feeds back on precipitation during the rainy season. Second, in the same example over West Africa, whatever rainfall there is over Sept-May is actually probably the end of the monsoon, Sept-Nov. Because TWS in September is likely to be influenced by precip in September, and Sept. precip is likely to represent large part of the 'response' variable, the causation is muddied a little bit: a clearer temporal offset would be needed in such a case. But more importantly, even precip in the months following September (Oct, Nov) is likely to be correlated with precip in the previous months – for instance, a year with a strong monsoon that has more rain in Jun-Sept may well tend to also have more rain in Sept-Nov. Because September TWS will largely reflect JJAS rainfall, the TWS-based metric will then show a strong feedback - but the inferred causation would be a misinterpretation. This brings me to a more general point: the authors do not discuss how autocorrelation, here at the seasonal time scale, of climate variables, may impact their estimate of land-atmosphere coupling. This is a major issue affecting all empirical studies of land-atmosphere coupling – see, for instance, Wei et al. (2008) and Orlowsky and Seneviratne (2010). The authors do cite the latter study, but, it seems, simply to say that if models underestimate SSTs influence on land climate, they will then appear to overestimate local L-A coupling. They somehow miss the point of that paper in how it should apply to to their own results. I just gave one example above (the Sahel) of how that might be the case. Other monsoon regions (e.g., India) might similarly be affected. Interannual variability in the coupled ocean-atmosphere (eg., ENSO) might also the source of confounding influence at the time scales investigated here. So, overall, I recommend these caveats be considered and discussed by the authors in their interpretation of their results. Personally, I would need to see some further analysis to be more convinced of the physical reality of the land-atmosphere feedbacks the authors claim to show (e.g,. some sensitivity test to the months and time lags considered, some investigation of atmospheric variability and persistence, etc.).

3) Another main comment has to do with the discussion section. The authors discuss why models might exhibit stronger feedback (and forcing) metrics than observations. As mentioned above, I think uncertainty in observations should be mentioned as a primary reason. The authors propose that ET may be consistently overestimated in climate models, and a large part of the discussion then consists in speculation as to why that may be the case. First, while I appreciate the effort to discuss things further and not just show results, I found this whole section a bit too speculative. IF the models overestimate ET, then IF stomatal conductance, IF convection, IF bare soil, etc... Can the authors actually point to any evidence that ET is consistently overestimated in climate models, in the first place (or at least in CESM)? Second, if soil water is too readily available in models, and ET is overestimated, wouldn't that actually mean that feedbacks should be underestimated in models? Indeed, ET would then be less water-limited and more energy-limited, with less potential for soil moisture-atmosphere feedbacks. Surface climate variability would then be influenced by the atmosphere to a greater extent. Along the same lines, the authors claim that their results, showing an overestimation of land-atmosphere feedback by models, are consistent with prior

studies, and have implications for projected warming (e.g., Cheruy et al., 2014). However, these previous studies, it seems to me, point to ET being underestimated in these models, and models getting to easily "locked" in a dry and warm surface mode. So, in effect, while the authors agree with prior studies that land-atmosphere feedbacks are overestimated in models, they provide opposite reasons for that (overestimated versus underestimated of ET). I would like to see the authors clarify that apparent contradiction.

4) Finally, the author interpret the relationship they find between the strength of the feedback and forcing metrics in CMIP5 models as showing that: " the response limb of the feedback loop is important for understanding how conditions are set up for subsequent forcing via land–atmosphere coupling". They claim that it highlights "the importance of the land surface response in priming the system for subsequent forcing on the atmosphere". I am not convinced by this interpretation, which sounds a bit hand-wavy to me. I don't see a strong physical reason why a model where, for instance, TWS responds strongly to precipitation, should have a strong feedback of TWS onto precipitation. Couldn't the relationship on Figure 10 be due to intermodel differences in what TWS (or its estimate, here) encompasses in each model? For instance, different soil depths? A deeper soil would lead to weaker links between TWS and climate both in terms of response and feedback to the atmosphere. In any case, I found Figure 10 to be insufficiently explained and encourage the authors to discuss this further.

Here are some further comments along the text:

- p.2 line 4: "cloud radiative coupling": please explain and clarify.

- P.2 line 24: actually, no: a surprising result of GLACE II was that predictive skill was not enhanced over the Great Plains "hot spot" from GLACE I, but rather to the North of it (see Koster et al. 2011). Consider rephrasing.

- P.3 line 5: the text should make it clear that GLACE-like metrics cannot be directly compared to observations, and that other more simple metrics, not strictly equivalent,

have to be used, like SM-ET correlations, etc.

- P.3 line 20: Findell et al. 2011 is actually based on reanalysis data, not "modeling". Also, Findell et al. 2015 should be included in this discussion, to highlight the issue, discussed above, of data length requirements to estimate land-atmosphere coupling.

- P.5 line 15: is that version of the GRACE data downscaled in any way, and if so, how? I thought the basic GRACE data was at coarse resolution (e.g., 500km).

- P.5 line 17: "the TWS time series". I read that GRACE data are actually anomalies compared to the mean over 2004-2009. How is that accounted for in the computation of the metrics? Are the other variables centered on the same years? Does that affect results in any way? What about model outputs?

- P.6 line 32: see main comment above: I am not sure this is the most relevant time of year to investigate, and they are issues of rainfall autocorrelations.

- P.7 lines 12-15: that is, if the feedback is actually a positive moisture feedback. In other words, the authors adopt the a priori view that they are looking at a positive, moisture recycling feedback. This should be stated more explicitly, and perhaps earlier in the manuscript.

- P.8 line 18: what about AMIP simulations?

- P.8 line 28: It's unclear to me why the authors restrict themselves to the GLACE-CMIP5 models. There is no further comparison in the manuscript, on a model-by-model basis, with results from that experiment. So why not use the whole CMIP5 ensemble?

- P.9 line 9: so what? What is made of that? What are the implications for the correlation-based metrics? This comment applies to the whole sub-section, actually, including the result about climate variability. If anything, higher variability in model outputs would point to lower correlations, if the covariance between TWS and climate is similar.

- P.9 line 11: aren't trends removed from this data? Please clarify.

- P.9 line 26: still, why would the covariance be positive?

- P.9 whole section 3.2: this whole subsection feels very descriptive. On the other hand, there is not much description of the processes themselves. This might feel obvious to the authors, but some further discussion of what the correlations mean physically, when describing the figures, may be welcome.

- P.10 line 13: the link with cloud cover and precipitation should be explicitly mentioned here.

- P. 11 lines 14-15: see main comment #4 above.

- P.11 line 23: "Discussion".

- P.11 line 28: as mentioned above, these "well understood mechanisms" are actually never really explained.

- P.12 lines 3-4: that's exaggerated. Feedback results on Figures 5-7 are very noisy, and even from a simply qualitative perspective, it is a stretch to say that they agree with results from GLACE 1. One could just as well point out all the regions on Figures 5-7 that do NOT show up in GLACE 1 and say results are completely different. Besides, I find it a bizarre impulse (or maybe, a testament to the strength of the GLACE 1 study) that every land-atmosphere study seemingly feels the need to point out some level of agreement with GLACE results, even when, as is the case here, the match is very weak at best, and more importantly, when different data (observations versus models), processes and spatio-temporal scales are considered. Consider removing that comparison.

- P.12 line 16: see main comment #4 above.

- P.12 line 26: the authors could still look at this in models results, though. In fact, showing the link between TWS and ET, for instance, would reinforce their results and
the physical interpretation that they propose.

- P. 13, first paragraph: this is unclear. Do the author mean that that the models underestimate remote influences of SSTs, for instance, and thus appear to have too strong a coupling?

- P.13 lines 16-18: see main comment #3 above.

- P. 14 line 18: but here observations show positive coupling, too! Please clarify.

- P. 14 line 21: but reduced stomatal opening would be associated with reduced ET, too. Please clarify.

- P. 14 lines 18-30: See main comment #2. There is a fundamental issue with the manuscript here.

- P.15 line 3: see main comment #3.

- P.15 lines 8-9. not really: Seneviratne etal. (2013) show that long-term soil moisture change leads to more warming, differently across models in the GLACE-CMIP5. That, in and of itself, could be considered an estimate of (long-term) soil moisture-atmosphere coupling in these models; but, in any case, there is no comparison to estimates of present-day coupling.

- P.15 lines 11: No. Warmer air "holding" more water vapor and leading to more precipitation would lead to positive temperature-precipitation correlations – not negative.

- P.15 line 13: "determined": not really. What Berg et al. (2015) show is that because of land-atmosphere interactions, the interannual negative temperature-precipitation relationship that they identify in present-day climate holds on longer time scales, including in the case of climate change. This may be interpreted as suggesting, as the authors say here, that models with too strong a coupling will then overestimate future warming; however, it is not directly shown by that study. Consider rephrasing.

- P.16 line 10: see comment above on P.12.

- P.16 line 11: "regions of strong RESPONSE metrics", I believe.

- P.16 line 14: the implication is bit too implicit here. Consider being more explicit.

Figures

- Figure 1: nice figure that helps understand the study. The y-axis on a) refers to anomalies, I presume – see comment on GRACE values above.

- As noted above, Figure 3 and 4 are nice, but not much is made of them in the analysis.

- Figure 5-7: I suggest the authors modify the color legend here. More color shades is not always better. It is actually not easy to see differences in color shades on a continuous bi-color palette like here, and for the reader things essentially end up being two colors, one positive (green) and one negative (red). It would actually be easier to have fewer shades, more clearly separated, and with perhaps several different colors as well.

- Figure 8: I suggest showing the mean of the CESM distribution as well.

References cited in this review:

Findell, Kirsten L., P Gentine, B R Lintner, and B P Guillod, 2015: Data Length Requirements for Observational Estimates of Land-Atmosphere Coupling Strength. Journal of Hydrometeorology, 16(4).

Livezey R. E., and W. Y. Chen, 1983: Statistical field significance and its determination by Monte Carlo techniques. Mon. Wea. Rev., 111, 46–59.

Wei, J., R. E. Dickinson, and H. Chen, 2008: A negative soil mois- ture–precipitation relationship and its causes. J. Hydrometeor., 9, 1364–1376.

---

## Author Comment (AC1) · 10 Aug 2016

We thank Referee 1 for their supportive and thoughtful review. The reviewer presented a highly positive perspective on our analysis in their general comments, and provided several constructive criticisms in their specific comments. Below, we address the reviewer's specific comments by quoting each comment in *italicized* font, providing our response in roman font, and quoting our proposed revisions as indented roman font.

*Methods: Page 5, line 16: "with temporal gaps filled using linear interpolation." More detail is needed here. How many months in the data record are filled? What was the typical time interval of missing data; for example, were the "gaps" filled predominantly just 1 month or multiple consecutive missing months of data? Is a linear interpolation*

[Figure]

*reasonable for temporal gap-filling GRACE data?*

Missing months in the GRACE record are discussed at the following webpage: http://grace.jpl.nasa.gov/data/grace-months/. We limited the time series to September, 2002 through November, 2014, in order to reduce the number of temporal gaps. Within this time range, there are eight non-consecutive gaps of one single month and one gap of two missing months. Linear interpolation was chosen based on personal communication with colleagues experienced in the use of gridded land data from GRACE Tellus. We plan to clarify this by revising the text as follows:

> We obtained Level-3 TWSA data from GRACE using the University of Texas at Austin Center for Space Research (CSR) spherical harmonic solutions (Swenson, 2012). Global land data at a 1° resolution were scaled using the coefficients provided by Landerer and Swenson (2012). The study period was limited to September, 2002 through November, 2014, in order to minimize temporal gaps. GRACE data during the study period included eight non-consecutive and two consecutive missing months, which were smoothed using linear interpolation.

*Page 5, lines 20-22: "...the use of TWS data allows us to include surface storage, canopy storage... all of which may be sources of moisture that are potentially limiting factors for ET". This is just one specific instance of discussing TWS for land-atmosphere analysis, but in general I think the paper would be better served with a more complete discussion of the advantages and limitations of using TWS for this purpose. For example, is the groundwater component of TWS a limiting factor for ET if the rooting depth does not reach the water table? In many temperate agrarian or grassland landscapes this is the case. How do you think these issues could potentially affect the results, or more specifically do you think your findings would be different if you just used (e.g.,) surficial soil moisture or root-zone soil moisture?*

These points are well taken. The primary advantage of using TWS data from GRACE is the fact that it is the only multi-year global remote sensing product that includes root-zone soil moisture, unlike AMSR-E and others that only include near-surface soil moisture. The importance of including root-zone soil moisture is mentioned in the Introduction section, which we plan to revise, as detailed below, in order to clarify the use of TWS instead of surface soil moisture.

Regarding groundwater, if the rooting depth does not reach the water table, then groundwater would not serve as a limitation on ET. However, in agricultural and other inhabited areas, aquifer withdrawal for irrigation does provide a connection between groundwater and the atmosphere. Furthermore, the rate of aquifer recharge depends on how much available water is removed via ET. Finally, rooting depths are not always well understood, and the boundary between rooting zone and aquifer may not be known. Each of these points indicates an advantage to including the aquifer component of TWS. The disadvantage of including groundwater is that the metrics could be sensitive to long-term trends in groundwater, but these changes are likely to be relatively small on seasonal timescales and therefore do not provide a strong disadvantage.

We plan to remove segment in question, Page 5, lines 20–22, from the Methods section and to add the following text to the Introduction section in order to clarify and expand upon the importance of including the full TWS column for capturing root-zone soil moisture and other important components:

> Until recently, studies using remote sensing data to look for evidence of land–atmosphere coupling relied on products that provide information about surface soil moisture (Ferguson et al., 2012; Taylor et al., 2012). Consideration of root-zone soil moisture has been accomplished only indirectly via data-assimilated estimates (Guillod et al., 2015). The inability to directly consider root-zone soil moisture has been suggested as an explanation for the relatively weak coupling observed using remote sensing data (Hirschi

et al., 2014). In order to include root-zone soil moisture, as well as other sources of moisture available across entire seasons, the present study uses remote sensing data of the entire terrestrial water storage (TWS) column.

The metrics we introduce here were designed to utilize the monthly TWS anomaly (TWSA) anomaly product from the Gravity Recovery and Climate Experiment (GRACE) mission (Landerer and Swenson, 2012; Wahr et al., 2004). The GRACE TWSA product integrates soil moisture at all layers along with surface, canopy, snow/ice, and aquifer storage, as each of these components represents a potential source of moisture for fulfilling evaporative demand. For example, in areas where agricultural ecosystems are important, diversion of lake and river water resources and withdrawal from aquifers may contribute to ET. Furthermore, surface storage of liquid water and snow represent sources of water that are available for and potentially limiting to ET. Under these conditions, month-to-month TWS anomalies capture portions of the terrestrial water cycle that soil moisture alone may not.

*Results: Page 10, lines 23-26: you find consistently weaker forcing relationships in boreal regions and attribute this to "high levels of climate variability in many high latitude regions" because of AO, NAO, and the like. However, do you think the predominantly energy-limited evaporative regime of many boreal regions contributes to limiting the feedback connection between soil moisture and atmospheric conditions? I would not expect the forcing "limb" to match the strength of the "response" limb in such conditions where evaporative fraction is more a function of incoming radiation than soil moisture availability.*

This is a very good point, and we plan to add the following to address this:

Furthermore, at high latitudes, ET is generally energy limited rather than

moisture limited, which would lead to weak forcing metrics as moisture availability does not strongly influence atmospheric conditions.

*Page 11, lines 28-29: "Furthermore, the well-understood physical mechanisms allow causality to be inferred even when not directly demonstrated." The argument in the previous sentence is well-taken, but I disagree with the premise of this statement, particularly inferring causality in the forcing "limb" in the absence of discussion/analysis of possible confounding effects. At the very least I would like to see some assessment or acknowledgement of the role of atmospheric persistence as a confounding factor when quantifying the forcing limb. For example, is a strong forcing limb caused by the physical constraint of soil moisture on energy partitioning and modification of boundary layer dynamics and thermodynamics (as suggested by the authors), or is it the result of large-scale atmospheric circulation and persistence of synoptic-scale patterns that modify precipitation and atmospheric demand throughout the duration of the TWS draw down season? A correlation coefficient cannot adequately address the question of large-scale atmospheric persistence vs. soil moisture feedback, and indeed this is beyond the scope of this manuscript. However, this does also mean that one cannot infer causality and mechanistic connections between soil moisture and VPD/PPT/SW based on the evidence provided here.*

This point is well taken, and we plan to remove the sentence in question so as not to claim any indication of causality. We plan to expand the discussion just prior to the sentence in question to address the issue of persistence:

> The use of correlation coefficients in this study does not enable a direct assessment of whether the relationships are directly causal, as correlation between atmospheric and terrestrial conditions could result from atmospheric persistence and remote forcing from SST (Orlowsky and Seneviratne, 2010). Nonetheless, the satellite-derived metrics provide a meaningful constraint against which coupled models can be benchmarked, as these

models need to correctly represent the combined effects of persistence, remote SST forcing, and land–atmosphere coupling.

We also plan to address this by adding a paragraph to Section 4.5 Uncertainties and future applications:

> Finally, the issue of causality and the possibility that correlations result primarily from atmospheric persistence and remote forcing from SST rather than land–atmosphere interactions may be addressed using sensitivity experiments similar to those of GLACE and GLACE-CMIP. While the previous experiments have tested the importance of soil moisture interaction with the atmosphere, additional experiments could expand upon these methods by treating SST variability similar to terrestrial soil moisture availability. Such experiments could determine the relative importance of remote SST forcing, including the effect of atmospheric persistence, and local land–atmosphere coupling in explaining correlations between TWS and atmospheric conditions.

*Page 12, lines 14-16: "That models and ensemble members with high forcing metrics were also found to have high response metric... highlights the importance of the land surface response in priming the system for subsequent forcing on the atmosphere..." I don't understand how the land surface response "primes the system" for subsequent forcing when your analysis (and Figure 2) suggest the forcing occurs prior to the response. What am I missing? Can you please expound?*

We plan to clarify the discussion by replacing the lines in question with the following:

> The inclusion of the response metrics allows the full feedback loop to be considered by recognizing the two-way dependence between the land surface and the atmosphere. The generally higher correlation coefficients in

observed response metrics indicates the importance of the land surface response in priming the system for subsequent forcing on the atmosphere. For example, if TWS response too strongly coupled to atmospheric forcing, a small change in atmospheric conditions could yield an unrealistically large change in TWS. The unrealistically large TWS anomaly would have the potential to impart a larger land surface forcing of the atmosphere in subsequent time steps. That models and ensemble members with high forcing metrics were also generally found to have high response metrics (Figure 10) highlights the need to consider this.

*Conclusion: Page 16, lines 13-14: "...which suggests that some of these models may have difficulty properly predicting warming trends and climatic extremes." You include an excellent discussion of the potential links between model overestimation of land surface forcing and warming trends, founded in the body of literature. However, you do not explicitly quantify this linkage in this manuscript. The ability of models to "properly" predict warming trends and climatic extremes is not evaluated here, so this statement should probably be removed.*

This point is well taken. We plan to rephrase this section to qualify our conclusion and avoid making conclusions that were not evaluated in our manuscript:

Modeled feedback metrics are generally found to be stronger than those observed in the satellite record. If this discrepancy is due to models overestimating the two-way feedback between the land surface and the atmosphere, this could bias projections of future warming trends and climatic extremes.

**References cited in this response:**

Ferguson, C. R., Wood, E. F. and Vinukollu, R. K.: A global intercomparison of modeled and observed land–atmosphere coupling, J. Hydrometeorol., 13(3), 749–784, doi:10.1175/JHM-D-11-0119.1, 2012.

Guillod, B. P., Orlowsky, B., Miralles, D. G., Teuling, A. J. and Seneviratne, S. I.: Reconciling spatial and temporal soil moisture effects on afternoon rainfall, Nat Commun, 6, doi:10.1038/ncomms7443, 2015.

Hirschi, M., Seneviratne, S. I., Alexandrov, V., Boberg, F., Boroneant, C., Christensen, O. B., Formayer, H., Orlowsky, B. and Stepanek, P.: Observational evidence for soil-moisture impact on hot extremes in southeastern Europe, Nat. Geosci., 4(1), 17–21, doi:10.1038/ngeo1032, 2011.

Landerer, F. W. and Swenson, S. C.: Accuracy of scaled GRACE terrestrial water storage estimates, Water Resour. Res., 48(4), W04531, doi:10.1029/2011WR011453, 2012.

Orlowsky, B. and Seneviratne, S. I.: Statistical analyses of land–atmosphere feedbacks and their possible pitfalls, J. Clim., 23(14), 3918–3932, doi:10.1175/2010JCLI3366.1, 2010.

Swenson, S. C.: GRACE MONTHLY LAND WATER MASS GRIDS NETCDF RELEASE 5.0. Ver. 5.0, Datasets, doi:10.5067/TELND-NC005, 2012.

Taylor, C. M., de Jeu, R. A. M., Guichard, F., Harris, P. P. and Dorigo, W. A.: Afternoon rain more likely over drier soils, Nature, 489(7416), 423–426, doi:10.1038/nature11377, 2012.

Wahr, J., Swenson, S., Zlotnicki, V. and Velicogna, I.: Time-variable gravity from GRACE: First results, Geophys. Res. Lett., 31(11), doi:10.1029/2004GL019779, 2004.

---

## Author Comment (AC2) · 10 Aug 2016

We thank the Referee 2 for their supportive and thoughtful review. THe reviewer presented a highly positive perspective on our analysis in their general comments, and provided several constructive criticisms in their specific comments. Below, we address the reviewer's specific comments by quoting each comment in *italicized* font, providing our response in roman font, and quoting our proposed revisions as indented roman font.

*Page 4, line 16-17: "Until recently, studies using remote sensing data to look for evidence of land–atmosphere coupling relied on data products that provide information about surface soil moisture." There are some exceptions and some remote-sensing*

[Figure]

*products also estimate soil moisture in the whole root zone (for instance, I recall that Guillod et al. (2015) used such estimates).*

We re-read the Guillod et al. (2015) paper, and found that while they did account for root-zone soil moisture in their estimation of evaporative stress, this was done indirectly through a data-assimilated estimate. Their direct observations of soil moisture were of surface soil moisture in the top few cm of the soil column from AMSR-E. We plan to add the following explanation of this to the text in order to clarify the distinction between direct and indirect observations of root-zone soil moisture:

> Consideration of root-zone soil moisture has been accomplished only indirectly via data-assimilated estimates (Guillod et al., 2015).

*Section 3.3: These results on the role of internal variability are interesting. Another source of discrepancy between a single model run and observations can be observational error combined with short record (e.g., Findell et al., 2015), which may artificially decrease the coupling found in observations. Together, this effect and internal variability probably tell us something about the data length required to robustly assess land-atmosphere coupling.*

This point is well taken, and plan to expand upon this point with an addition to our discussion section in which we acknowledge the relevance of Findell et al., 2015:

> One important factor contributing toward stronger feedback metrics in models relative to observations is the effect of observational uncertainty combined with a relatively short time series. Adding error to one or more variables in a correlation analysis will reduce the correlation coefficient, and this degradation has been shown to be sensitive to the length of data sets used to establish metrics of land–atmosphere interactions (Findell et al., 2015). Given the relatively short time series available for the current analysis, the

correlation coefficients from remote sensing data may be reduced due to observational uncertainty, unlike those derived from internally-consistent models. We obtained a qualitative estimate of the influence of observational uncertainty on derived feedback metrics by replacing the atmospheric remote sensing data with reanalysis data from ERA-Interim. We found that both sets of observationally based metrics were weaker than those from LENS and several other models, suggesting that some of the overestimated feedback metrics in models may not be fully explained by observational uncertainty.

This addition serves to strengthen the conclusion that the utility of our approach will increase as the satellite data record grows longer, and reinforces the importance of the GRACE follow-on mission in lengthening the time series of TWS anomalies. We plan to further modify our discussion section to emphasize this point as follows:

Furthermore, we acknowledge that observational error over an insufficiently long time series could reduce the apparent strength of correlations (Findell et al., 2015). Therefore, the utility of the feedback metrics will increase alongside the length of the time series available from remote sensing platforms. This emphasizes the importance of the GRACE follow-on mission (Flechtner et al., 2014) and the need for continuity in the record between missions.

*Page 8, line 24 and Page 11, line 5: Supplementary figures references could be more specific. I think that the first mention on page 8 refers to Fig. S3 while the second mention (page 11) refers to Figs. S1 and S2. The order of the supplementary figures could also be adapted so they are cited in sequence.*

We agree that the supplementary figures should be referenced more specifically, and plan to adapt the relevant sections on both Page 8 and Page 11 to address this. The

first mention in page 8 is intended to refer to all three of the supplementary figures, so we plan to add additional text to refer specifically to what each of the supplementary figures show:

> To extend our analysis to models that do not output an explicit TWS field, the accumulated residuals of precipitation, evapotranspiration, and total runoff (surface and subsurface) were compared with TWS in LENS (Figure S1). We also compared feedback metrics calculated from LENS using accumulated residuals with those calculated from the explicit TWS field (Figures S2 and S3).

This revision includes a change to the order of figures, so that the Taylor diagrams will become S1 (previously S3) and the histograms will become S2 and S3 (previously S1 and S2).

*Page 13, line 20-29: The authors write that overestimating ET would lead to excessive land-atmosphere coupling. This is a bit confusing: if the coupling is strongest in transitional regions between wet and dry climate, why would a too high ET in an already wet region lead to an increase in coupling? I agree that model errors with respect to the distribution of water between storage reservoirs can be an issue; however I do not think that this necessarily leads to stronger coupling, but maybe I am missing something here.*

We thank the reviewer for pointing out the need to clarify the basis of this argument. We plan to replace the paragraph in question with the following:

> A set of possible explanations involves models overestimating the amount of water available for ET during the drawdown interval. The land surface influence on the atmosphere requires water to be a limiting factor to ET but not limiting enough to prevent it altogether. Under more moisture-limited

conditions, a drawdown interval may experience multiple shorter time periods during which ET is inhibited due to insufficient water, and the terrestrial moisture state exerts no control over flux partitioning. These periods of insufficient moisture would tend to reduce the overall feedback strength integrated across the duration of the drawdown interval. Model shortcomings that make water too readily available for ET could reduce the amount of time spent in a periods of insufficient moisture during the drawdown interval, thereby unrealistically strengthening the longer-term feedback. We note that the opposite could take place under near-saturated conditions if a model overestimates the amount of time in which ET is energy-limited, but we would not expect these conditions to be as prevalent during the drawdown interval that was the time period of focus in our analysis.

**References cited in this response:**

Findell, K. L., Gentine, P., Lintner, B. R. and Guillod, B. P.: Data length requirements for observational estimates of land–atmosphere coupling strength, J. Hydrometeorol., 16(4), 1615–1635, doi: 10.1175/JHM-D-14-0131.1, 2015.

Flechtner, F., Morton, P., Watkins, M. and Webb, F.: Status of the GRACE Follow-On Mission, in Gravity, Geoid and Height Systems: Proceedings of the IAG Symposium, edited by U. Marti, pp. 117–121, Springer International Publishing, 2014.

Guillod, B. P., Orlowsky, B., Miralles, D. G., Teuling, A. J. and Seneviratne, S. I.: Reconciling spatial and temporal soil moisture effects on afternoon rainfall, Nat Commun, 6, doi:10.1038/ncomms7443, 2015.

---

## Author Comment (AC3) · 10 Aug 2016

We thank the Referee 3 for their thorough and thoughtful review. The reviewer was very critical of several aspects of our paper, and identified issues that they believe warrant several important revisions. We agree with many of the points the reviewer makes, and we plan to make significant revisions to address these points, which we detail below in our response to the reviewer's comments. However, we disagree with the reviewer's perspective that these represent "fundamental issues" with the manuscript and the research presented in it. We believe that the research represents a meaningful and novel contribution to the field, as indicated by the other two referees who reviewed our manuscript. We believe that the major revisions that we plan to make, detailed

below, will clarify our position and improve our acknowledgement of certain limitations and caveats. We hope these clarifications and revisions will satisfy the reviewer, but we would be happy to engage in an ongoing dialogue with this reviewer as well as the community at large if there are any unresolved issues.

Below, we address the reviewer's general and specific comments by quoting each comment in *italicized* font, providing our response in roman font, and quoting our proposed revisions as indented roman font.

**Main comments:**

*1) My first main comment has to do with the metric used. First, the brevity of the time period of analysis is obviously an issue. The metrics are essentially interannual correlations over 13 years, i.e., correlations over 13 data points. That's very short. I am not sure I have seen many studies looking at interannual variability over 13 years only, in particular in terms of land-atmosphere studies. Accordingly, results for the feedback metrics appear very noisy. First, I believe a discussion of field significance is warranted here: patches of apparently significant values may still be random in that context (e.g., see Livezey and Chen (1983)).*

*Second, note that recent research underscores the need for long-record datasets to establish land-atmosphere coupling, that coupling metrics require more data than single-variable simple statistics (e.g., mean and variance) to be robustly estimated, and finally that, unlike single-variable statistics, coupling metrics are actually degraded by observational uncertainty (Findell et al. 2015). The latter point, in particular, is in my view a much likelier explanation for the weaker correlations found here in observations – between uncertain observation datasets that are independent of each other – compared to correlations computed with model outputs, which are by definition perfectly consistent with each other. The authors touch on the issue of observational uncertainty by computing correlations with the ERA reanalysis, but I don't think enough is made of that.*

*So, given the brevity and uncertainty of observations, even without consideration of any other issues (but, see below), I am really uncomfortable with the approach proposed by this paper, which is to consider the observational estimate as a benchmark for model evaluation. Personally, I think an approach where observations and model results are used together to try to infer the "real" coupling would make more sense here. But, this would lead to a very different paper. Overall, if the authors are going to go on with their approach, I would recommend much more caution in how things are presented, including in the title of the study and the conclusions.*

The reviewer makes several excellent points, and we are appreciative of the thoughtful perspective. We acknowledge that several of these points warrant revisions and additions to the text (detailed below) in order to clarify the goals of our approach and more appropriately emphasize its limitations. However, we believe that despite these limitations, our approach still represents a valuable and novel system for evaluating model performance using observational constraints.

We recognize that the relatively short time series available from the satellite record warrants caution while interpreting these results. We agree with the reviewer that the findings of Findell et al. (2015) emphasize this limitation and suggest that observational uncertainty would be expected to decrease correlations. As such, we plan to discuss this in Section 4.3 with the following addition:

> One important factor contributing toward stronger feedback metrics in models relative to observations is the effect of observational uncertainty combined with a relatively short time series. Adding error to one or more variables in a correlation analysis will reduce the correlation coefficient, and this degradation has been shown to be sensitive to the length of data sets used to establish metrics of land–atmosphere interactions (Findell et al., 2015). Given the relatively short time series available for the current analysis, the correlation coefficients from remote sensing data may be reduced due to

observational uncertainty, unlike those derived from internally-consistent models. We obtained a qualitative estimate of the influence of observational uncertainty on derived feedback metrics by replacing the atmospheric remote sensing data with reanalysis data from ERA-Interim. We found that both sets of observationally based metrics were weaker than those from LENS and several other models, suggesting that some of the overestimated feedback metrics in models may not be fully explained by observational uncertainty.

This acknowledgement further supports our argument that the utility of these metrics will increase as the time series of global satellite data grows longer with continuations of current missions and initiation of new missions (i.e., GRACE follow on) as mentioned in Section 4.5:

Furthermore, we acknowledge that observational error over an insufficiently long time series could reduce the apparent strength of correlations (Findell et al., 2015). Therefore, the utility of the feedback metrics will increase alongside the length of the time series available from remote sensing platforms. This emphasizes the importance of the GRACE follow-on mission (Flechtner et al., 2014) and the need for continuity in the record between missions.

We do not believe the issue of field significance is relevant in the current context. Our metrics compare a single time series of TWS anomalies with a single time series of atmospheric data in the same region. We are not calculating correlations between a single explanatory variable and a geographically distributed field of dependent variables. Therefore, we are not engaging in the type of hypothesis testing that would warrant consideration of field significance.

We agree with the reviewer that "an approach where observations and model results are used together to try to infer the "real" coupling" would be valuable, and represents a research priority for the community. Our intention was not to present the observationally derived forcing metric as representing the "real" land–atmosphere coupling strength. Instead, it represents the combined effects of land–atmosphere coupling (the "real" coupling strength) along with the remote effects of SST forcing on both the atmosphere and land surface. We believe that despite the relatively short time series, these metrics provide a useful constraint on models' ability to represent this combined set of processes. The reviewer's recommendation of greater caution in our presentation is appreciated, and we plan to add the following clarification to section 2.2 when introducing the metrics:

> We note that while these metrics provide information about land–atmosphere coupling as a forcing mechanism on the atmosphere and the response of the land surface to the atmosphere, they are potentially influenced by atmospheric and soil moisture persistence, as well as remote forcing from sea surface temperatures (SST) (Orlowsky and Seneviratne, 2010; Mei and Wang, 2011). Nevertheless, these metrics still provide useful benchmarks against which to evaluate the ability to ESMs to reproduce the proper relationships based on the combination of these factors.

We plan to further clarify this in a major revision to the first paragraph of Section 4.1

> The metrics developed here from satellite observations provide a means for evaluating land–atmosphere feedback strength on seasonal to interannual timescales in coupled ESMs. The use of correlation coefficients in this study does not enable a direct assessment of whether the relationships are directly causal, as correlation between atmospheric and terrestrial conditions could result from atmospheric persistence and remote forcing from

SST (Orlowsky and Seneviratne, 2010; Mei and Wang, 2011). Nonetheless, the satellite-derived metrics provide a meaningful constraint against which coupled models can be benchmarked, as these models need to correctly represent the combined effects of persistence, remote SST forcing, and land–atmosphere coupling.

We also plan to emphasize the importance of disentangling the influence of land–atmosphere coupling from that of atmospheric persistence and remote SST forcing with the following addition to Section 4.5:

> Finally, the issue of causality and the possibility that correlations result primarily from atmospheric persistence and remote forcing from SST rather than land–atmosphere interactions may be addressed using sensitivity experiments similar to those of GLACE and GLACE-CMIP. While the previous experiments have tested the importance of soil moisture interaction with the atmosphere, additional experiments could expand upon these methods by treating SST variability similar to terrestrial soil moisture availability. Such experiments could determine the relative importance of remote SST, including the effect of atmospheric persistence, and local land–atmosphere coupling in explaining correlations between TWS and atmospheric conditions.

We believe that despite the limitations of a relatively short time series and the inability to attribute the sources of covariability, our approach is still valuable. We believe that the revisions described above emphasize our goal of conceptually illustrating an approach towards model benchmarking that will become increasingly useful with longer time series from remote sensing. At this point, we would prefer to retain our title, which we believe is succinct and accurately conveys the content of our paper.

*2) Another issue with the metrics involves the definition of the (feedback) metric. The way it is defined, it is looking at the impact of TWS at the end or peak of the rainy season on climate in the subsequent months. The authors indicate as much, and say they want to consider, in the Tropics, the impact of late-rainy season TWS on dry-season climate. I see two issues with that. First, in my view, while that may useful in the deep Tropics where the dry season is short, this approach is problematic in monsoon regions, or regions of the Tropics that have a well defined rainy season (i.e., outside of the deep Tropics): basically, after the rainy season, there is not much rain to look at any more. For instance, over the Sahel, what the authors are computing is the impact of September TWS on precipitation over September-May. But it doesn't rain much over that time period.*

We believe that focusing on the drawdown interval is an important part of our approach. Our algorithm is novel in allowing a global-scale analysis across ecosystems. In mid-latitudes, the drawdown interval contains the summer season that has been the focus of research in land–atmosphere coupling. In tropical latitudes, the drawdown interval contains the dry season, during which precipitation recycling is important for maintaining ecosystems, allowing forests to persist in the absence of circulation-driven precipitation [Keys et al., 2016]. In the example of the Sahel, our algorithm is working as intended, by measuring how variations in TWS at the onset of the dry season are related to atmospheric conditions during the dry season.

*In my view, it would be much more interesting to look at the impact of end-of-dry season TWS on the subsequent rainy season to see if, in these regions, available land moisture feeds back on precipitation during the rainy season. Second, in the same example over West Africa, whatever rainfall there is over Sept-May is actually probably the end of the monsoon, Sept-Nov. Because TWS in September is likely to be influenced by precip in September, and Sept. precip is likely to represent large part of the 'response' variable, the causation is muddied a little bit: a clearer temporal offset would be needed in such a case. But more importantly, even precip in the months following September (Oct,*

*Nov) is likely to be correlated with precip in the previous months – for instance, a year with a strong monsoon that has more rain in Jun-Sept may well tend to also have more rain in Sept-Nov. Because September TWS will largely reflect JJAS rainfall, the TWS-based metric will then show a strong feedback - but the inferred causation would be a misinterpretation.*

We believe that ET in the wet season tropics would be energy limited, and therefore would not reflect the influence of land surface moisture availability on the atmosphere. We acknowledge the issue with persistence, which we expand upon below.

*This brings me to a more general point: the authors do not discuss how autocorrelation, here at the seasonal time scale, of climate variables, may impact their estimate of land-atmosphere coupling. This is a major issue affecting all empirical studies of land-atmosphere coupling – see, for instance, Wei et al. (2008) and Orlowsky and Seneviratne (2010). The authors do cite the latter study, but, it seems, simply to say that if models underestimate SSTs influence on land climate, they will then appear to overestimate local L-A coupling. They somehow miss the point of that paper in how it should apply to to their own results. I just gave one example above (the Sahel) of how that might be the case. Other monsoon regions (e.g., India) might similarly be affected. Interannual variability in the coupled ocean-atmosphere (eg., ENSO) might also the source of confounding influence at the time scales investigated here. So, overall, I recommend these caveats be considered and discussed by the authors in their interpretation of their results. Personally, I would need to see some further analysis to be more convinced of the physical reality of the land-atmosphere feedbacks the authors claim to show (e.g,. some sensitivity test to the months and time lags considered, some investigation of atmospheric variability and persistence, etc.).*

The reviewer's point is well taken, and has already been partially addressed above in the additions to Section 2.2 and Section 4.5 (quoted above). In addition, we plan to discuss these issues more explicitly in a revision of Section 4.3 to include the following:

Another possible explanation stems from the fact that our feedback metrics include the influence of both direct interactions between the land-surface and the atmosphere as well as indirect covariability due to atmospheric persistence and remote forcing by SST (Orlowsky and Seneviratne, 2010; Mei and Wang, 2011). For this reason, we caution that overestimates of feedback metrics do not imply that the land–atmosphere feedback is necessarily stronger, but could be due to an overestimate of SST-driven correlations between the land surface and the atmosphere. Wei et al. (2008) demonstrated that negative correlations between soil moisture and subsequent precipitation can be explained by precipitation persistence combined with negative temporal autocorrelation of precipitation associated with subseasonal modes such as the Madden-Julian Oscillation (MJO). Poor representation of the MJO period in CMIP5 models leads to unrealistic patterns of precipitation persistence (Hung et al, 2013). If models are failing to capture MJO-driven negative correlations, this could lead to overly strong positive correlations relative to observations. However, this would depend on the length of the drawdown season relative to persistence time and the period of intra-seasonal modes.

This supports our planned addition to Section 4.5 (quoted above), discussing the importance of modeling experiments to determine relative importance of remote SST forcing, including the effect of atmospheric persistence, and local land–atmosphere coupling in explaining correlations between TWS and atmospheric conditions.

*3) Another main comment has to do with the discussion section. The authors discuss why models might exhibit stronger feedback (and forcing) metrics than observations. As mentioned above, I think uncertainty in observations should be mentioned as a primary reason.*

We now more explicitly address observational uncertainty as well as uncertainty due

to the short time series in Section 4.5 quoted above.

*The authors propose that ET may be consistently overestimated in climate models, and a large part of the discussion then consists in speculation as to why that may be the case. First, while I appreciate the effort to discuss things further and not just show results, I found this whole section a bit too speculative. IF the models overestimate ET, then IF stomatal conductance, IF convection, IF bare soil, etc. . . Can the authors actually point to any evidence that ET is consistently overestimated in climate models, in the first place (or at least in CESM)?*

These points are well taken, and while this section is speculative by its very nature, we agree that it warrants revision. We plan to modify the discussion so that it does not center on models overestimating ET, but instead focuses on ways in which models could make moisture too readily available for ET. We plan to clarify the basis of our argument with the following revision to Section 4.3:

> A set of possible explanations involves models overestimating the amount of water available for ET during the drawdown interval. The land surface influence on the atmosphere requires water to be a limiting factor to ET but not limiting enough to prevent it altogether. Under more moisture-limited conditions, a drawdown interval may experience multiple shorter time periods during which ET is inhibited due to insufficient water, and the terrestrial moisture state exerts no control over flux partitioning. These periods of insufficient moisture would tend to reduce the overall feedback strength integrated across the duration of the drawdown interval. Model shortcomings that make water too readily available for ET could reduce the amount of time spent in a periods of insufficient moisture during the drawdown interval, thereby unrealistically strengthening the longer-term feedback. We note that the opposite could take place under near-saturated conditions if a model overestimates the amount of time in which ET is energy-limited, but

we would not expect these conditions to be as prevalent during the draw-down interval that was the time period of focus in our analysis.

We also plan to add further discussion to Section 4.5 citing evidence of models over-estimating ET:

> CMIP5 models are known to have a high ET bias (Mueller and Seneviratne, 2014), which could be due in part to the explanations proposed as possible reasons for overestimated feedback metrics in models.

*Second, if soil water is too readily available in models, and ET is overestimated, wouldn't that actually mean that feedbacks should be underestimated in models? In-deed, ET would then be less water-limited and more energy-limited, with less potential for soil moisture-atmosphere feedbacks.*

We designed our metrics around the drawdown interval in order to specifically consider the time of year during which ET would be water-limited. The issues we discuss with insufficient representation of bare soil processes and big leaf parameterizations would, during this time of year, unrealistically make too much water available for ET. This would allow ET to take place in the model when in reality that water would have run off or is unavailable for transpiration. Under these conditions, the atmosphere in the model would be influenced by moisture availability when in reality no moisture is available. These points will be clarified with the revision to Section 4.3 quoted above.

*Surface climate variability would then be influenced by the atmosphere to a greater extent. Along the same lines, the authors claim that their results, showing an overesti-mation of land-atmosphere feedback by models, are consistent with prior studies, and have implications for projected warming (e.g., Cheruy et al., 2014). However, these previous studies, it seems to me, point to ET being underestimated in these models,*

*and models getting to easily "locked" in a dry and warm surface mode. So, in effect, while the authors agree with prior studies that land-atmosphere feedbacks are overestimated in models, they provide opposite reasons for that (overestimated versus underestimated of ET). I would like to see the authors clarify that apparent contradiction.*

We plan to modify our discussion, described above, that clarifies our point so as not to rely on whether models overestimate or underestimate ET.

*4) Finally, the author interpret the relationship they find between the strength of the feedback and forcing metrics in CMIP5 models as showing that: "the response limb of the feedback loop is important for understanding how conditions are set up for subsequent forcing via land–atmosphere coupling". They claim that it highlights "the importance of the land surface response in priming the system for subsequent forcing on the atmosphere". I am not convinced by this interpretation, which sounds a bit hand-wavy to me. I don't see a strong physical reason why a model where, for instance, TWS responds strongly to precipitation, should have a strong feedback of TWS onto precipitation.*

Conceptually, we disagree with this perspective. The strength of land–atmosphere coupling depends on moisture availability enabling some ET while still limiting it. Models must therefore simulate the correct moisture availability in order to simulate the proper amount of land–atmosphere coupling. The response metrics reflect whether models simulate the right moisture availability based on precipitation and evaporative demand, and whether this is the right amount to set up subsequent land–atmosphere coupling. We plan to clarify our reasoning with the following modification to Section 4.1:

> The inclusion of the response metrics allows the full feedback loop to be considered by recognizing the two-way dependence between the land surface and the atmosphere. The generally higher correlation coefficients in observed response metrics indicates the importance of the land surface response in priming the system for subsequent forcing on the atmosphere. For example, if TWS response too strongly coupled to atmospheric forcing, a small change in atmospheric conditions could yield an unrealistically large change in TWS. The unrealistically large TWS anomaly would have the potential to impart a larger land surface forcing of the atmosphere in subsequent time steps. That models and ensemble members with high forcing metrics were also generally found to have high response metrics (Figure 10) highlights the need to consider this.

*Couldn't the relationship on Figure 10 be due to intermodel differences in what TWS (or its estimate, here) encompasses in each model? For instance, different soil depths? A deeper soil would lead to weaker links between TWS and climate both in terms of response and feedback to the atmosphere.*

No, because we are using the total terrestrial water storage column. In the case of LENS, this is an explicitly output field that includes this entire column. In the case of the CMIP5 output, we used the accumulated residuals of the surface water balance (i.e., the integral of precipitation minus evaporation and runoff) to approximate this quantity.

*In any case, I found Figure 10 to be insufficiently explained and encourage the authors to discuss this further.*

We plan to add the following to the figure caption in order to clarify how TWS was determined for each model.

> For CMIP5 models, we estimated TWSA using the accumulated residuals of the surface water balance. For LENS, TWSA values were internally calculated from water masses in soils and other terrestrial reservoirs

**Specific comments:**

*- p.2 line 4: "cloud radiative coupling": please explain and clarify.*

We plan to clarify the text as follows:

> Cloud radiative coupling can likewise lead to positive or negative feedbacks,
> as enhanced (suppressed) cloud formation decreases (increases) insola-
> tion and evaporative demand (Betts, 2009; Cheruy et al., 2014).

*- P.2 line 24: actually, no: a surprising result of GLACE II was that predictive skill was
not enhanced over the Great Plains "hot spot" from GLACE I, but rather to the North of
it (see Koster et al. 2011). Consider rephrasing.*

We thank the reviewer for pointing out this discrepancy, which plan to correct by re-
moving the reference to GLACE II

*- P.3 line 5: the text should make it clear that GLACE-like metrics cannot be directly
compared to observations, and that other more simple metrics, not strictly equivalent,
have to be used, like SM-ET correlations, etc.*

We agree that this warrants clarification, and we plan to modify the text as follows:

> GLACE metrics are based on model experiments with no direct observa-
> tional equivalents. However, correlation based metrics that do enable direct
> comparison with observations suggest that models may overestimate land–
> atmosphere coupling strength (Dirmeyer et al., 2006).

*- P.3 line 20: Findell et al. 2011 is actually based on reanalysis data, not "modeling".
Also, Findell et al. 2015 should be included in this discussion, to highlight the issue,
discussed above, of data length requirements to estimate land-atmosphere coupling.*

We reference Findell et al. (2011) in the context of Guillod et al. (2014), which empha-
sizes that the surface state and fluxes are still model based, even if the atmosphere

is constrained by some observations. However, to avoid confusion, we now omit the word "modeling" from the description of Findell et al. (2011). Furthermore, as discussed above, we now include Findell et al. (2015) in the discussion in Section 4.3 (quoted above).

*- P.5 line 15: is that version of the GRACE data downscaled in any way, and if so, how? I thought the basic GRACE data was at coarse resolution (e.g., 500km).*

The GRACE gridded land product that we use is provided at a 1-degree resolution. We plan to clarify this in the methods section by rewording the beginning of Section 2.1 Remote sensing data as follows:

> We obtained Level-3 TWSA data from GRACE using the University of Texas at Austin Center for Space Research (CSR) spherical harmonic solutions (Swenson, 2012). Global land data at a 1° resolution were scaled using the coefficients provided by Landerer and Swenson (2012).

*- P.5 line 17: "the TWS time series". I read that GRACE data are actually anomalies compared to the mean over 2004-2009. How is that accounted for in the computation of the metrics? Are the other variables centered on the same years? Does that affect results in any way? What about model outputs?*

In the context of our metrics, the baseline against which GRACE Anomalies are compared is arbitrary. In our calculations, the baseline only affects the intercepts of the linear correlations; it does not affect the correlation coefficients that comprise our metrics.

*- P.6 line 32: see main comment above: I am not sure this is the most relevant time of year to investigate, and they are issues of rainfall autocorrelations.*

We have addressed this in our response to the reviewer's main comment above, in which we explain why we chose to focus on this time of year.

*- P.7 lines 12-15: that is, if the feedback is actually a positive moisture feedback. In other words, the authors adopt the a priori view that they are looking at a positive, moisture recycling feedback. This should be stated more explicitly, and perhaps earlier in the manuscript.*

This point is well taken, and we plan to modify the Methods section to more explicitly state this assumption by removing lines 12–15 and 19–20 on page 7, and replacing them with the following:

> Here we note that our evaluation of both the forcing and response metrics will follow a nomenclature that considers strong coupling as acting in the direction of an overall positive feedback loop. In regions with a strong positive feedback, higher than average TWS would be followed by lower than average VPD, as more available water is able to fulfill evaporative demand. Therefore, strong TWS forcing on VPD would be associated with a negative correlation coefficient. Higher VPD during the drawdown interval would increase evaporative demand, potentially leading to a lower TWS anomaly, therefore a strong response of the land surface to VPD would also be associated with a negative correlation coefficient.

> Because the partitioning of surface fluxes can, depending on the spatiotemporal scale, cause a change of either sign to both cloudiness and precipitation (Taylor et al., 2012; Guillod et al., 2015), correlation coefficients of either sign could indicate strong land surface forcing on PPT and SW↓. However, the response metrics would be expected to show greater consistency. Higher PPT during the drawdown interval would be expected to increase TWS (positive correlation), while higher SW↓ would increase evaporative demand, thereby decreasing TWS (negative correlation). Therefore, to maintain consistent nomenclature based on evaluating the strength of a positive moisture feedback, we consider strong coupling in both the forcing and response metrics to be associated with a positive correlation in the case of PPT and a negative correlation in the case of SW↓.

*- P.8 line 18: what about AMIP simulations?*

Correlations in our forcing metric come from both land–atmosphere coupling and the effects of remote SST forcing. AMIP simulations could reduce internal variability, but will not capture ocean-atmosphere interactions. We are interested in evaluating fully coupled models that are used for 21st century projections.

*- P.8 line 28: It's unclear to me why the authors restrict themselves to the GLACE-CMIP5 models. There is no further comparison in the manuscript, on a model-by-model basis, with results from that experiment. So why not use the whole CMIP5 ensemble?*

The purpose of this manuscript is not to evaluate the entire CMIP5 ensemble, but rather to introduce a new approach toward model benchmarking using a small ensemble of models as an example.

*- P.9 line 9: so what? What is made of that? What are the implications for the correlation-based metrics? This comment applies to the whole sub-section, actually, including the result about climate variability. If anything, higher variability in model outputs would point to lower correlations, if the covariance between TWS and climate is similar.*

This point is well taken, and we address this with an addition to this section that clarifies the implications of this analysis:

> Comparing both the timing of TWS dynamics and the interannual variability of TWS and the atmospheric variables between the observations and model output provides context for interpreting the correlation-based metrics we present next. Although there are some inconsistencies, as noted

above, the model largely reproduces the same patterns evident in the re-
mote sensing data. In most regions, interannual variability in model output
is within an order of magnitude of the observed variability, indicating that
CESM can reasonably simulate the baseline properties (timing and vari-
ability) that influence the feedback dynamics.

*- P.9 line 11: aren't trends removed from this data? Please clarify.*

Trends are removed only for the purpose of generating the annual climatology, as indi-
cated in the methods section, are retained in the correlation analysis in order to capture
decadal-scale variability that would represent a trend in the relatively short time series.

*- P.9 line 26: still, why would the covariance be positive?*

We plan to address this question by adding the following text to this section:

> Positive correlations are unlikely to reflect direct land–atmosphere interac-
> tions. Instead, they demonstrate how remote SST forcing can, depending
> on lag times, lead to apparent negative relationships such as those demon-
> strated by Wei et al. (2008).

*- P.9 whole section 3.2: this whole subsection feels very descriptive. On the other hand,
there is not much description of the processes themselves. This might feel obvious to
the authors, but some further discussion of what the correlations mean physically, when
describing the figures, may be welcome.*

The purpose of this section is to help the reader understand how the metrics work
for a single ensemble member before we present the aggregated results for the whole
ensemble. We feel this is a critical step in explaining how our metrics work and justifying
their use in comparing models with observations.

*- P.10 line 13: the link with cloud cover and precipitation should be explicitly mentioned here.*

The link is now explicitly mentioned:

> This is consistent with coupling between cloud cover and terrestrial moisture being both positive and negative on shorter time scales, which sometimes yields negative coupling over shorter time scales (Taylor et al., 2012; Guillod et al., 2015).

*- P. 11 lines 14-15: see main comment 4 above.*

We have addressed this in the response to the reviewer's comment 4 above.

*- P.11 line 23: "Discussion".*

Thank you for pointing out this typographical error, which has been corrected.

*- P.11 line 28: as mentioned above, these "well understood mechanisms" are actually never really explained.*

The planned revision to Section 4.1, described above in the response to the first general comment, removes this phrase and more clearly describes what is being shown.

*- P.12 lines 3-4: that's exaggerated. Feedback results on Figures 5-7 are very noisy, and even from a simply qualitative perspective, it is a stretch to say that they agree with results from GLACE 1. One could just as well point out all the regions on Figures 5-7 that do NOT show up in GLACE 1 and say results are completely different. Besides, I find it a bizarre impulse (or maybe, a testament to the strength of the GLACE 1 study) that every land-atmosphere study seemingly feels the need to point out some level of agreement with GLACE results, even when, as is the case here, the match is very weak at best, and more importantly, when different data (observations versus models), processes and spatio-temporal scales are considered. Consider removing*

*that comparison.*

The reviewer's point is well taken, and we do not believe the comparison with GLACE 1 is a necessary component of our discussion. As part of the major revision of the discussion section, as described above, we plan to remove this reference to GLACE.

*- P.12 line 16: see main comment 4 above.*

We have addressed this in the response to main comment 4 above.

*- P.12 line 26: the authors could still look at this in models results, though. In fact, showing the link between TWS and ET, for instance, would reinforce their results and the physical interpretation that they propose.*

This is limited by the lack of a global remote sensing ET data set spanning the study period.

*- P. 13, first paragraph: this is unclear. Do the author mean that that the models underestimate remote influences of SSTs, for instance, and thus appear to have too strong a coupling?*

Along side the other major revisions to our discussion section, we plan to remove this passage from the text and replace it with a clearer explanation, which we have quoted above in response to main comment 2.

*- P.13 lines 16-18: see main comment 3 above.*

This is addressed in major revisions to this section, quoted above in the response to main comment 3.

*- P. 14 line 18: but here observations show positive coupling, too! Please clarify.*

The observational metrics in this study include the effects of remote SST forcing as well as land–atmosphere interactions integrated across seasonal time scales. The negative soil moisture–precipitation coupling mechanism found from observations by Taylor

et al. (2013) would tend to reduce the overall positive correlation. If parameterized convection prevents models from correctly capturing this mechanism, then the correlations may be overestimated. We plan to clarify this with the following addition to the discussion:

> If negative coupling mechanisms are present in reality but absent from models, this could contribute to an overestimate of feedback metrics and under-representation of negative feedbacks in models.

*- P. 14 line 21: but reduced stomatal opening would be associated with reduced ET, too. Please clarify.*

We plan to remove the discussion of stomatal opening, both for the sake of brevity and to prevent any confusion.

*- P. 14 lines 18-30: See main comment 2. There is a fundamental issue with the manuscript here.*

We plan to make major revisions to this section, as addressed above in the response to main comments 1 and 2. As we explained above, we do not feel that this represents a fundamental issue with our approach or our manuscript.

*- P.15 line 3: see main comment 3.*

We have addressed this above in our response to comment 3.

*- P.15 lines 8-9. not really: Seneviratne etal. (2013) show that long-term soil moisture change leads to more warming, differently across models in the GLACE-CMIP5. That, in and of itself, could be considered an estimate of (long-term) soil moisture-atmosphere coupling in these models; but, in any case, there is no comparison to estimates of present-day coupling.*

We agree that the linkage between our metrics and the results of GLACE-CMIP5 are

too speculative. For this reason and for the sake of brevity, we plan to remove this portion of the discussion section.

*- P.15 lines 11: No. Warmer air "holding" more water vapor and leading to more precipitation would lead to positive temperature-precipitation correlations – not negative.*

We thank the reviewer for pointing out this erroneous characterization of the results of Berg et al. (2015). The phrase "in which higher air temperatures can hold more precipitable water" was intended to read "cloud cover variability drives precipitation and temperature in opposite directions" However, as mentioned in the response to the previous comment, we intend to remove this portion of the text from our discussion.

*- P.15 line 13: "determined": not really. What Berg et al. (2015) show is that because of land-atmosphere interactions, the interannual negative temperature-precipitation relationship that they identify in present-day climate holds on longer time scales, including in the case of climate change. This may be interpreted as suggesting, as the authors say here, that models with too strong a coupling will then overestimate future warming; however, it is not directly shown by that study. Consider rephrasing.*

As mentioned in the response to the previous two comments, we plan to remove this portion of the text from our discussion.

*- P.16 line 10: see comment above on P.12*

As mentioned in the response to the comment above, we plan to remove the assertion that our observed metrics reflect the patterns found in GLACE. In addition to the revisions to the discussion section, mentioned above, we also plan to revise the conclusion section to remove this portion.

*- P.16 line 11: "regions of strong RESPONSE metrics", I believe.*

We thank the reviewer for pointing out this typographical error

*- P.16 line 14: the implication is bit too implicit here. Consider being more explicit.*

We plan to make this statement more explicit with the following revision:

> Modeled feedback metrics are generally found to be stronger than those observed in the satellite record. If this discrepancy is due to models over-estimating the two-way feedback between the land surface and the atmosphere, this could lead to models incorrectly projecting future warming trends and climatic extremes.

Figures:

*- Figure 1: nice figure that helps understand the study. The y-axis on a) refers to anomalies, I presume – see comment on GRACE values above.*

Correct, as we mentioned in response to the comment above. We plan to clarify this by replacing "TWS" with "TWSA" in the caption to Figure 1.

*- As noted above, Figure 3 and 4 are nice, but not much is made of them in the analysis.*

As described in response to the reviewer's comment above, we plan to add some additional text to expand upon and clarify the purpose of these figures. As quoted above, we feel these figures are important for demonstrating that LENS is able to capture the baseline properties of our analysis (timing and variability) before presenting the correlation coefficients.

*- Figure 5-7: I suggest the authors modify the color legend here. More color shades is not always better. It is actually not easy to see differences in color shades on a continuous bi-color palette like here, and for the reader things essentially end up being two colors, one positive (green) and one negative (red). It would actually be easier to have fewer shades, more clearly separated, and with perhaps several different colors as well.*

After experimenting with several color and shading schemes, we determined that the

spatial variability in Figures 5–7 are best illustrated using the employed color scheme combined with crosshatching to indicate statistically significant correlations.

*- Figure 8: I suggest showing the mean of the CESM distribution as well.*

We considered including this, but determined that it made the figures too busy without adding useful information.

**References cited in this response:**

[revised manuscript text omitted]

———————————————————

---

## Author Comment (AC4) · 16 Aug 2016

This response is intended to supplement the already-submitted response to the review by Referee #3. As discussed in our main response, the reviewer was concerned that our observational metrics may be compromised by the uncertainty inherent in the satellite data combined with the relatively short time scale ($\sim$13 years). It was suggested that our finding of stronger feedback metrics in models relative to those from the observational data could be the result of observational error degrading the correlations that our metrics are based on.

In order to test the sensitivity of our metrics to this type of error, we used Monte Carlo sampling of the CESM Large Ensemble (LENS) with random noise. At each grid cell,

we added random noise to each data point within the 12–13 year time series. Random noise was generated with numbers sampled from a random Gaussian distribution with a mean of zero and a standard deviation equal to 25% of the standard deviation of the source data. This was repeated 10 times for each of the 38 ensemble members from LENS.

The results of this test are illustrated in the figures below. The perturbed ensemble members did yield weaker correlations, but the difference between the perturbed and original ensemble members was very small relative to the difference between the observed and modeled feedback metrics when averaged across latitudes. This indicates that our metrics are fairly insensitive to this level of random noise, which we believe is a reasonable amount considering the error associated with the remote sensing data sets we use.

We plan to include the figures below as additional supplementary figures to our manuscript. We also plan to mention this test and discuss its implications with some additions and revisions to our text. We plan to add a new sub-section to our methods section, **2.4 Assessment of uncertainty**. We plan to move Page 2, lines 2–7 to this new section, along with the following new text:

> We assessed the sensitivity of our metrics to observational uncertainty using a Monte Carlo sampling approach. For each of the 38 members of LENS, we calculated feedback metrics ten times with random noise added to both TWSA and atmospheric variables. The noise was randomly generated from a Gaussian distribution with a mean of zero and a standard deviation equal to 25% of the standard deviation of the original data. Comparing these results with the original metrics provided some indication of how much our feedback metrics are degraded by random noise as an approximation of observational uncertainty.

We plan to add the following paragraph to the end of **Section 3.3 Evaluating the CESM Large Ensemble**:

Comparison of the original LENS forcing and response metrics with those calculated after adding random noise to LENS provided an estimate of the metrics' sensitivity to observational error. Adding random noise with 25% of the standard deviation of the original data does degrade the metrics slightly, causing areal averages to be closer to zero, but the difference is relatively small compared to the difference between observed and modeled averages as well as the spread of the ensemble itself (Figures S4 and S5). This indicates that we should expect observational error to have a relatively small impact on the quality of our satellite-derived metrics.

Finally, we plan to include the following revised paragraph in our discussion section:

One factor that could contribute toward stronger feedback metrics in models relative to observations is the effect of observational uncertainty combined with a relatively short time series. Adding error to one or more variables in a correlation analysis will reduce the correlation coefficient, and this degradation has been shown to be sensitive to the length of data sets used to establish metrics of land–atmosphere interactions (Findell et al., 2015). Given the relatively short time series available for the current analysis, the correlation coefficients from remote sensing data may be reduced due to observational uncertainty, unlike those derived from internally-consistent models. We found that adding random noise to LENS at 25% of the variance of the original data causes a minor degradation of our area-averaged feedback metrics, but only by a small amount relative to the difference between LENS and the observations (Figure S4 and S5). This indicates that our

feedback metrics, when averaged across large areas, should be relatively insensitive to random error in the observational data.

We believe this additional analysis and the accompanying revisions to the text serve to demonstrate that the issue of observational uncertainty is not the serious issue that Referee 3 indicated they believed it to be.

**Figure captions:**

**Figure 1 (Figure S4 in revised manuscript).** Ensemble histogram of forcing metrics from LENS (grey bars) and LENS plus 25

**Figure 2 (Figure S5 in revised manuscript).** Ensemble histogram of response metrics from LENS (grey bars) and LENS plus 25% random noise (white bars), with the satellite observations from GRACE/AIRS/GPCP/CERES (solid black line) and the alternate observations from GRACE and ERA-Interim (dashed black line), averaged across land regions within different latitude bands.

**References cited in this response:**

Findell, K. L., Gentine, P., Lintner, B. R. and Guillod, B. P.: Data length requirements for observational estimates of land–atmosphere coupling strength, J. Hydrometeorol., 16(4), 1615–1635, doi: 10.1175/JHM-D-14-0131.1, 2015.

[Figure]

**Fig. 1.** Figure S4 in revised manuscript (see caption in text)

[Figure]

**Fig. 2.** Figure S5 in revised manuscript (see caption in text)

---

## Author Response (AR1)

**Author's responses to reviewers' comments**

5

We thank each of the three reviewers for their consideration of our manuscript and for their helpful comments. Referee #1 and Referee #2 each presented a positive perspective on our analysis in their general comments, and provided several constructive criticisms in their specific comments. Referee #3 was critical of several aspects of our paper, and identified issues that they believe warrant important revisions. We agree with many of the points this reviewer makes, and we have made significant revisions to address these points, which we detail below in our response to the reviewer's comments. However, we disagree with the reviewer's perspective that these represent "fundamental issues" with the manuscript and the research presented in it. We believe that the major revisions that we have made, detailed below, serve to clarify our position and improve our acknowledgement of certain limitations and caveats.

15

Specifically, we have clarified that we do not consider our observational metrics as measuring the "true" strength of landatmosphere coupling, but instead recognize that it represents the effects of land-atmosphere coupling combined with remote forcing and atmospheric persistence. This combination of factors comprises a useful set metric that can be derived using remote sensing data and which models must still be able to realistically simulate. In addition, we have greatly enhanced our assessment of observational uncertainty, having given it unique subsections in the Methods and Results sections, and incorporating additional analysis with supplementary figures indicating the implications for our analysis. We believe that with these significant changes to the manuscript, the research represents a meaningful and novel contribution to the field.

20 Below, we have responded to each of the general and specific comments made by each reviewer. We have included each reviewer comment in *italics*, and have responded to each one in roman font. In several instances, we have included quoted passages from our revised manuscript with reduced margins so that these sections can be visually distinguished with ease. Following the point-by-point responses, we have included a marked up revision of the document comparing it with the original submission.

**25 **Point-by-point responses to Referee #1**

This manuscript evaluates the strength in the two-way relationship between total water storage (TWS) and atmospheric conditions within a suite of satellite remote sensing, reanalysis, and climate model systems. The correlation method for evaluating the coupling strength is not novel, but the concurrent assessment of both response and feedback "limbs" is. Further, comparison of the response and feedback strengths with model and satellite datasets provides additional evidence

30

1

of the dichotomy in the apparent strength (and in some documented cases, the sign) of soil moisture-atmosphere coupling between model and observational frameworks. As the authors allude to, these model system issues can influence estimates of future warming/drying trends. I enjoyed reading this very well written manuscript. The results and implications are international in scope, and it will make a worthy contribution to the land-atmosphere interaction body of knowledge. The outstanding issues I outline below are mostly minor, and the manuscript will be suitable for publication once they are addressed/clarified appropriately.

5

10

15

**We thank the reviewer for these positive comments**

Methods: Page 5, line 16: "with temporal gaps filled using linear interpolation." More detail is needed here. How many months in the data record are filled? What was the typical time interval of missing data; for example, were the "gaps" filled predominantly just 1 month or multiple consecutive missing months of data? Is a linear interpolation reasonable for temporal gap-filling GRACE data?

Missing months in the GRACE record are discussed at the following webpage: http://grace.jpl.nasa.gov/data/grace-months/. We limited the time series to September, 2002 through November, 2014, in order to reduce the number of temporal gaps. Within this time range, there are eight non-consecutive gaps of one single month and one gap of two missing months. Linear

interpolation was chosen based on personal communication with colleagues experienced in the use of gridded land data from GRACE Tellus. We clarified this with the following revised text:

We obtained Level-3 TWSA data from GRACE using the University of Texas at Austin Center for Space Research (CSR) spherical harmonic solutions (Swenson, 2012). Global land data at a 1° resolution were scaled using the coefficients provided by Landerer and Swenson (2012). The study period was limited to September 2002 through November 2014, in order to minimize temporal gaps. GRACE data during the study period included eight non-consecutive and two consecutive missing months, which were filled using linear interpolation.

25

30

Page 5, lines 20-22: "...the use of TWS data allows us to include surface storage, canopy storage... all of which may be sources of moisture that are potentially limiting factors for ET". This is just one specific instance of discussing TWS for land-atmosphere analysis, but in general I think the paper would be better served with a more complete discussion of the advantages and limitations of using TWS for this purpose. For example, is the groundwater component of TWS a limiting factor for ET if the rooting depth does not reach the water table? In many temperate agrarian or grassland landscapes this is the case. How do you think these issues could potentially affect the results, or more specifically do you think your findings would be different if you just used (e.g.,) surficial soil moisture or root-zone soil moisture?

These points are well taken. The primary advantage of using TWS data from GRACE is the fact that it is the only multi-year global remote sensing product that includes root-zone soil moisture, unlike AMSR-E and others that only include nearsurface soil moisture. The importance of including root-zone soil moisture is mentioned in the Introduction section, which we have revised, as detailed below, in order to clarify the use of TWS instead of surface soil moisture.

5

10

20

25

30

Regarding groundwater, if the rooting depth does not reach the water table, then groundwater would not serve as a limitation on ET. However, in agricultural and other inhabited areas, aquifer withdrawal for irrigation does provide a connection between groundwater and the atmosphere. Furthermore, the rate of aquifer recharge depends on how much available water is removed via ET. Finally, rooting depths are not always well understood, and the boundary between rooting zone and aquifer may not be known. Each of these points indicates an advantage to including the aquifer component of TWS. The disadvantage of including groundwater is that the metrics could be sensitive to long-term trends in groundwater, but these changes are likely to be relatively small on seasonal timescales and therefore do not provide a strong disadvantage.

We have removed segment in question, Page 5, lines 20–22, from the Methods section and have added the following text to the Introduction section in order to clarify and expand upon the importance of including the full TWS column for capturing root-zone soil moisture and other important components:

Until recently, studies using remote sensing data to look for evidence of land-atmosphere coupling relied on products that provide information about surface soil moisture (Ferguson et al., 2012; Taylor et al., 2012). Consideration of root-zone soil moisture has recently been accomplished only indirectly via dataassimilated estimates (Guillod et al., 2015). The inability to directly consider root-zone soil moisture has been suggested as an explanation for the relatively weak coupling observed using remote sensing data (Hirschi et al., 2014). In order to include root-zone soil moisture, as well as other sources of moisture available across entire seasons, here we analyzed remote sensing data of the entire terrestrial water storage (TWS) column.

The metrics introduced here were specifically designed to use the monthly TWS anomaly (TWSA) anomaly product from the Gravity Recovery and Climate Experiment (GRACE) mission (Landerer and Swenson, 2012; Wahr et al., 2004). The GRACE TWSA product integrates soil moisture at all layers along with surface, canopy, snow/ice, and aquifer storage, as each of these components represents a potential source of moisture for fulfilling evaporative demand. For example, in areas where agricultural ecosystems are important, diversion of lake and river water resources and withdrawal from aquifers may contribute to irrigation fluxes and thus ET. Furthermore, surface storage of liquid water and snow represent sources of

water that are available for and potentially limiting to ET. Under these conditions, month-to-month TWS anomalies capture portions of the terrestrial water cycle that soil moisture alone may not.

Results: Page 10, lines 23-26: you find consistently weaker forcing relationships in boreal regions and attribute this to
"high levels of climate variability in many high latitude regions" because of AO, NAO, and the like. However, do you think the predominantly energy-limited evaporative regime of many boreal regions contributes to limiting the feedback connection between soil moisture and atmospheric conditions? I would not expect the forcing "limb" to match the strength of the "response" limb in such conditions where evaporative fraction is more a function of incoming radiation than soil moisture availability.

10

This is a very good point, and we have added the following to address this:

Furthermore, at high latitudes, ET is generally energy limited rather than moisture limited, which would lead to weak forcing metrics as moisture availability would not strongly influence atmospheric conditions.

15

20

25

Page 11, lines 28-29: "Furthermore, the well-understood physical mechanisms allow causality to be inferred even when not directly demonstrated." The argument in the previous sentence is well-taken, but I disagree with the premise of this statement, particularly inferring causality in the forcing "limb" in the absence of discussion/analysis of possible confounding effects. At the very least I would like to see some assessment or acknowledgement of the role of atmospheric persistence as a confounding factor when quantifying the forcing limb. For example, is a strong forcing limb caused by the physical constraint of soil moisture on energy partitioning and modification of boundary layer dynamics and thermodynamics (as suggested by the authors), or is it the result of large-scale atmospheric circulation and persistence of synoptic-scale patterns that modify precipitation and atmospheric demand throughout the duration of the TWS draw down season? A correlation coefficient cannot adequately address the question of large-scale atmospheric persistence vs. soil moisture feedback, and indeed this is beyond the scope of this manuscript. However, this does also mean that one cannot infer causality and mechanistic connections between soil moisture and VPD/PPT/SW based on the evidence provided here.

This point is well taken, and we have removed the sentence in question so as not to claim any indication of causality. We have expanded the discussion just prior to the sentence in question to address the issue of persistence:

30

The metrics developed here from satellite observations provide a means for evaluating land-atmosphere feedback strength on seasonal to interannual timescales in coupled ESMs. The use of correlation coefficients in this study does not enable a direct assessment of whether the relationships are directly causal, as correlation between atmospheric and terrestrial conditions could result from atmospheric

persistence and remote forcing from SSTs (Orlowsky and Seneviratne, 2010; Mei and Wang, 2011). Nonetheless, the satellite-derived metrics provide a meaningful constraint against which coupled models can be benchmarked, as these models need to correctly represent the combined effects of persistence, remote SST forcing, and land–atmosphere coupling.

5

10

15

20

25

30

We also have addressed this by adding a paragraph to Section 4.5 Uncertainties and future applications:

Finally, the issue of causality and the possibility that correlations result primarily from atmospheric persistence and remote forcing from SST rather than land-atmosphere interactions may be addressed using sensitivity experiments similar to those of GLACE and GLACE-CMIP. While the previous experiments have tested the importance of soil moisture interaction with the atmosphere, additional experiments could expand upon these methods by treating SST variability similar to terrestrial soil moisture availability. Such experiments could determine the relative importance of remote SST forcing, including the effect of atmospheric persistence, and local land–atmosphere coupling in explaining correlations between TWS and atmospheric conditions.

Page 12, lines 14-16: "That models and ensemble members with high forcing metrics were also found to have high response metric... highlights the importance of the land surface response in priming the system for subsequent forcing on the atmosphere..." I don't understand how the land surface response "primes the system" for subsequent forcing when your analysis (and Figure 2) suggest the forcing occurs prior to the response. What am I missing? Can you please expound?

We have clarified the discussion by replacing the lines in question with the following:

The inclusion of response metrics in our analysis allows the full feedback loop to be considered by recognizing the two-way dependence between the land surface and the atmosphere. The generally higher correlation coefficients in observed response metrics indicates the importance of the land surface response in priming the system for subsequent forcing on the atmosphere. For example, if the TWS response is too strongly coupled to the atmosphere, a small change in atmospheric conditions could yield an unrealistically large change in TWS. The unrealistically large TWS anomaly, in turn, would have the potential to impart a larger land surface forcing of the atmosphere in subsequent time steps. That models and ensemble members with high forcing metrics were also generally found to have high response metrics (Figure 10) highlights the need to consider this.

Conclusion: Page 16, lines 13-14: "...which suggests that some of these models may have difficulty properly predicting warming trends and climatic extremes." You include an excellent discussion of the potential links between model overestimation of land surface forcing and warming trends, founded in the body of literature. However, you do not explicitly quantify this linkage in this manuscript. The ability of models to "properly" predict warming trends and climatic extremes is not evaluated here, so this statement should probably be removed.

This point is well taken. We have rephrased this section as follows, to qualify our conclusion and avoid making conclusions that were not evaluated in our manuscript:

10 Modeled coupling metrics are generally found to be stronger than those observed in the satellite record. If this discrepancy is due to models overestimating the two-way feedback between the land surface and the atmosphere, this could lead to models incorrectly projecting future warming trends and climatic extremes (e.g., Hirschi et al., 2011; Seneviratne et al., 2013; Cheruy et al., 2014; Miralles et al., 2014).

15 Page 11: I'm going to guess the section 4 title should be "Discussion" and not "Introduction"

Page 12, line 21: parentheses should only be around the publication year - Guo et al., (2006)...

We thank the reviewer for pointing out these typographical errors, which have been corrected

**20 **Point-by-point responses to Referee #2**

This paper describes a new methodology for the analysis of land-atmosphere coupling that focuses on the seasonal time scale and that addresses both the widely-studied impact of the land surface on the atmosphere and the somewhat less often considered (in this context) sensitivity of the land surface to atmospheric forcing. This methodology is applied to remote-sensing data of terrestrial water storage (TWS), the CESM Large Ensemble (LENS) and a few CMIP5 models.

25

Although many methodologies to quantify land-atmosphere coupling already exist, the focus on seasonal time scales and the investigation of both the forcing and feedback loop of land-atmosphere coupling make this methodology valuable. The use of LENS is also valuable and in particular is useful to consider potential impacts of internal variability on the metrics.

- 30 The manuscript is very well written and very clear. The authors have done a good job at keeping the text relatively short and straightforward despite substantial analyses. Most of my comments below are minor and I think the manuscript can be published after considering these.
  - 6

5

We thank the reviewer for these supportive comments.

Page 4, line 16-17: "Until recently, studies using remote sensing data to look for evidence of land-atmosphere coupling
relied on data products that provide information about surface soil moisture." There are some exceptions and some remotesensing products also estimate soil moisture in the whole root zone (for instance, I recall that Guillod et al. (2015) used such estimates).

- We re-read the Guillod et al. (2015) paper, and found that while they did account for root-zone soil moisture in their estimation of evaporative stress, this was done indirectly through a data-assimilated estimate. Their direct observations of soil moisture were of surface soil moisture in the top few cm of the soil column from AMSR-E. We have added the following explanation of these details to the text in order to clarify the distinction between direct and indirect observations of root-zone soil moisture:
- 15 Consideration of root-zone soil moisture has recently been accomplished only indirectly via dataassimilated estimates (Guillod et al., 2015).

Section 3.3: These results on the role of internal variability are interesting. Another source of discrepancy between a single model run and observations can be observational error combined with short record (e.g., Findell et al., 2015), which may artificially decrease the coupling found in observations. Together, this effect and internal variability probably tell us something about the data length required to robustly assess land-atmosphere coupling.

This point is well taken, and we have expanded upon this issue by performing a Monte Carlo simulation in order to test the effects of observational uncertainty combined with our relatively short time series. We have made additions to the Methods and Results sections to report on the details of this approach, as well as an addition to our Discussion section, as follows, in which we acknowledge the relevance of Findell et al., 2015:

30

20

25

One factor that could contribute toward stronger coupling metrics in models relative to observations is the effect of observational uncertainty combined with a relatively short time series. Adding random error to one or more variables in a correlation analysis will reduce the correlation coefficient, and this degradation has been shown to be sensitive to the length of data sets used to establish metrics of land–atmosphere coupling (Findell et al., 2015). Given the relatively short time series available for the current analysis, the correlation coefficients derived from remote sensing data may be reduced due to observational uncertainty, unlike those derived from internally consistent models. We found that adding random noise to LENS at

25% of the standard deviation of the original data caused some degradation of our area-averaged coupling metrics, but only by a small amount relative to the difference between LENS and the observations (Figures S1 and S2). We chose 25% as a qualitative upper bound on likely uncertainties introduced from random observational error within the TWSA and atmospheric variable time series. This highlights the need for developing more quantitative error estimates in remote sensing and reanalysis products. More generally, this sensitivity analysis suggests that our coupling metrics, when averaged across large areas, may be useful in identifying robust data-model differences.

This addition serves to strengthen the conclusion that the utility of our approach will increase as the satellite data record grows longer, and reinforces the importance of the GRACE follow-on mission in lengthening the time series of TWS anomalies. We have further modified our discussion section to emphasize this point as follows:

Furthermore, we acknowledge that observational error over an insufficiently long time series could reduce the apparent strength of correlations (Findell et al., 2015). Therefore, the utility of the coupling metrics we present will increase alongside the length of the time series available from remote sensing platforms. This emphasizes the importance of the GRACE follow-on mission (Flechtner et al., 2014) and the need for continuity in the record between missions.

Page 8, line 24 and Page 11, line 5: Supplementary figures references could be more specific. I think that the first mention
on page 8 refers to Fig. S3 while the second mention (page 11) refers to Figs. S1 and S2. The order of the supplementary
figures could also be adapted so they are cited in sequence.

We thank the reviewer for pointing out the need to clarify our references to the supplementary figures and the order in which we present them. The first mention, on page 8, was intended to refer generally to all three figures, but the lack of specificity
did not yield clarity in our order of presentation. We have modified our references to the supplementary figures to be more specific, and have made sure that the supplementary figures are numbered in the order in which they are referred to in the text.

Page 13, line 20-29: The authors write that overestimating ET would lead to excessive land-atmosphere coupling. This is a
bit confusing: if the coupling is strongest in transitional regions between wet and dry climate, why would a too high ET in an
already wet region lead to an increase in coupling? I agree that model errors with respect to the distribution of water
between storage reservoirs can be an issue; however I do not think that this necessarily leads to stronger coupling, but
maybe I am missing something here.

5

15

We thank the reviewer for pointing out the need to clarify the basis of this argument. We have replaced the paragraph in question with the following:

A set of possible explanations involves models overestimating the amount of water available for ET during the drawdown interval. The land surface influence on the atmosphere requires water to be a limiting factor to ET but not limiting enough to prevent it altogether. Under more moisture-limited conditions, a drawdown interval may experience multiple shorter time periods during which ET is inhibited due to insufficient water, and the terrestrial moisture state exerts no control over flux partitioning. These periods of insufficient moisture would tend to reduce the overall feedback strength integrated across the duration of the drawdown interval. Model shortcomings that make water too readily available for ET could reduce the amount of time spent in a periods of insufficient moisture during the drawdown interval, thereby unrealistically strengthening the longer-term feedback. We note that the opposite could take place under near-saturated conditions if a model overestimates the amount of time in which ET is energy-limited, but we would not expect these conditions to be as prevalent during the drawdown interval that was the time period of focus in our analysis.

Page 11, line 23: I assume that "Introduction" should in fact be "Discussion".

Page 12, line 22: "(Guo et al., 2006) explain..." should be "Guo et al. (2006) explain...".

20

15

5

10

Page 13, line 13: "and underestimation of and remote SST forcing" should be "and underestimation of remote SST forcing" (?).

Page 16, line 11: "Regions of strong coupling metrics": I think the authors mean "Regions of strong response metrics" 25 here.

We thank the reviewer for pointing out these typographical errors, which have been corrected.

**Point-by-point responses to Referee #3**

30

In this study, the authors aim to assess the feedbacks between terrestrial water storage (TWS) and the atmosphere in remote sensing observations (using GRACE data for TWS), and they aim to use the estimate they obtain to evaluate climate model simulations from the CESM large-ensemble and from (a subset of) CMIP5 models. They use correlations at the interannual time-scale, with lead-lags of a few months between TWS and atmospheric variables, to assess both how the atmosphere

(during months of TWS decrease) influences the minimum annual TWS (the forcing response), and how TWS (at time of maximum annual TWS) feeds back on the atmosphere in the following months (the feedback response). Their main result include some characterization of these relationships in model and observations, which show that the CESM model, and most CMIP5 models, appear to overestimate both forcing and feedback relationships. Some discussion is then proposed on the possible reasons for that.

5

20

Although I like the idea of trying to use GRACE observations to derive a large scale "land-atmosphere moisture coupling" metrics to compare to models, I see a number of issues with the study as it stands, that I think warrant some major revision.

10 We thank the reviewer for their thorough and thoughtful analysis of our manuscript. While we disagree with several of the issues raised by this reviewer, many of their comments proved to be highly constructive criticism. We have addressed these comments with some major revisions to our manuscript as well as several minor corrections and clarifications, all of which we detail below in our point-by-point responses. In particular, we have made significant revisions in order to clarify the goals of our approach and more appropriately emphasize its limitations. We hope these clarifications and revisions will 15 satisfy the reviewer, and we believe that despite the limitations, our approach still represents a valuable and novel system for evaluating model performance using observational constraints.

1) My first main comment has to do with the metric used. First, the brevity of the time period of analysis is obviously an issue. The metrics are essentially interannual correlations over 13 years, i.e., correlations over 13 data points. That's very short. I am not sure I have seen many studies looking at interannual variability over 13 years only, in particular in terms of land-atmosphere studies. Accordingly, results for the feedback metrics appear very noisy. First, I believe a discussion of field significance is warranted here: patches of apparently significant values may still be random in that context (e.g., see Livezey and Chen (1983)).

- 25 We do not believe the issue of field significance is relevant in the current context. Our metrics compare a single time series of TWS anomalies with a single time series of atmospheric data in the same region. We are not calculating correlations between a single explanatory variable and a geographically distributed field of dependent variables. Therefore, we are not engaging in the type of hypothesis testing that would warrant consideration of field significance.
- 30 Second, note that recent research underscores the need for long-record datasets to establish land-atmosphere coupling, that coupling metrics require more data than single-variable simple statistics (e.g., mean and variance) to be robustly estimated, and finally that, unlike single-variable statistics, coupling metrics are actually degraded by observational uncertainty (Findell et al. 2015). The latter point, in particular, is in my view a much likelier explanation for the weaker correlations found here in observations – between uncertain observation datasets that are independent of each other – compared to

correlations computed with model outputs, which are by definition perfectly consistent with each other. The authors touch on the issue of observational uncertainty by computing correlations with the ERA reanalysis, but I don't think enough is made of that.

We recognize that the relatively short time series available from the satellite record warrants caution while interpreting these results. We agree with the reviewer that the findings of Findell et al. (2015) emphasize this limitation and suggest that observational uncertainty would be expected to decrease correlations. We have addressed this using a Monte Carlo approach, which we describe in the revised manuscript and present the results of in additional supplementary figures. In summary, we treated the CESM Large Ensemble (LENS) as "truth" in order to test how much the correlation coefficients that comprise our metrics are degraded by random noise as a proxy for observational uncertainty. We added random noise with a mean of zero and a standard deviation equal to 25% of the standard deviation of the variables in question before performing the correlation analysis. We found that perturbing the data did reduce the correlation coefficients by a small amount, but the difference between the metrics from the original and perturbed model data is much smaller than the difference between the metrics from the remote sensing observations. We therefore conclude that while observational uncertainty combined with the relatively short time series is a caveat that must be considered, it does not invalidate the overall findings of our analysis.

Along with the addition of this Monte Carlo approach, we have made major revisions to the presentation of our uncertainty analysis in the manuscript. There are now dedicated subsections in both the Methods and Results section detailing our uncertainty analysis, as well as the following addition to our Discussion section, in which we specifically reference Findell et al. (2015):

20

25

30

One factor that could contribute toward stronger coupling metrics in models relative to observations is the effect of observational uncertainty combined with a relatively short time series. Adding random error to one or more variables in a correlation analysis will reduce the correlation coefficient, and this degradation has been shown to be sensitive to the length of data sets used to establish metrics of land–atmosphere coupling (Findell et al., 2015). Given the relatively short time series available for the current analysis, the correlation coefficients derived from remote sensing data may be reduced due to observational uncertainty, unlike those derived from internally consistent models. We found that adding random noise to LENS at 25% of the standard deviation of the original data caused some degradation of our area-averaged coupling metrics, but only by a small amount relative to the difference between LENS and the observations (Figures S1 and S2). We chose 25% as a qualitative upper bound on likely uncertainties introduced from random observational error within the TWSA and atmospheric variable time series. This highlights the need for developing more quantitative error estimates in remote sensing and reanalysis products. More generally,

this sensitivity analysis suggests that our coupling metrics, when averaged across large areas, may be useful in identifying robust data-model differences.

This acknowledgement further supports our argument that the utility of these metrics will increase as the time series of global satellite data grows longer with continuations of current missions and initiation of new missions (i.e., GRACE follow on) as mentioned in the Discussion section:

5

10

15

The current study demonstrates the utility of the coupling metrics presented here, but conclusions are limited by the time span of the satellite record. While LENS enables the internal variability of these relationships to be investigated within the model, it is unclear how much natural climate variability affects these relationships in reality on timescales longer than the satellite record. Furthermore, we acknowledge that observational error over an insufficiently long time series could reduce the apparent strength of correlations (Findell et al., 2015). Therefore, the utility of the coupling metrics we present will increase alongside the length of the time series available from remote sensing platforms. This emphasizes the importance of the GRACE follow-on mission (Flechtner et al., 2014) and the need for continuity in the record between missions.

So, given the brevity and uncertainty of observations, even without consideration of any other issues (but, see below), I am really uncomfortable with the approach proposed by this paper, which is to consider the observational estimate as a
benchmark for model evaluation. Personally, I think an approach where observations and model results are used together to try to infer the "real" coupling would make more sense here. But, this would lead to a very different paper. Overall, if the authors are going to go on with their approach, I would recommend much more caution in how things are presented, including in the title of the study and the conclusions.

We agree with the reviewer that "an approach where observations and model results are used together to try to infer the 'real' coupling" would be valuable, and represents a research priority for the community. Our intention was not to present the observationally derived forcing metric as representing the "real" land–atmosphere coupling strength. Instead, it represents the combined effects of land–atmosphere coupling (the "real" coupling strength) along with the remote effects of SST forcing on both the atmosphere and land surface. We believe that despite the relatively short time series, these metrics provide a useful constraint on models' ability to represent this combined set of processes. The reviewer's recommendation of greater caution in our presentation is appreciated, and we have added the following clarification to Methods section when introducing the metrics:

We note that these metrics do not provide distinctive information for measuring the strength of landatmosphere coupling or the land surface response. While the metrics include the influence of direct landatmosphere interactions, they are also potentially influenced by atmospheric and soil moisture persistence, as well as remote forcing from sea surface temperatures (SSTs) (Orlowsky and Seneviratne, 2010; Mei and Wang, 2011). Nevertheless, these metrics may still serve as useful benchmarks against which to evaluate the ability to ESMs to reproduce the proper relationships based on the combination of these factors.

We further clarify this point in the following addition to the Discussion section:

10 The metrics developed here from satellite observations provide a means for evaluating land-atmosphere feedback strength on seasonal to interannual timescales in coupled ESMs. The use of correlation coefficients in this study does not enable a direct assessment of whether the relationships are directly causal, as correlation between atmospheric and terrestrial conditions could result from atmospheric persistence and remote forcing from SSTs (Orlowsky and Seneviratne, 2010; Mei and Wang, 2011).
15 Nonetheless, the satellite-derived metrics provide a meaningful constraint against which coupled models can be benchmarked, as these models need to correctly represent the combined effects of persistence, remote SST forcing, and land-atmosphere coupling.

We also plan to emphasize the importance of disentangling the influence of land–atmosphere coupling from that of atmospheric persistence and remote SST forcing with the following addition to the Discussion section:

Finally, the issue of causality and the possibility that correlations result primarily from atmospheric persistence and remote forcing from SST rather than land-atmosphere interactions may be addressed using sensitivity experiments similar to those of GLACE and GLACE-CMIP. While the previous experiments have tested the importance of soil moisture interaction with the atmosphere, additional experiments could expand upon these methods by treating SST variability similar to terrestrial soil moisture availability. Such experiments could determine the relative importance of remote SST forcing, including the effect of atmospheric persistence, and local land–atmosphere coupling in explaining correlations between TWS and atmospheric conditions.

30

25

5

We believe that despite the limitations of a relatively short time series and the inability to attribute the sources of covariability, our approach is still valuable. We believe that the revisions described above emphasize our goal of conceptually illustrating an approach towards model benchmarking that will become increasingly useful with longer time

series from remote sensing. At this point, we would prefer to retain our title, which we believe is succinct and accurately conveys the content of our paper.

2) Another issue with the metrics involves the definition of the (feedback) metric. The way it is defined, it is looking at the
impact of TWS at the end or peak of the rainy season on climate in the subsequent months. The authors indicate as much, and say they want to consider, in the Tropics, the impact of late-rainy season TWS on dry-season climate. I see two issues with that. First, in my view, while that may useful in the deep Tropics where the dry season is short, this approach is problematic in monsoon regions, or regions of the Tropics that have a well defined rainy season (i.e., outside of the deep Tropics): basically, after the rainy season, there is not much rain to look at any more. For instance, over the Sahel, what the authors are computing is the impact of September TWS on precipitation over September-May. But it doesn't rain much over that time period.

We believe that focusing on the drawdown interval is an important part of our approach. Our algorithm is novel in allowing a global-scale analysis across ecosystems. In mid-latitudes, the drawdown interval contains the summer season that has been
the focus of research in land-atmosphere coupling. In tropical latitudes, the drawdown interval contains the dry season, during which precipitation recycling is important for maintaining ecosystems, allowing forests to persist in the absence of circulation-driven precipitation (Keys et al., 2016). In the example of the Sahel, our algorithm is working as intended, by measuring how variations in TWS at the onset of the dry season are related to atmospheric conditions during the dry season.

- 20 In my view, it would be much more interesting to look at the impact of end-of-dry season TWS on the subsequent rainy season to see if, in these regions, available land moisture feeds back on precipitation during the rainy season. Second, in the same example over West Africa, whatever rainfall there is over Sept-May is actually probably the end of the monsoon, Sept-Nov. Because TWS in September is likely to be influenced by precip in September, and Sept. precip is likely to represent large part of the 'response' variable, the causation is muddied a little bit: a clearer temporal offset would be needed in such
- 25 a case. But more importantly, even precip in the months following September (Oct, Nov) is likely to be correlated with precip in the previous months – for instance, a year with a strong monsoon that has more rain in Jun-Sept may well tend to also have more rain in Sept-Nov. Because September TWS will largely reflect JJAS rainfall, the TWS-based metric will then show a strong feedback - but the inferred causation would be a misinterpretation.
- 30 We believe that ET in the wet season tropics would be energy limited, and therefore would not reflect the influence of land surface moisture availability on the atmosphere. We acknowledge the issue with persistence, which we expand upon below.

This brings me to a more general point: the authors do not discuss how autocorrelation, here at the seasonal time scale, of climate variables, may impact their estimate of land-atmosphere coupling. This is a major issue affecting all empirical

studies of land-atmosphere coupling – see, for instance, Wei et al. (2008) and Orlowsky and Seneviratne (2010). The authors do cite the latter study, but, it seems, simply to say that if models underestimate SSTs influence on land climate, they will then appear to overestimate local L-A coupling. They somehow miss the point of that paper in how it should apply to to their own results. I just gave one example above (the Sahel) of how that might be the case. Other monsoon regions (e.g., India) might similarly be affected. Interannual variability in the coupled ocean-atmosphere (eg., ENSO) might also the source of

5

might similarly be affected. Interannual variability in the coupled ocean-atmosphere (eg., ENSO) might also the source of confounding influence at the time scales investigated here. So, overall, I recommend these caveats be considered and discussed by the authors in their interpretation of their results. Personally, I would need to see some further analysis to be more convinced of the physical reality of the land-atmosphere feedbacks the authors claim to show (e.g., some sensitivity test to the months and time lags considered, some investigation of atmospheric variability and persistence, etc.).

10

The reviewer's point is well taken, and has already been partially addressed above in the additions and revisions to the Methods and Discussion sections (quoted above). In addition, we discuss these issues more explicitly in an additional revision to the Discussion section:

15 Another possible explanation stems from the fact that our coupling metrics include covariability due to atmospheric persistence and remote forcing by SST (Orlowsky and Seneviratne, 2010; Mei and Wang, 2011) alongside the direct influence of land-atmosphere interactions. For this reason, we caution that overestimates of coupling metrics do not imply that the land-atmosphere feedback is necessarily stronger, but could be due to an overestimate of SST-driven correlations between the land surface and the 20 atmosphere. Wei et al. (2008) demonstrated that negative correlations between soil moisture and subsequent precipitation can be explained by precipitation persistence combined with negative temporal autocorrelation of precipitation associated with intra-seasonal modes such as the Madden-Julian Oscillation (MJO). Poor representation of the MJO period in CMIP5 models leads to unrealistic patterns of precipitation persistence (Hung et al, 2013). If models are failing to capture MJO-driven negative 25 correlations, this could lead to overly strong positive correlations relative to observations. However, this would depend on the length of the drawdown interval relative to persistence time and the period of intraseasonal modes.

30

This supports our addition the Discussion section (quoted above), discussing the importance of modeling experiments to determine relative importance of remote SST forcing, including the effect of atmospheric persistence, and local landatmosphere coupling in explaining correlations between TWS and atmospheric conditions.

3) Another main comment has to do with the discussion section. The authors discuss why models might exhibit stronger feedback (and forcing) metrics than observations. As mentioned above, I think uncertainty in observations should be mentioned as a primary reason.

5 We now more explicitly address observational uncertainty as well as uncertainty due to the short time series in the Discussion section, quoted above.

The authors propose that ET may be consistently overestimated in climate models, and a large part of the discussion then consists in speculation as to why that may be the case. First, while I appreciate the effort to discuss things further and not just show results, I found this whole section a bit too speculative. IF the models overestimate ET, then IF stomatal conductance, IF convection, IF bare soil, etc. . . Can the authors actually point to any evidence that ET is consistently overestimated in climate models, in the first place (or at least in CESM)?

10

These points are well taken, and while this section is speculative by its very nature, we agree that it warrants revision. We have modified this part of the Discussion section so that it does not center on models overestimating ET, but instead focuses on ways in which models could make moisture too readily available for ET. We have clarified the basis of our argument with the following revision to the Discussion section:

A set of possible explanations involves models overestimating the amount of water available for ET during the drawdown interval. The land surface influence on the atmosphere requires water to be a limiting factor to ET but not limiting enough to prevent it altogether. Under more moisture-limited conditions, a drawdown interval may experience multiple shorter time periods during which ET is inhibited due to insufficient water, and the terrestrial moisture state exerts no control over flux partitioning. These periods of insufficient moisture would tend to reduce the overall feedback strength integrated across the duration of the drawdown interval. Model shortcomings that make water too readily available for ET could reduce the amount of time spent in a periods of insufficient moisture during the drawdown interval, thereby unrealistically strengthening the longer-term feedback. We note that the opposite could take place under near-saturated conditions if a model overestimates the amount of time in which ET is energy-limited, but we would not expect these conditions to be as prevalent during the drawdown interval that was the time period of focus in our analysis.

We have also added the following to the Discussion section, citing evidence of models overestimating ET:

CMIP5 models are known to have a high ET bias (Mueller and Seneviratne, 2014), which could be due in part to the explanations proposed as possible reasons for overestimated feedback metrics in models.

Second, if soil water is too readily available in models, and ET is overestimated, wouldn't that actually mean that feedbacks should be underestimated in models? Indeed, ET would then be less water-limited and more energy-limited, with less potential for soil moisture-atmosphere feedbacks.

We designed our metrics around the drawdown interval in order to specifically consider the time of year during which ET would be water-limited. The issues we discuss with insufficient representation of bare soil processes and big leaf parameterizations would, during this time of year, unrealistically make too much water available for ET. This would allow ET to take place in the model when in reality that water would have run off or is unavailable for transpiration. Under these conditions, the atmosphere in the model would be influenced by moisture availability when in reality no moisture is available. These points have been clarified with the revision to the Discussion section quoted above.

- 15 Surface climate variability would then be influenced by the atmosphere to a greater extent. Along the same lines, the authors claim that their results, showing an overestimation of land-atmosphere feedback by models, are consistent with prior studies, and have implications for projected warming (e.g., Cheruy et al., 2014). However, these previous studies, it seems to me, point to ET being underestimated in these models, and models getting to easily "locked" in a dry and warm surface mode. So, in effect, while the authors agree with prior studies that land-atmosphere feedbacks are overestimated in models, they
- 20 provide opposite reasons for that (overestimated versus underestimated of ET). I would like to see the authors clarify that apparent contradiction.

We have revised our Discussion section, described above, so as not to rely on whether models overestimate or underestimate ET.

**25**

30

4) Finally, the author interpret the relationship they find between the strength of the feedback and forcing metrics in CMIP5 models as showing that: "the response limb of the feedback loop is important for understanding how conditions are set up for subsequent forcing via land–atmosphere coupling". They claim that it highlights "the importance of the land surface response in priming the system for subsequent forcing on the atmosphere". I am not convinced by this interpretation, which sounds a bit hand-wavy to me. I don't see a strong physical reason why a model where, for instance, TWS responds strongly to precipitation, should have a strong feedback of TWS onto precipitation.

Conceptually, we disagree with this perspective. The strength of land-atmosphere coupling depends on moisture availability enabling some ET while still limiting it. Models must therefore simulate the correct moisture availability in order to simulate

5

the proper amount of land-atmosphere coupling. The response metrics reflect whether models simulate the right moisture availability based on precipitation and evaporative demand, and whether this is the right amount to set up subsequent land-atmosphere coupling. We have clarified our reasoning with the following modification to the Discussion section:

- 5 The inclusion of response metrics in our analysis allows the full feedback loop to be considered by recognizing the two-way dependence between the land surface and the atmosphere. The generally higher correlation coefficients in observed response metrics indicates the importance of the land surface response in priming the system for subsequent forcing on the atmosphere. For example, if the TWS response is too strongly coupled to the atmosphere, a small change in atmospheric conditions could yield an unrealistically large change in TWS. The unrealistically large TWS anomaly, in turn, would have the potential to impart a larger land surface forcing of the atmosphere in subsequent time steps. That models and ensemble members with high forcing metrics were also generally found to have high response metrics (Figure 10) highlights the need to consider this.
- 15 Couldn't the relationship on Figure 10 be due to intermodel differences in what TWS (or its estimate, here) encompasses in each model? For instance, different soil depths? A deeper soil would lead to weaker links between TWS and climate both in terms of response and feedback to the atmosphere.

No, because we are using the total terrestrial water storage column. In the case of LENS, this is an explicitly output field that includes this entire column. In the case of the CMIP5 output, we used the accumulated residuals of the surface water balance (i.e., the integral of precipitation minus evaporation and runoff) to approximate this quantity.

In any case, I found Figure 10 to be insufficiently explained and encourage the authors to discuss this further.

25 We have revised the caption for Figure 10 to more clearly explain what it comprises:

Scatter plots of forcing and response metrics for LENS and CMIP5 models with observations, averaged across land regions within different latitude bands. For LENS, we show metrics calculated using the explicit TWS output (darker grey) and TWSA estimates from the accumulated residuals of the surface water balance (lighter grey). For CMIP5 models, we calculated metrics using TWSA estimates from the accumulated residuals of the surface water balance.

p.2 line 4: "cloud radiative coupling": please explain and clarify.

30

We have clarified this as follows:

Cloud radiative coupling can likewise lead to positive or negative feedbacks as insolation and evaporative demand, as a function of cloud cover, are either enhanced or suppressed (Betts, 2009; Cheruy et al., 2014).

5

*P.2 line 24: actually, no: a surprising result of GLACE II was that predictive skill was not enhanced over the Great Plains "hot spot" from GLACE I, but rather to the North of it (see Koster et al. 2011). Consider rephrasing*

We thank the reviewer for pointing out this discrepancy, which we have corrected by removing the reference to GLACE II

10

P.3 line 5: the text should make it clear that GLACE-like metrics cannot be directly compared to observations, and that other more simple metrics, not strictly equivalent, have to be used, like SM-ET correlations, etc.

We agree that this warrants clarification, and we have done so with the following revision:

15

30

The metrics developed for GLACE are based on model experiments with no direct observational equivalents. However, correlation-based metrics that do enable direct comparison with observations suggest that models may overestimate land–atmosphere coupling strength (Dirmeyer et al., 2006a).

20 P.3 line 20: Findell et al. 2011 is actually based on reanalysis data, not "modelling". Also, Findell et al. 2015 should be included in this discussion, to highlight the issue, discussed above, of data length requirements to estimate land-atmosphere coupling.

We reference Findell et al. (2011) in the context of Guillod et al. (2014), which emphasizes that the surface state and fluxes are still model based, even if the atmosphere is constrained by some observations. However, to avoid confusion, we now omit the word "modelling" from the description of Findell et al. (2011). Furthermore, as discussed above, we now include Findell et al. (2015) in the discussion in Section 4.3 (quoted above).

*P.5 line 15: is that version of the GRACE data downscaled in any way, and if so, how? I thought the basic GRACE data was at coarse resolution (e.g., 500km).*

The GRACE gridded land product that we use is provided at a 1-degree resolution. We have clarified this by rewording the beginning of the Methods section as follows:

We obtained Level-3 TWSA data from GRACE using the University of Texas at Austin Center for Space Research (CSR) spherical harmonic solutions (Swenson, 2012). Global land data at a 1° resolution were scaled using the coefficients provided by Landerer and Swenson (2012).

5 P.5 line 17: "the TWS time series". I read that GRACE data are actually anomalies compared to the mean over 2004-2009. How is that accounted for in the computation of the metrics? Are the other variables centered on the same years? Does that affect results in any way? What about model outputs?

In the context of our metrics, the baseline against which GRACE Anomalies are compared is arbitrary. In our calculations, the baseline only affects the intercepts of the linear correlations; it does not affect the correlation coefficients that comprise our metrics.

*P.6 line 32: see main comment above: I am not sure this is the most relevant time of year to investigate, and they are issues of rainfall autocorrelations.*

**15**

We have addressed this in our response to the reviewer's main comment above, in which we explain why we chose to focus on this time of year.

P.7 lines 12-15: that is, if the feedback is actually a positive moisture feedback. In other words, the authors adopt the a
priori view that they are looking at a positive, moisture recycling feedback. This should be stated more explicitly, and perhaps earlier in the manuscript.

This point is well taken, and have modified the Methods section to more explicitly state this assumption by removing lines 12–15 and 19–20 on page 7, and replacing them with the following:

25

30

Here we note that our evaluation of both the forcing and response metrics will follow a nomenclature that considers strong coupling as acting in the direction of an overall positive feedback loop. In regions with a strong positive feedback, higher than average TWS would be followed by lower than average VPD, as more available water is able to fulfill evaporative demand. Therefore, strong TWS forcing on VPD would be associated with a negative correlation coefficient. Higher VPD during the drawdown interval would increase evaporative demand, potentially leading to a more negative TWS anomaly, therefore a strong response of the land surface to VPD would also be associated with a negative correlation coefficient.

Because the partitioning of surface fluxes can, depending on the spatiotemporal scale, cause a change of either sign to cloudiness and precipitation (Taylor et al., 2012; Guillod et al., 2015), correlation coefficients of either sign could indicate strong land surface forcing on PPT and SW $\downarrow$ . However, the response metrics would be expected to show greater consistency. Higher PPT during the drawdown interval would be expected to increase TWS (positive correlation), while higher SW $\downarrow$  would increase evaporative demand, thereby decreasing TWS (negative correlation). Therefore, to maintain consistent nomenclature based on evaluating the strength of a positive moisture feedback, we consider strong coupling in both the forcing and response metrics to be associated with a positive correlation in the case of PPT and a negative correlation in the case of SW $\downarrow$ .

10

15

5

P.8 line 18: what about AMIP simulations?

Correlations in our forcing metric come from both land–atmosphere coupling and the effects of remote SST forcing. AMIP simulations could reduce internal variability, but will not capture ocean-atmosphere interactions. We are interested in evaluating fully coupled models that are used for 21st century projections.

P.8 line 28: It's unclear to me why the authors restrict themselves to the GLACE- CMIP5 models. There is no further comparison in the manuscript, on a model-by-model basis, with results from that experiment. So why not use the whole CMIP5 ensemble?

20

The purpose of this manuscript is not to evaluate the entire CMIP5 ensemble, but rather to introduce a new approach toward model benchmarking using a small ensemble of models as an example.

P.9 line 9: so what? What is made of that? What are the implications for the correlation-based metrics? This comment
applies to the whole sub-section, actually, including the result about climate variability. If anything, higher variability in model outputs would point to lower correlations, if the covariance between TWS and climate is similar.

This point is well taken, and we have addressed this with an addition to this section that clarifies the implications of this analysis:

30

Comparing both the timing of TWS dynamics and the interannual variability of TWS and the atmospheric variables between the observations and model output provides context for interpreting the correlation-based metrics we present next. Although there are some inconsistencies, as noted above, the model largely reproduced the same patterns evident in the remote sensing data. In many regions, the interannual

variability in model output was similar to the observed variability, indicating that CESM was able to simulate reasonably well the baseline properties (timing and variability) that influence feedback dynamics.

P.9 line 11: aren't trends removed from this data? Please clarify.

5

Trends are removed only for the purpose of generating the annual climatology, as indicated in the methods section, are retained in the correlation analysis in order to capture decadal-scale variability that would represent a trend in the relatively short time series.

10 *P.9 line 26: still, why would the covariance be positive?*

We have addressed this question with the following addition to this section:

Positive correlations are unlikely to reflect direct land-atmosphere interactions. Instead, they demonstrate how remote SST forcing can, depending on lag times, lead to apparent negative relationships such as those demonstrated by Wei et al. (2008).

*P.9* whole section 3.2: this whole subsection feels very descriptive. On the other hand, there is not much description of the processes themselves. This might feel obvious to the authors, but some further discussion of what the correlations mean physically, when describing the figures, may be welcome.

The purpose of this section is to help the reader understand how the metrics work for a single ensemble member before we present the aggregated results for the whole ensemble. We feel this is a critical step in explaining how our metrics work and justifying their use in comparing models with observations.

25

15

20

P.10 line 13: the link with cloud cover and precipitation should be explicitly mentioned here.

The link is now explicitly mentioned:

30 The forcing metrics for SW $\downarrow$  showed a mixture of positive and negative correlations, indicating that higher *TWSAmax* was either positively or negatively coupled with shortwave radiation (Figure 7a). This finding is consistent with both positive and negative coupling between cloud cover and terrestrial moisture observed over shorter time scales (Taylor et al., 2012; Guillod et al., 2015).

We have addressed this in the response to the reviewer's comment #4 above.

**5 P.11 line 23: "Discussion".**

We thank the reviewer for pointing out this typographical error, which has been corrected.

P.11 line 28: as mentioned above, these ``well understood mechanisms" are actually never really explained.

**10**

Our revised Discussion section, described above in the response to the first general comment, removes this phrase and more clearly describes what is being shown.

P.12 lines 3-4: that's exaggerated. Feedback results on Figures 5-7 are very noisy, and even from a simply qualitative perspective, it is a stretch to say that they agree with results from GLACE 1. One could just as well point out all the regions 15 on Figures 5-7 that do NOT show up in GLACE 1 and say results are completely different. Besides, I find it a bizarre impulse (or maybe, a testament to the strength of the GLACE 1 study) that every land-atmosphere study seemingly feels the need to point out some level of agreement with GLACE results, even when, as is the case here, the match is very weak at best, and more importantly, when different data (observations versus models), processes and spatio-temporal scales are considered. Consider removing that comparison.

20

The reviewer's point is well taken, and we do not believe the comparison with GLACE 1 is a necessary component of our discussion. As part of the major revision of our Discussion section, as described above, we have removed this reference to GLACE.

**25**

P.12 line 16: see main comment #4 above.

We have addressed this in the response to main comment #4 above.

P.12 line 26: the authors could still look at this in models results, though. In fact, showing the link between TWS and ET, for 30 instance, would reinforce their results and the physical interpretation that they propose.

This is limited by the lack of a global remote sensing ET data set spanning the study period.

*P. 13, first paragraph: this is unclear. Do the author mean that that the models underestimate remote influences of SSTs, for instance, and thus appear to have too strong a coupling?*

Along with the other major revisions to our Discussion section, we have removed this passage from the text and replaced it with a clearer explanation, which we have quoted above in response to main comment #2.

P.13 lines 16-18: see main comment #3 above.

This is addressed in major revisions to the Discussion section, quoted above in the response to main comment #3.

10

15

5

P. 14 line 18: but here observations show positive coupling, too! Please clarify.

The observational metrics in this study include the effects of remote SST forcing as well as land-atmosphere interactions integrated across seasonal time scales. The negative soil moisture-precipitation coupling mechanism found from observations by Taylor et al. (2013) would tend to reduce the overall positive correlation. If parameterized convection prevents models from correctly capturing this mechanism, then the correlations may be overestimated. We have clarified this with the following addition to the Discussion section:

Taylor et al. (2013) similarly found parameterized convection in an RCM yielding a positive coupling in contrast to the negative coupling found in both observations and model runs with explicitly simulated convection. If negative coupling mechanisms are present in reality but absent from models, this could contribute to an overestimate of coupling metrics and underrepresentation of negative feedbacks in models.

P. 14 line 21: but reduced stomatal opening would be associated with reduced ET, too. Please clarify.

**25**

Here we are suggesting that if model parameterizations cause stomatal conductance to be overly sensitive to soil moisture, this would lead to unrealistically high model feedbacks. This is a separate explanation than the possibility that too much water is available for ET, which we describe in the previous paragraphs.

**30 *P. 14 lines 18-30: See main comment #2. There is a fundamental issue with the manuscript here.**

We have made major revisions to the manuscript, and particularly the Discussion section, as addressed above in the response to main comments #1 and #2. As we explained above, we do not feel that this represents a fundamental issue with our approach or our manuscript.

P.15 line 3: see main comment #3.

We have addressed this above in our response to comment #3.

5

P.15 lines 8-9. not really: Seneviratne etal. (2013) show that long-term soil moisture change leads to more warming, differently across models in the GLACE-CMIP5. That, in and of itself, could be considered an estimate of (long-term) soil moisture-atmosphere coupling in these models; but, in any case, there is no comparison to estimates of present-day coupling.

10

We agree that the linkage between our metrics and the results of GLACE-CMIP5 are too speculative. For this reason and for the sake of brevity, we have removed this portion of the Discussion section as part of our major revisions.

P.15 lines 11: No. Warmer air "holding" more water vapor and leading to more precipitation would lead to positive 15 temperature-precipitation correlations - not negative.

We thank the reviewer for pointing out this erroneous characterization of the results of Berg et al. (2015). The phrase "in which higher air temperatures can hold more precipitable water" was intended to read "cloud cover variability drives precipitation and temperature in opposite directions" However, as mentioned in the response to the previous comment, we have removed this portion of the text from our discussion.

20

P.15 line 13: "determined": not really. What Berg et al. (2015) show is that because of land-atmosphere interactions, the interannual negative temperature-precipitation relationship that they identify in present-day climate holds on longer time scales, including in the case of climate change. This may be interpreted as suggesting, as the authors say here, that models with too strong a coupling will then overestimate future warming; however, it is not directly shown by that study. Consider

rephrasing.

As mentioned in the response to the previous two comments, we have removed this portion of the text from our Discussion section.

30

25

P.16 line 10: see comment above on P.12

As mentioned in the response to the comment above, we have removed the assertion that our observed metrics reflect the patterns found in GLACE. In addition to the revisions to the Discussion section, mentioned above, we have also revised the Conclusions section to remove this portion.

**5 *P.16 line 11: "regions of strong RESPONSE metrics", I believe.**

We thank the reviewer for pointing out this typographical error, which has been corrected

P.16 line 14: the implication is bit too implicit here. Consider being more explicit.

10

We have made this statement more explicit with the following revision:

Modeled coupling metrics are generally found to be stronger than those observed in the satellite record. If this discrepancy is due to models overestimating the two-way feedback between the land surface and the atmosphere, this could lead to models incorrectly projecting future warming trends and climatic extremes (e.g., Hirschi et al., 2011; Seneviratne et al., 2013; Cheruy et al., 2014; Miralles et al., 2014).

Figure 1: nice figure that helps understand the study. The y-axis on a) refers to anomalies, I presume – see comment on GRACE values above.

20

15

Correct, as we mentioned in response to the comment above. We have clarified this by replacing "TWS" with "TWSA" in the caption to Figure 1.

As noted above, Figure 3 and 4 are nice, but not much is made of them in the analysis.

**25**

As described in response to the reviewer's comment above, we have included some additional text to expand upon and clarify the purpose of these figures. As quoted above, we feel these figures are important for demonstrating that LENS is able to capture the baseline properties of our analysis (timing and variability) before presenting the correlation coefficients.

30 Figure 5-7: I suggest the authors modify the color legend here. More color shades is not always better. It is actually not easy to see differences in color shades on a continuous bi-color palette like here, and for the reader things essentially end up being two colors, one positive (green) and one negative (red). It would actually be easier to have fewer shades, more clearly separated, and with perhaps several different colors as well.

After experimenting with several color and shading schemes, we determined that the spatial variability in Figures 5–7 are best illustrated using the employed color scheme combined with crosshatching to indicate statistically significant correlations.

5 Figure 8: I suggest showing the mean of the CESM distribution as well.

We considered including this, but determined that it made the figures too busy without adding useful information.

**References cited in reviewer responses:**

Berg, A., Lintner, B. R., Findell, K., Seneviratne, S. I., van den Hurk, B., Ducharne, A., Chéruy, F., Hagemann, S.,
Lawrence, D. M., Malyshev, S., Meier, A. and Gentine, P.: Interannual Coupling between Summertime Surface Temperature and Precipitation over Land: Processes and Implications for Climate Change, J. Clim., 28(3), 1308--1328, doi:10.1175/JCLI-D-14-00324.1, 2015.

Betts, A. K., Desjardins, R., Worth, D. and Beckage, B.: Climate coupling between temperature, humidity, precipitation, and
cloud cover over the Canadian Prairies, J. Geophys. Res. Atmos., 119(23), 13,305--13,326, doi:10.1002/2014JD022511, 2014.

Cheruy, F., Dufresne, J. L., Hourdin, F. and Ducharne, A.: Role of clouds and land-atmosphere coupling in midlatitude continental summer warm biases and climate change amplification in CMIP5 simulations, Geophys. Res. Lett., 41(18), 6493--6500, doi:10.1002/2014GL061145, 2014.

Dirmeyer, P. A., Koster, R. D. and Guo, Z.: Do global models properly represent the feedback between land and atmosphere?, J. Hydrometeorol., 7(6), 1177--1198, doi:10.1175/JHM532.1, 2006.

25 Ferguson, C. R., Wood, E. F. and Vinukollu, R. K.: A global intercomparison of modeled and observed land-atmosphere coupling, J. Hydrometeorol., 13(3), 749--784, doi:10.1175/JHM-D-11-0119.1, 2012.

Findell, K. L., Gentine, P., Lintner, B. R. and Kerr, C.: Probability of afternoon precipitation in eastern United States and Mexico enhanced by high evaporation, Nat. Geosci, 4(7), 434--439, doi:10.1038/ngeo1174c, 2011.

30

20

Findell, K. L., Gentine, P., Lintner, B. R. and Guillod, B. P.: Data length requirements for observational estimates of landatmosphere coupling strength, J. Hydrometeorol., 16(4), 1615--1635, doi: 10.1175/JHM-D-14-0131.1, 2015.

Flechtner, F., Morton, P., Watkins, M. and Webb, F.: Status of the GRACE Follow-On Mission, in Gravity, Geoid and Height Systems: Proceedings of the IAG Symposium, edited by U. Marti, pp. 117--121, Springer International Publishing, 2014.

5

Guillod, B. P., Orlowsky, B., Miralles, D., Teuling, A. J., Blanken, P. D., Buchmann, N., Ciais, P., Ek, M., Findell, K. L., Gentine, P., Lintner, B. R., Scott, R. L., den Hurk, B. and I. Seneviratne, S.: Land-surface controls on afternoon precipitation diagnosed from observational data: uncertainties and confounding factors, Atmos. Chem. Phys., 14(16), 8343--8367, doi:10.5194/acp-14-8343-2014, 2014.

10

Guillod, B. P., Orlowsky, B., Miralles, D. G., Teuling, A. J. and Seneviratne, S. I.: Reconciling spatial and temporal soil moisture effects on afternoon rainfall, Nat Commun, 6, doi:10.1038/ncomms7443, 2015.

Hirschi, M., Seneviratne, S. I., Alexandrov, V., Boberg, F., Boroneant, C., Christensen, O. B., Formayer, H., Orlowsky, B.
and Stepanek, P.: Observational evidence for soil-moisture impact on hot extremes in southeastern Europe, Nat. Geosci., 4(1), 17--21, doi:10.1038/ngeo1032, 2011.

Hung, M.-P., Lin, J.-L., Wang, W., Kim, D., Shinoda, T. and Weaver, S. J..: MJO and convectively coupled equatorial waves simulated by CMIP5 climate models, J. Clim., 26(17), 6185--6214, doi: 10.1175/JCLI-D-12-00541.1, 2013.

20

Keys, P. W., Wang-Erlandsson, L., and Gordon, L. J.: Revealing invisible water: Moisture recycling as an ecosystem service, PLoS ONE, 11(3), 1--16, doi:10.1371/journal.pone.0151993, 2016.

[revised manuscript text omitted]

**author 9/7/2016 1:58 PM**

**Deleted:** The metrics introduced in this study were specifically designed to take advantage of the monthly TWS anomaly (TWSA) anomaly product from the Gravity Recovery and Climate Experiment (GRACE) mission (Landerer and Swenson, 2012; Wahr et al., 2004).

that are available for and potentially limiting to ET. Under these conditions, month-to-month TWS anomalies capture portions of the terrestrial water cycle that soil moisture alone may not.

Previous studies have largely focused on land surface moisture availability as a forcing mechanism on the atmosphere, as

- 5 this relationship has important implications for seasonal predictability as well as the projection of the frequency and severity of climatic extremes. However, the land surface response to the atmosphere is governed by many of the same processes through which terrestrial moisture availability forces atmospheric conditions, and it determines the conditions that drive subsequent land surface forcing. It is therefore critical to assess the response of land surface moisture to atmospheric conditions, as an accurate representation of these processes is essential for generating the correct terrestrial moisture
- 10 variability that will go on to influence the atmosphere. As far as we can tell, this response limb of the land surface feedback loop has not been systematically integrated with existing analyses of land-atmosphere coupling strength.

Our globally applicable approach used the annual cycle of TWS drawdown and recharge to isolate the months of the year during which the land surface loses moisture, which we refer to as the drawdown interval (Figure 1a). We selected this interval because past work has shown that the land surface's influence on the atmosphere is most prevalent during summer in

15 interval because past work has shown that the land surface's influence on the atmosphere is most prevalent during summer in the northern hemisphere (Cheruy et al., 2014; Phillips and Klein, 2014) and during the dry season in tropical forests (Harper et al., 2013; Lorenz and Pitman, 2014). This approach allowed us to investigate land surface coupling at a global scale, and to extend metrics developed in previous work for pre-defined monthly intervals corresponding to boreal summer (e.g., Guo and Dirmever, 2013; Koster et al., 2006) to be applicable to any seasonality.

20

In our analysis, separate metrics were calculated to consider the influence of TWS at the onset of the drawdown interval on atmospheric conditions in subsequent months, and simultaneously, the influence of atmospheric conditions during the drawdown interval on terrestrial water storage at the end of the season. We refer to these two relationships as the forcing and response limbs, respectively, of the fully coupled feedback loop between the land surface and the atmosphere (Figure 1b).

We estimated the strength of these feedbacks during 2002-2014 using GRACE and other satellite remote sensing data (Table 1). We then used the satellite observations to evaluate the strength of these feedbacks in the Community Earth System Model (CESM) Large Ensemble (LENS) (Kay et al., 2014) and in several models that contributed simulations to CMIP5 (Table 2).

**2 Methods**

**2.1 Remote sensing data**

30

We obtained Level-3 TWSA data from GRACE using the University of Texas at Austin Center for Space Research (CSR) spherical harmonic solutions (Swenson, 2012). Global land data at a J° resolution were scaled using the coefficients provided by Landerer and Swenson (2012). The study period was limited to September 2002 through November 2014, in

34

author 9/7/2016 1:58 PM **Deleted:** However, it is also

author 9/7/2016 1:58 PM Deleted: uses author 9/7/2016 1:58 PM Moved (insertion) [1]

| 1  | author 9/7/2016 1:58 PM                                           |
|----|-------------------------------------------------------------------|
| (  | Moved (insertion) [2]                                             |
| /  | author 9/7/2016 1:58 PM                                           |
| /( | Deleted: full land surface-atmospheric                            |
| 1  | author 9/7/2016 1:58 PM                                           |
| 4  | Deleted: 2015                                                     |
|    | author 9/7/2016 1:58 PM                                           |
| A  | Deleted:                                                          |
|    | author 9/7/2016 1:58 PM                                           |
| /[ | Deleted: the Gravity Recovery and Climate Experiment (     |
|    | author 9/7/2016 1:58 PM                                           |
| /l | Deleted: ) platform                                               |
| 1  | author 9/7/2016 1:58 PM                                           |
| /\ | Deleted: GRACE data are available globally over                   |
| A  | author 9/7/2016 1:58 PM                                           |
| 4  | Deleted: monthly,                                                 |
| 1  | author 9/7/2016 1:58 PM                                           |
|    | Deleted: from September, 2002 through September, 2015, and |
| 1  | author 9/7/2016 1:58 PM                                           |
|    | Deleted: ), with                                                  |

order to minimize temporal gaps. GRACE data during the study period included eight non-consecutive and two consecutive missing months, which were filled using linear interpolation. At each grid cell, the TWSA time series was decomposed into linear trend, seasonal cycle, and interannual variability components using ordinary least squares regression. This decomposition allowed us to estimate a mean annual cycle at each grid cell with minimal influence of any long-term trend.

Level-3 near-surface temperature and relative humidity were obtained globally at a monthly, 1° resolution from the ascending (daytime) orbit of the Atmospheric Infrared Sounder (AIRS) platform (Susskind et al., 2014). Vapor pressure deficit (VPD) was calculated from the AIRS data using the August–Roche–Magnus approximation to the Clausius– Clapeyron relation (Lawrence, 2005). Precipitation (PPT) data were obtained from the Global Precipitation Climatology

Project (GPCP), a merged satellite and gauge-based data set (Huffman et al., 2009) at a daily, 1° resolution and then integrated monthly. Downwelling shortwave radiation (SW↓) was obtained globally at a monthly, 1° resolution from the Clouds and the Earth's Radiant Energy System (CERES) Energy Balanced And Filled (EBAF) Surface product (Loeb et al., 2009). More information describing the remote sensing and reanalysis data products used in our analysis is summarized in Table 1.

**15 **2.2 Drawdown interval**

5

As a first step, we used the mean annual cycle from GRACE to determine the months of the maximum and minimum TWS  $\int$ anomalies in order to define the drawdown interval at each 1° land grid cell (Figure 2), Northern hemisphere middle and high latitudes exhibited a drawdown interval beginning in the spring (MAM) and ending in the late summer or fall (ASO), reflecting the timing of the boreal summer growing season. At lower latitudes, the North American, African, and Asian

20 monsoons were evident, with Mexico, India, and the Sahel showing a drawdown interval beginning in September, after the monsoonal precipitation has peaked, and ending the following spring after the winter dry season. The onset of the drawdown interval reversed abruptly at the equator in Africa and Asia, with the drawdown interval reflecting a winter dry season in the austral low latitudes transitioning to a summer growing season in the austral midlatitudes. Within the months of our study period, the portion of land grid cells that experience 11, 12, and 13 complete drawdown intervals are 9.4%, 90.5%, and 0.1%
25 respectively.

**2.3 Coupling metrics,**

Existing literature generally defines "land-atmosphere coupling" as the extent to which atmospheric conditions are forced by the land surface state, and would use the term "atmosphere-land coupling" to refer to the land surface response to atmospheric drivers (Seneviratne et al., 2010). In this study, we develop what we refer to as "coupling metrics" to indicate

30 the strength of both limbs of the fully coupled land, atmosphere feedback loop. We use the terms "forcing" and "response" to indicate whether we are considering the forcing of the atmosphere by the land surface, or the response of the land surface to the atmosphere (Figure 1b),

35

**author 9/7/2016 1:58 PM**

| author 9/7/2016 1:58 PM                                                                                                                                                                                                                                                                                                                                                                                                                                                                    |
|--------------------------------------------------------------------------------------------------------------------------------------------------------------------------------------------------------------------------------------------------------------------------------------------------------------------------------------------------------------------------------------------------------------------------------------------------------------------------------------------|
| Deleted: this                                                                                                                                                                                                                                                                                                                                                                                                                                                                              |
| author 9/7/2016 1:58 PM                                                                                                                                                                                                                                                                                                                                                                                                                                                                    |
| Formatted: English (UK)                                                                                                                                                                                                                                                                                                                                                                                                                                                                    |
| author 9/7/2016 1:58 PM                                                                                                                                                                                                                                                                                                                                                                                                                                                                    |
| Deleted:                                                                                                                                                                                                                                                                                                                                                                                                                                                                                   |
| author 9/7/2016 1:58 PM                                                                                                                                                                                                                                                                                                                                                                                                                                                                    |
| Deleted: satellite data, we substituted data                                                                                                                                                                                                                                                                                                                                                                                                                                               |
| author 9/7/2016 1:58 PM                                                                                                                                                                                                                                                                                                                                                                                                                                                                    |
| Moved down [3]: We only used atmospheric
reanalysis data for this sensitivity analysis, as these
data benefit from assimilation of observations, while
we continued to use GRACE for TWSA. Comparing
results from this GRACE-reanalysis hybrid to those
using only satellite data provided                                                                                                                                                                                  |
| author 9/7/2016 1:58 PM                                                                                                                                                                                                                                                                                                                                                                                                                                                                    |
| Deleted: an indication of how sensitive our coupling metrics were tohe data source.                                                                                                                                                                                                                                                                                                                                                                                                 |
| author 9/7/2016 1:58 PM                                                                                                                                                                                                                                                                                                                                                                                                                                                                    |
| Moved up [1]: We selected this interval because
past work has shown that the land surface's
influence on the atmosphere is most prevalent during
summer in the northern hemisphere (Cheruy et al.,
2014; Phillips and Klein, 2014) and during the dry
season in tropical forests (Harper et al., 2013; Lorenz
and Pitman, 2014). This approach allowed us to
investigate land surface coupling at a global scale,
and to extend metrics developed in previous [[5] |
| author 9/7/2016 1:58 PM                                                                                                                                                                                                                                                                                                                                                                                                                                                                    |
| Deleted: 2006) for pre-defined monthly intervals
corresponding with boreal summer to be applicable
at any location regardless of seasonality.                                                                                                                                                                                                                                                                                                                                        |
| author 9/7/2016 1:58 PM                                                                                                                                                                                                                                                                                                                                                                                                                                                                    |
| Deleted:                                                                                                                                                                                                                                                                                                                                                                                                                                                                                   |
| author 9/7/2016 1:58 PM                                                                                                                                                                                                                                                                                                                                                                                                                                                                    |
| Formatted: English (UK)                                                                                                                                                                                                                                                                                                                                                                                                                                                                    |
| author 9/7/2016 1:58 PM                                                                                                                                                                                                                                                                                                                                                                                                                                                                    |
| Deleted: for forcing                                                                                                                                                                                                                                                                                                                                                                                                                                                                       |
| author 9/7/2016 1:58 PM
Moved (insertion) [4]                                                                                                                                                                                                                                                                                                                                                                                                                                           |

author 9/7/2016 1:58 PM

We defined our forcing metric as the Pearson product-moment correlation coefficient between the TWS anomaly at the onset of the drawdown interval ( $TWSA_{max}$ ) and the surface atmospheric conditions during the drawdown interval ( $ATM_{di}$ ). In our analysis, we selected 3 variables to represent the atmospheric state: VPD, SW $\downarrow$ , and PPT. These atmospheric variables were

averaged during the drawdown interval, including during the months of climatological maximum and minimum TWSA. We
 chose these variables because they represent various aspects of evaporative supply (PPT) and demand (VPD and SW↓).

Similarly, we defined our response metric as the correlation coefficient between atmospheric conditions during the drawdown interval  $(ATM_{di})$  and the land surface state at the end of the drawdown interval  $(TWSA_{min})$ . Although most previous diagnoses of land-atmosphere coupling has focused on the forcing limb, we argue the response limb is equally important as a metric for model evaluation. Specifically, if variability in the balance between evaporative supply and demand does not lead to the correct TWS variability, then the incorrect TWS response will feed back into subsequent forcing on the

atmosphere.

 We note that these metrics do not provide distinctive information for measuring the strength of land-atmosphere coupling or the land surface response. While the metrics include the influence of direct land-atmosphere interactions, they are also potentially influenced by atmospheric and soil moisture persistence, as well as remote forcing from sea surface temperatures (SSTs) (Orlowsky and Seneviratne, 2010; Mei and Wang, 2011). Nevertheless, these metrics may still serve as useful benchmarks against which to evaluate the ability to ESMs to reproduce the proper relationships based on the combination of these factors.

Here we note that our evaluation of both the forcing and response metrics will follow a nomenclature that considers strong coupling as acting in the direction of an overall positive feedback loop. In regions with a strong positive feedback, higher than average TWS would be followed by lower than average VPD, as more available water is able to fulfill evaporative

25 demand. Therefore, strong TWS forcing on VPD would be associated with a negative correlation coefficient. Higher VPD during the drawdown interval would increase evaporative demand, potentially leading to a more negative TWS anomaly, therefore a strong response of the land surface to VPD would also be associated with a negative correlation coefficient.

Because the partitioning of surface fluxes can, depending on the spatiotemporal scale, cause a change of either sign to
 cloudiness and precipitation (Taylor et al., 2012; Guillod et al., 2015), correlation coefficients of either sign could indicate strong land surface forcing on PPT and SW↓. However, the response metrics would be expected to show greater consistency. Higher PPT during the drawdown interval would be expected to increase TWS (positive correlation), while higher SW↓ would increase evaporative demand, thereby decreasing TWS (negative correlation). Therefore, to maintain consistent nomenclature based on evaluating the strength of a positive moisture feedback, we consider strong coupling in both the

36

**author 9/7/2016 1:58 PM**

**author 9/7/2016 1:58 PM**

**Deleted: Monthly GRACE, AIRS, GPCP, and CERES observations from September 2002 through September 2015 provided 12 or 13 complete drawdown intervals at each grid cell, depending on the months of TWSAmax and TWSAmin. In regions of strong land surface forcing of the atmosphere, higher than average TWS would typically be followed by lower than average VPD, lower than average SW $\downarrow$ , and higher than average PPT. Thus we would expect strongly coupled regions to have negative correlation between TWSAmax and VPDdi, negative correlation between TWSAmax and SW1di, and positive correlations between TWSAmar and PPTa author 9/7/2016 1:58 PM Deleted: to assess the impact of the atmosphere on the land surface state (the author 9/7/2016 1:58 PM Deleted: limb), we examined author 9/7/2016 1:58 PM Deleted: relationship author 9/7/2016 1:58 PM Deleted: variables author 9/7/2016 1:58 PM Deleted: Strong land surface response to the atmospheric conditions yielded negative correlation between VPDdi and TWSAmin, negative correlation between SW1 and TWSAmin, and positive correlation between PPTdi and TWSAmin author 9/7/2016 1:58 PM Deleted: ... [9]**

forcing and response metrics to be associated with a positive correlation in the case of PPT and a negative correlation in the case of SWL.

**24 Community Earth System Model Large Ensemble**

We used the metrics described above to evaluate feedback strength in the Community Earth System Model (CESM) Large Ensemble (LENS). LENS comprises an ensemble of 38 fully coupled runs in which air temperature initial conditions are perturbed slightly (by an amount less than round-off error) to reveal the internal variability inherent within the coupled climate model. LENS has demonstrated that the uncertainty in climate projections due to internal climate variability inherent in CESM is comparable to the ranges of output within the entire CMIP5 experiment (Kay et al., 2014). LENS uses version 1 of CESM (CESM1) with version 5 of the Community Atmosphere Model (CAM5) and version 4 of the Community Land

10 Model (CLM4) at a horizontal resolution of 1°. The ensemble run follows protocols from the CMIP5 experiment, with historical radiative forcing for the 20th century and representative concentration pathway 8.5 (RCP8.5) forcing for the 21st century.

The LENS data were chosen as a starting point for feedback evaluation for two reasons. First, the availability of a TWS variable in these simulations enabled a direct comparison with metrics derived using data from GRACE. The TWS field in

CLM4 included water from surface and canopy storage, snow and ice, soil moisture, and a dynamic aquifer, in addition to river water storage terms from the coupled River Transport Module (RTM). The coupling of CLM4 with RTM has been shown to be important for simulating both the annual cycle and interannual variability of TWS in comparison with GRACE (Kim et al., 2009).

20

25

15

Second, the ensemble allowed us to test the importance of internal model variability for the diagnosis of feedback strength. Because the complete satellite record was relatively short (containing no more than 12 drawdown intervals at any location), comparison with an equivalent single time series of model output could be influenced by a model's internal decadal-scale variability (Kay et al., 2014). Analyzing the full ensemble from LENS enabled us to assess the sensitivity of our forcing and response metrics to this variability. We extracted from each ensemble member the equivalent months of the satellite record (September, 2002 through November 2014), with data prior to December, 2005 coming from the historical runs, and data

**2.5 Assessment of uncertainty**

from January 2006 onward coming from the RCP8.5 simulations.

30 To assess the sensitivity of our metrics to observational uncertainty, we used a Monte Carlo sampling approach. For each of the 38 members of LENS, we calculated coupling metrics ten times with random noise added to both TWSA and atmospheric variable time series at each grid cell. The noise was randomly generated from a Gaussian distribution with a

37

**author 9/7/2016 1:58 PM**

author 9/7/2016 1:58 PM Deleted: coupling...eedback strength in th...[11]

author 9/7/2016 1:58 PM

author 9/7/2016 1:58 PM Moved (insertion) [5]

author 9/7/2016 1:58 PM Deleted: is author 9/7/2016 1:58 PM Moved up [5]: important for simulating both the

annual cycle and interannual variability of TWS in comparison with GRACE (Kim et al., 2009). . author 9/7/2016 1:58 PM **Deleted:** . author 9/7/2016 1:58 PM

mean of zero and a standard deviation equal to 25% of the standard deviation of the original data. Comparing these results with those from the unaltered data provided some indication of how much our coupling metrics are degraded by random noise as an approximation of observational uncertainty.

5 In addition, to assess how our analysis may be influenced by uncertainty due to the selection of satellite data, we substituted data from the European Centre For Medium-range Weather Forecasting (ECMWF) Interim Reanalysis (ERA-Interim) (Dee et al., 2011) in place of AIRS, GPCP, and CERES-derived variables, We only used atmospheric reanalysis data for this sensitivity analysis, as these data benefit from assimilation of observations, while we continued to use GRACE for TWSA. Comparing results from this GRACE-reanalysis hybrid to those using only satellite data provided a general indication of how sensitive our coupling metrics were to the data source.

**2.6 CMIP5 analysis**

To extend our analysis to models that did not output an explicit TWS field, we compared accumulated residuals of precipitation, evapotranspiration, and total runoff (surface and subsurface) with the explicit TWS variable in the LENS simulations. We also compared coupling metrics calculated from LENS using accumulated residuals with those calculated from the explicit TWS field. After we determined that the accumulated residuals of the water balance represented much of

from the explicit TWS field. After we determined that the accumulated residuals of the water balance represented much of the variability in the explicit TWS variable and yielded coupling metrics with similar distributions within LENS, we calculated equivalent metrics for several model simulations in the CMIP5 archive (Table 2). We selected the CMIP5 models that were similar to LENS (CESM1-CAM5 and CESM1-BGC) as well as the models that participated in the GLACE–
 CMIP5 experiment (Seneviratne et al., 2013) for which each necessary output field was available (CCSM4, GFDL-ESM2M, 20
 GFDL-ESM2G, IPSL-CM5A-MR, and IPSL-CM5A-LR).

**3 Results**

**3.1 Drawdown interval and interannual variability**

A comparison of the months of maximum and minimum terrestrial water storage as determined by climatologies of GRACE and the LENS ensemble mean indicated that the model largely reproduces the timing of TWSA seasonality evident in the

- 25 satellite observations (Figure 2). Geographic patterns of seasonality were consistent between the model and observations, though a phase shift in the drawdown interval is apparent in eastern Canada and central Eurasia where LENS had a one-month early bias for both the maximum and minimum TWSA, in southeast North America where the onset of the modeled drawdown interval was slightly later than the observations, and in parts of east Asia and Australia where the modeled drawdown interval ended earlier than in the observations. However, despite capturing generally correct timing, the model
- 30 exhibited higher interannual variability of  $TWSA_{max}$  and  $TWSA_{min}$  across the 11-12 drawdown intervals compared with the satellite data (Figure 3) particularly in the southern United States, southern South America, central and eastern Africa,

38

**author 9/7/2016 1:58 PM Moved (insertion) [6]**

author 9/7/2016 1:58 PM Moved (insertion) [3]

author 9/7/2016 1:58 PM Deleted:

**author 9/7/2016 1:58 PM Deleted: do author 9/7/2016 1:58 PM Deleted: the author 9/7/2016 1:58 PM Deleted: were compared with TWS in LENS (supplementary figures).**

| author 9/7/2016 1:58 PM |
|-------------------------|
| Deleted: all            |
| author 9/7/2016 1:58 PM |
| Deleted: fields were    |
| author 9/7/2016 1:58 PM |
| Deleted:                |

| 1 | author 9/7/2016 1:58 PM |  |  |  |  |
|---|-------------------------|--|--|--|--|
|   | Deleted: phase          |  |  |  |  |
| ١ | author 9/7/2016 1:58 PM |  |  |  |  |
|   | Deleted: and            |  |  |  |  |
| 1 | author 9/7/2016 1:58 PM |  |  |  |  |
|   | Deleted: -13            |  |  |  |  |

southern Asia, and eastern Australia. One possible explanation for this is the presence of multi-year trends in aquifer storage in CLM that are not consistent with GRACE (Swenson and Lawrence, 2015).

A comparison of the interannual variability of atmospheric variables across multiple drawdown intervals between the model

5 and satellite data showed various degrees of consistency (Figure 4). The magnitude and geographic pattern of VPDdi was generally consistent, though LENS showed greater interannual variability than AIRS in central and western North America, South America, northern and southern Africa, and southern Asia, In the case of PPTdi, LENS showed less interannual variability than GPCP in Southeast North America and much of South America, but the two were largely consistent elsewhere.  $SW_{\downarrow di}$  was the least consistent between the model and satellite data, as LENS showed greater interannual variability than CERES in southern North America, northern Eurasia, most of Africa, and most of Australasia,

10

Comparing both the timing of TWS dynamics and the interannual variability of TWS and the atmospheric variables between the observations and model output provides context for interpreting the correlation-based metrics we present next. Although there are some inconsistencies, as noted above, the model largely reproduced the same patterns evident in the remote sensing

data. In many regions, the interannual variability in model output was similar to the observed variability, indicating that 15 CESM was able to simulate reasonably well the baseline properties (timing and variability) that influence feedback dynamics.

**3.2 Evaluating feedbacks for a single model simulation**

coupling strength across many regions in temperate Asia.

The forcing metric for VPD derived from GRACE and AIRS showed regions of strong coupling, in which TWSAmax was 20 negatively correlated with VPDdi, in the northern Great Plains, northern South America, southern Africa, southern and western India, north central Eurasia, and northern Australia (Figure 5a). Regions with strong positive correlation were much less common, and were largely confined to areas of very low GRACE-derived TWSAmax variability (Figure 3a). Positive correlations are unlikely to reflect direct land-atmosphere coupling. Instead, they demonstrate how remote SST forcing can, depending on persistence and time delays with atmospheric responses, lead to apparent negative relationships such as those 25 demonstrated by Wei et al. (2008). In comparison with the satellite data, the VPD forcing metrics from the first ensemble member of LENS (Figure 5c) showed much stronger coupling in the southern and eastern Amazon, and marginally stronger

30

author 9/7/2016 1:58 PM

LENS showed widespread negative correlations, and did not show the positive correlations found in the satellite data. 39

The response metrics for VPD showed much stronger coupling than the forcing metric in both the satellite data and the

model (Figure 5b,d). Satellite data yielded negative correlation coefficients nearly everywhere, with positive correlations

found only in arid regions of low TWS variability. Particularly strong response metrics were found in eastern North

America, northern South America, western Eurasia, the Sahel, India, and eastern Australia. The first ensemble member from

author 9/7/2016 1:58 PM Deleted: compared with AIRS

author 9/7/2016 1:58 PM Deleted: compared with CERES author 9/7/2016 1:58 PM Formatted: English (US)

Response coupling in LENS was much more spatially homogeneous than in the satellite data, though northern South America and western Eurasia still showed stronger coupling than elsewhere.

Many of the areas that showed a strong forcing metric for VPD also showed a relatively strong forcing metric for PPT,

- 5 though the PPT forcing metric was overall weaker than that for VPD (Figure 6a). The response metric for PPT was generally positive, indicating that for much of the globe, a more positive TWSAmin was associated with higher precipitation rates (Figure 6b). Both the forcing and response metrics were somewhat stronger in the LENS member relative to those evident in the satellite data (Figure 6c,d).
- 10 The forcing metrics for SW $\downarrow$  showed a mixture of positive and negative correlations, indicating that higher TWSAmax was either positively or negatively coupled with shortwave radiation (Figure 7a). This finding is consistent with both positive and negative coupling between cloud cover and terrestrial moisture observed over shorter time scales (Taylor et al., 2012; Guillod et al., 2015). The response metrics for SW were generally negative, indicating that greater seasonal shortwave radiation was associated with more negative TWSAmin (stronger coupling), with West Africa being a notable exception (Figure 7b). The LENS member showed generally stronger coupling in both the forcing and response metrics for SWL 15
- (Figure 7c,d)

**3.3 Evaluating the CESM Large Ensemble**

In temperate and tropical regions, forcing metrics were generally stronger in LENS (more positive correlations for PPT, more negative for VPD and SWJ) than in the satellite and reanalysis data, indicating a stronger land surface forcing of the

20 surface atmospheric state in the model than in the observations (Figure 8). In boreal regions, forcing metrics were much weaker (closer to zero) than at lower latitudes in both the satellite data and in LENS, indicating very little relationship between TWSAmax and ATMdi. This is consistent with high levels of climate variability in many high latitude regions driven by the Arctic Oscillation, the North Atlantic Oscillation, and other dynamical modes (Cohen and Barlow, 2005). Furthermore, at high latitudes, ET is generally energy limited rather than moisture limited, which would lead to weak forcing 25 metrics as moisture availability would not strongly influence atmospheric conditions.

Response metrics were also generally higher in LENS than in both the satellite and reanalysis data (Figure 9). Noticeable exceptions were the VPD and PPT response metrics in the tropics, which were close to the satellite observations, and the boreal SW4 and tropical PPT response metrics, which were close to the reanalysis estimates. Despite the internal variability

30

evident within the model ensemble, and the difference between metrics as measured by the satellite data compared with the reanalysis data, the general pattern indicated that modeled response metrics were higher than those from observations and reanalysis.

40

author 9/7/2016 1:58 PM Deleted: were associated with larger TWSAmin

author 9/7/2016 1:58 PM Deleted: could be

**author 9/7/2016 1:58 PM**

Deleted: The internal variability in LENS yielded a distribution of forcing (Figure 8) and response (Figure 9) metrics with a spread on the same order of magnitude as the difference between modeled and satellite-derived zonal averages author 9/7/2016 1:58 PM

author 9/7/2016 1:58 PM Deleted: NAO

**author 9/7/2016 1:58 PM Deleted: author 9/7/2016 1:58 PM**

author 9/7/2016 1:58 PM Deleted: indicates

**3.4 Analysis of uncertainty**

The internal variability across the ensemble of simulations in LENS yielded a distribution of forcing and response metrics with a spread on the same order of magnitude as the difference between modeled and satellite-derived zonal averages. The distribution of coupling metrics from LENS revealed the sensitivity of the relationships to decadal climate variability given the relatively short TWS time series. Comparing this distribution with the spread between the purely satellite-derived metrics and GRACE-reanalysis hybrid indicated the sensitivity of our metrics to the choice of data source. The differences between satellite and reanalysis metrics were generally higher in the tropics, particularly for VPD and SW↓, and in mid-latitude VPD for both forcing and response variables. Elsewhere, the differences were generally similar to or less than the differences between the observationally constrained zonal averages and the LENS distributions.

10

5

Comparing the original LENS forcing and response metrics with those calculated after adding random noise to LENS (Figures S1 and S2) provided an estimate of the metrics' sensitivity to observational uncertainty. Adding random noise with 25% of the standard deviation of the original data to the model time series of TWSA and atmospheric variables at each grid cell does degrade the metrics slightly, causing areal averages to be closer to zero, but the differences are relatively small

15 compared to the differences between observed and modeled averages as well as the spread of the ensemble itself. This sensitivity analysis provided evidence that observational errors likely have a relatively small impact on the quality of our satellite-derived metrics.

**3.5 Evaluating CMIP5 models**

Comparison of the explicit TWS field from LENS with the accumulating residuals of the surface water budget (Figure S3), as well as the forcing and response metrics calculated using both (Figures S4 and S5), indicated that the alternative formulation provides an acceptable substitute when an explicit TWS field is not available from an ESM, More specifically, it suggests that water storage in rivers, lakes, and other parts of the terrestrial hydrologic system that are downstream from grid cell-level runoff did not significantly degrade the set of metrics evaluated here. Some degradation of the forcing metrics for PPT was apparent in the middle and low latitudes, but the remaining metrics are not highly sensitive to TWS formulation.

25 This suggests that metrics calculated for CMIP5 output using accumulating residuals could be reasonably and effectively compared with the metrics derived from LENS and the observations (Figure 10),

As with LENS, the metrics derived from CMIP5 output indicated generally stronger coupling metrics than the observations for both the forcing and response limbs. Exceptions include the VPD response metric in the tropics, the boreal PPT and SW↓
 forcing metrics, and the midlatitude SW↓ response metrics. The spread between various models was generally greater than the spread within any single model with a multi-member ensemble. Of the four models that use CLM4 for the land surface, the two that use CAM5 for the atmosphere (LENS and CESM1-CAM5) were clustered close together, and exhibited

41

author 9/7/2016 1:58 PM Deleted: forcing and response metrics calculated using the author 9/7/2016 1:58 PM Deleted: equivalent metrics calculated using the author 9/7/2016 1:58 PM Deleted: precipitation - ET - runoff) author 9/7/2016 1:58 PM Deleted: output of author 9/7/2016 1:58 PM Deleted: (see supplementary figures). author 9/7/2016 1:58 PM Deleted: can author 9/7/2016 1:58 PM Deleted: author 9/7/2016 1:58 PM Formatted: English (US) author 9/7/2016 1:58 PM Moved (insertion) [7] author 9/7/2016 1:58 PM Deleted: 3.4 CMIP5 models

generally the strongest forcing and response metrics. The two that use CAM4 (CCSM4 and CESM1-BGC) were close to each other, but with lower metrics in both forcing and response than the CAM5 models. The two GFDL models were both within the general ensemble range in the metrics for both VPD and PPT, but GFDL-ESM2M was an extreme outlier in both forcing and response metrics for SW $\downarrow$ .

5

15

Comparison of CMIP5 and LENS models indicated a mostly positive relationship between forcing and response metrics in temperate and tropical latitude bands. In boreal latitudes, there was little distinction between the forcing metrics of the different models, all of which were close to zero, though there were some clear differences within the response metrics. In temperate and tropical latitudes, models, that showed the strongest forcing metrics generally also showed the strongest response metrics for a given variable. This relationship suggests that analysis of the response limb of the feedback loop is

10 response metrics for a given variable. This relationship suggests that analysis of the response limb of the feedba important for understanding how conditions are set up for subsequent forcing via land-atmosphere coupling.

**4** Discussion**

**4.1 Benchmarking models with observed coupling metrics**

The metrics developed here from satellite observations provide a means for evaluating land-atmosphere feedback strength on seasonal to interannual timescales in coupled ESMs. The use of correlation coefficients in this study does not enable a direct assessment of whether the relationships are directly causal, as correlation between atmospheric and terrestrial conditions could result from atmospheric persistence and remote forcing from SSTs (Orlowsky and Seneviratne, 2010; Mei and Wang, 2011). Nonetheless, the satellite-derived metrics provide a meaningful constraint against which coupled models can be benchmarked, as these models need to correctly represent the combined effects of persistence, remote SST forcing,

**20 and land-atmosphere coupling.**

The forcing metrics, by indicating the relationship between antecedent TWS and subsequent atmospheric characteristics, provide observational constraints to complement previous research in largegscale land–atmosphere coupling in global models (e.g., Guo and Dirmeyer, 2013; Koster et al., 2006; Seneviratne et al., 2013). Observed forcing metrics were found to be

- 25 strong in some of the regions of intermediate wetness in which ET is limited by terrestrial moisture availability, in addition to some regions in the moist tropics in which ET is generally considered to be energy-limited. Recent observational analyses by (Hilker et al., 2014) demonstrate that at least in the Amazon, deep rooting zone water supplies can become seasonally depleted, leading to a stronger land-atmosphere coupling. This is consistent with findings that deep rooted plants vertically redistribute soil water to shallower layers, allowing higher levels of evapotranspiration to be sustained during the dry season
- 30 (Lee et al., 2005). It is also consistent with recent work demonstrating that TWSAs can be used as predictors for fire season severity in the Amazon (Chen et al., 2013).

**42**

**author 9/7/2016 1:58 PM**

**author 9/7/2016 1:58 PM**

Moved up [7]: Exceptions include the VPD response metric in the tropics, the boreal PPT and SW1 forcing metrics, and the midlatitude SW1 response metrics. The spread between various models was generally greater than the spread within any single model with a multi-member ensemble author 9/7/2016 1:58 PM

Deleted: , though the NCAR models (CCSM//CESM) were all relatively close to one another. The GFDL-ESM2M model is an exception, as both the forcing and response metrics of SW↓ were far from the remainder of the models. ....[16]

**author 9/7/2016 1:58 PM**

The inclusion of response metrics in our analysis allows the full feedback loop to be considered by recognizing the two-way dependence between the land surface and the atmosphere. The generally higher correlation coefficients in observed response metrics indicates the importance of the land surface response in priming the system for subsequent forcing on the atmosphere, For example, if the TWS response is too strongly coupled to the atmosphere, a small change in atmospheric

5 conditions could yield an unrealistically large change in TWS. The unrealistically large TWS anomaly, in turn, would have the potential to impart a larger land surface forcing of the atmosphere in subsequent time steps. That models and ensemble members with high forcing metrics were also generally found to have high response metrics (Figure 10) highlights the need to consider this.

10 Both the forcing and response,

> metrics as calculated from the output of the ESMs analyzed in the current study indicated generally stronger coupling compared with those derived from the satellite observations. There are exceptions to this pattern, but it holds generally true, particularly across middle and lower latitudes, and particularly in the LENS data. This is consistent with previous studies conducted at finer temporal resolutions (Ferguson et al., 2012) and across more limited spatial domains (Hirschi et al., 2011).

As described below, there are several possible explanations as to why models may simulate a stronger feedback than is 15 observed in the satellite record\_

**4.2 Possible explanations for enhanced feedback strength in models**

One set of possible explanations for the stronger coupling metrics in models relative to observations involves models overestimating the amount of water available for ET, during the drawdown interval. The land surface influence on the

- 20 atmosphere, requires water to be a limiting factor to ET but not limiting enough to prevent it altogether. Under more moisture-limited conditions, a drawdown interval may experience multiple shorter time periods during which ET is inhibited due to insufficient water, and the terrestrial moisture state exerts no control over flux partitioning. These periods of insufficient moisture would tend to reduce the overall feedback strength integrated across the duration of the drawdown interval. Model shortcomings that make water too readily available for ET could reduce the amount of time spent in a
- 25 periods of insufficient moisture during the drawdown interval, thereby unrealistically strengthening the longer-term feedback. We note that the opposite could take place under near-saturated conditions if a model overestimates the amount of time in which ET is energy-limited, but we would not expect these conditions to be as prevalent during the drawdown interval that was the time period of focus in our analysis.
- 30 ESMs are known to simulate unrealistically homogeneous rainfall intensity, with overestimates of drizzle and underestimates of large infrequent events (Dai, 2006). Infrequent high-intensity rainfall events would yield much more runoff from saturated soil, which would lead to a weaker connection between the land and atmosphere than frequent low-intensity drizzle. If a model simulates too much drizzle, precipitation could lead to too much storage, which would cause a model to overestimate

43

**author 9/7/2016 1:58 PM Deleted: the**

**author 9/7/2016 1:58 PM**

Deleted: , and the need to consider the full feedback loop. Assessment of modeled vs. observed coupling metrics elucidates whether landatmosphere feedbacks, which are known to significantly impact long-term climate projections (Berg et al., 2015; May et al., 2015), are of a realistic strength and sign.

**author 9/7/2016 1:58 PM**

Deleted: 4.2 Modeled feedback metrics [19] author 9/7/2016 1:58 PM Moved up [6]: In addition author 9/7/2016 1:58 PM Deleted: models must capture the correct response of TWS to the supply (precipitation) and demand .. [201 (ET) of the atmosphere. Incomplete or incor author 9/7/2016 1:58 PM Deleted: Each of these explanations repre...[21] author 9/7/2016 1:58 PM Deleted: . [22] author 9/7/2016 1:58 PM Deleted: , either by improperly partitionin ... [23] author 9/7/2016 1:58 PM Deleted: storage). If incoming precipitatio ... [24] author 9/7/2016 1:58 PM Deleted: . Such a model would then overe ... [25] author 9/7/2016 1:58 PM Deleted: high bias for author 9/7/2016 1:58 PM Deleted: . Similarly, any model author 9/7/2016 1:58 PM Deleted: cause water to move between sto ... [26] author 9/7/2016 1:58 PM Deleted: will cause the model to simulate . [27] author 9/7/2016 1.58 PM Deleted: these issues, described below, in ... [28] author 9/7/2016 1:58 PM Deleted: precipitation intensity and ... [29] author 9/7/2016 1:58 PM Deleted: bare soil fraction and processes author 9/7/2016 1:58 PM Formatted: Tabs: 3.99", Left

the response metrics. Too much storage also could allow water to be too readily available for ET, causing an overestimate of the forcing metrics. Contributions from drizzle could be offset if insufficient rainfall intensity does not allow high enough throughfall or soil moisture recharge. The issue of rainfall intensity is related to issues of convective parameterization (described below), and may be addressed in future versions of ESMs through atmospheric superparameterization, in which a

5 model's convective parameterization is replaced with embedded cloud resolving models (Kooperman et al., 2016).

A misrepresentation of either the amount of bare soil or of bare soil processes also could lead to overestimates of the amount of water available for ET and thereby coupling strength. Current land surface schemes of ESMs are based on the "big leaf" model paradigm, which could lead to overestimates of ET if runoff and groundwater recharge are underestimated as a consequence of too small of a bare soil fraction. In addition, even if bare soil fraction were correct, overestimates of ET due

10 consequence of too small of a bare soil fraction. In addition, even if bare soil fraction were correct, overestimates of ET due to an incomplete representation of surface resistance of bare soil, as found in CLM by Swenson and Lawrence (2014), would amplify positive feedbacks.

Additional explanations for why models may overestimate feedback strength include the parameterization of convection in 15 the PBL or stomatal conductance responses to soil moisture. Previous work using a regional climate model (RCM) with a higher spatial resolution have determined that convective parameterizations are as important as spatial resolution in the

simulation of precipitation coupling (Hohenegger et al., 2009). Taylor et al. (2013) similarly found parameterized convection in an RCM yielding a positive coupling in contrast to the negative coupling found in both observations and model runs with explicitly simulated convection, If negative coupling, mechanisms are present in reality but absent from models, this could

20 contribute to an overestimate of coupling metricsand underrepresentation of negative feedbacks in models. Similarly, the diversity of stomatal conductance parameterizations in CMIP5 ESMs is relatively low (Medlyn et al., 2011; Swann et al., 2016, in press), and if stomatal apertures close too rapidly in response to an initial deficit in terrestrial water storage, transpiration–humidity feedbacks may be intensifiedin an unrealistic manner.

- 25 One factor that could contribute toward stronger coupling metrics in models relative to observations is the effect of observational uncertainty combined with a relatively short time series. Adding random error to one or more variables in a correlation analysis will reduce the correlation coefficient, and this degradation has been shown to be sensitive to the length of data sets used to establish metrics of land-atmosphere coupling (Findell et al., 2015). Given the relatively short time series available for the current analysis, the correlation coefficients derived from remote sensing data may be reduced due to
- 30 observational uncertainty, unlike those derived from internally consistent models. We found that adding random noise to LENS at 25% of the standard deviation of the original data caused some degradation of our area-averaged coupling metrics, but only by a small amount relative to the difference between LENS and the observations (Figures S1 and S2). We chose 25% as a qualitative upper bound on likely uncertainties introduced from random observational error within the TWSA and atmospheric variable time series. This highlights the need for developing more quantitative error estimates in remote sensing

44

author 9/7/2016 1:58 PM Deleted:

author 9/7/2016 1:58 PM Deleted: and runoff author 9/7/2016 1:58 PM Deleted: underestimates author 9/7/2016 1:58 PM Deleted: is

author 9/7/2016 1:58 PM Deleted: , both of which yielded author 9/7/2016 1:58 PM Deleted: . It is therefore possible that the convective parameterizations of ESMs author 9/7/2016 1:58 PM Deleted: leading author 9/7/2016 1.58 PM Deleted: feedback author 9/7/2016 1:58 PM Deleted: 2011 author 9/7/2016 1:58 PM Deleted: artificially author 9/7/2016 1:58 PM Deleted: Finally, if the model underestimates the strength of remote forcing from SST anomalies or dynamic atmospheric processes, the strength of the coupling metrics will be overestimated. Orlowsky and Seneviratne (2010) urge caution when inferring land-atmosphere coupling relationships based on

and reanalysis products. More generally, this sensitivity analysis suggests that our coupling metrics, when averaged across large areas, may be useful in identifying robust data-model differences.

Another possible explanation stems from the fact that our coupling metrics include covariability due to atmospheric

- 5 persistence and remote forcing by SST (Orlowsky and Seneviratne, 2010; Mei and Wang, 2011) alongside the direct influence of land-atmosphere interactions. For this reason, we caution that overestimates of coupling metrics do not imply that the land-atmosphere feedback is necessarily stronger, but could be due to an overestimate of SST-driven correlations between the land surface and the atmosphere. Wei et al. (2008) demonstrated that negative correlations between soil moisture and subsequent precipitation can be explained by precipitation persistence combined with negative temporal
- 10 autocorrelation of precipitation associated with intra-seasonal modes such as the Madden-Julian Oscillation (MJO). Poor representation of the MJO period in CMIP5 models leads to unrealistic patterns of precipitation persistence (Hung et al, 2013). If models are failing to capture MJO-driven negative correlations, this could lead to overly strong positive correlations relative to observations. However, this would depend on the length of the drawdown interval relative to persistence time and the period of intra-seasonal modes.

**4.3** Uncertainties and future applications 15**

The current study demonstrates the utility of the coupling metrics presented here, but conclusions are limited by the time span of the satellite record. While LENS enables the internal variability of these relationships to be investigated within the model, it is unclear how much natural climate variability affects these relationships in reality on timescales longer than the satellite record. Furthermore, we acknowledge that observational error over an insufficiently long time series could reduce

- 20 the apparent strength of correlations (Findell et al., 2015). Therefore, the utility of the coupling metrics we present will increase alongside the length of the time series available from remote sensing platforms. This emphasizes the importance of the GRACE follow-on mission (Flechtner et al., 2014) and the need for continuity in the record between missions.
- Furthermore, incorporating additional remote sensing products can reduce uncertainties inherent in the satellite-derived data 25 sets. We presented metrics derived using ERA-Interim in place of AIRS, GPCP, and CERES in order to qualitatively illustrate this uncertainty. We found a non-negligible amount of uncertainty in both forcing and response metrics due to inconsistencies between the remote sensing and reanalysis products. Future work will address these uncertainties by incorporating additional observations and observationally constrained data sets such as those from the Global Soil Wetness Project (Dirmeyer et al., 2006b) and the Global Land Data Assimilation System, (Rodell et al., 2004). In addition, as 30 increasingly long time series of data become available from the Soil Moisture-Ocean Salinity (Mecklenburg et al., 2012) and Soil Moisture Active Passive (Panciera et al., 2014) missions, the metrics developed here can be applied to those data sets as well, which will elucidate the importance of surface soil moisture relative to the total TWS column in these interactions.

45

**author 9/7/2016 1:58 PM Deleted: feedback**

| 1 | author 9/7/2016 1:58 PM |
|---|-------------------------|
|   | Deleted: 13-year        |
| Υ | author 9/7/2016 1:58 PM |
|   | Deleted: The            |
| Υ | author 9/7/2016 1:58 PM |
|   | Deleted: feedback       |
|   |                         |

author 9/7/2016 1:58 PM Deleted: Energy and Water Experiment, the Global Precipitation Climatology Centre. author 9/7/2016 1:58 PM Deleted: author 9/7/2016 1:58 PM Deleted:

Finally, the issue of causality and the possibility that correlations result primarily from atmospheric persistence and remote forcing from SST rather than land-atmosphere interactions may be addressed using sensitivity experiments similar to those of GLACE and GLACE-CMIP. While the previous experiments have tested the importance of soil moisture interaction with the atmosphere, additional experiments could expand upon these methods by treating SST variability similar to terrestrial

5 soil moisture availability. Such experiments could determine the relative importance of remote SST forcing, including the effect of atmospheric persistence, and local land-atmosphere coupling in explaining correlations between TWS and atmospheric conditions.

As these sources of uncertainty are diminished, the coupling metrics introduced here may be used to assess whether improvements to model biogeophysics and parameterizations yield relationships that are more consistent with observations. CMIP5 models are known to have a high ET bias (Mueller and Seneviratne, 2014), which could be due in part to the explanations proposed as possible reasons for overestimated coupling metrics in models. As data become available from Phase 6 of the Coupled Model Intercomparison Project (CMIP6), these metrics could provide an assessment of whether improvements to ET processes in models also improves the relationship between the land surface and the atmosphere.

**15 5 Conclusion**

We have developed a new approach for measuring the strength of the two-way feedback relationships between TWS and the atmosphere. This approach was designed specifically to take advantage of TWSA data from the GRACE mission, along with concurrently collected remote sensing and reanalysis data sets of atmospheric variables, in a manner that could then be applied to earth system models. The coupling metrics described here quantify the relationships between both antecedent

20 TWS and subsequent atmospheric conditions, as well as antecedent atmospheric conditions and subsequent TWS.

Regions of strong forcing, in which the TWSA at the beginning of the drawdown interval was related to the subsequent atmospheric state, coincided with the semi-arid zones previously found to be hot spots of land atmosphere coupling, as well as some new tropical zones that may have moisture limited ET regimes. Regions of strong response metrics, in which the

25 TWSA at the end of the drawdown interval is related to the atmosphere, are much more widespread. Modeled coupling metrics are generally found to be stronger than those observed in the satellite record, 1f this discrepancy is due to models overestimating the two-way feedback between the land surface and the atmosphere, this could lead to models incorrectly projecting future warming trends and climatic extremes(e.g., Hirschi et al., 2011; Seneviratne et al., 2013; Cheruy et al., 2014; Miralles et al., 2014).

30

The results of this study are consistent with previous studies at smaller temporal scales indicating land-atmosphere coupling strength may be stronger in models than in observations. There are several possible mechanisms that may contribute to the

46

**author 9/7/2016 1:58 PM Deleted: the 13 years of author 9/7/2016 1:58 PM Deleted: coupled author 9/7/2016 1:58 PM Deleted: of the land and atmosphere. Feedback author 9/7/2016 1:58 PM Deleted: both author 9/7/2016 1:58 PM Deleted: relationship author 9/7/2016 1:58 PN Deleted: coupling author 9/7/2016 1:58 PM Deleted: feedback author 9/7/2016 1:58 PM Deleted: , which suggests that some of these author 9/7/2016 1:58 PM Deleted: may have difficulty properly predicting author 9/7/2016 1:58 PM Deleted:**

[revised manuscript text omitted]

| author 9/7/2016 1:58 PM                                                                                                                                                                                                                   |                                                                                   |
|-------------------------------------------------------------------------------------------------------------------------------------------------------------------------------------------------------------------------------------------|-----------------------------------------------------------------------------------|
| Moved up [8]: Dirmeyer, P.                                                                                                                                                                                                                |                                                                                   |
| author 9/7/2016 1:58 PM                                                                                                                                                                                                                   |                                                                                   |
| Deleted: A., Doblas-Reyes, F. J.
Gordon, C. T.,                                                                                                                                                                                        | , Drewitt, G.,                                                                    |
| author 9/7/2016 1:58 PM                                                                                                                                                                                                                   |                                                                                   |
| Moved up [10]: Guo, Z.,                                                                                                                                                                                                                   |                                                                                   |
| author 9/7/2016 1:58 PM                                                                                                                                                                                                                   |                                                                                   |
| Moved up [11]: L.,                                                                                                                                                                                                                        |                                                                                   |
| author 9/7/2016 1:58 PM                                                                                                                                                                                                                   |                                                                                   |
| Moved down [13]: E.                                                                                                                                                                                                                       |                                                                                   |
| author 9/7/2016 1:58 PM                                                                                                                                                                                                                   |                                                                                   |
| Moved down [14]: Res. Lett.,                                                                                                                                                                                                              |                                                                                   |
| author 9/7/2016 1:58 PM                                                                                                                                                                                                                   |                                                                                   |
| Moved up [9]: A.,                                                                                                                                                                                                                         |                                                                                   |
| author 9/7/2016 1:58 PM                                                                                                                                                                                                                   |                                                                                   |
| Deleted: Koster, R. D., Mahanan
Yamada, T. J., Balsamo, G., Berg,
M.,                                                                                                                                                        | na, S. P. P.,
A. A., Boisserie,                                                |
| author 9/7/2016 1:58 PM                                                                                                                                                                                                                   |                                                                                   |
| Deleted: Jeong, JH., Lawrence
S., Li, Z., Luo,                                                                                                                                                                                         | , D. M., Lee, W                                                                   |
| author 9/7/2016 1:58 PM                                                                                                                                                                                                                   |                                                                                   |
| Moved up [12]: L.,                                                                                                                                                                                                                        |                                                                                   |
| author 9/7/2016 1:58 PM                                                                                                                                                                                                                   |                                                                                   |
| Deleted: Malyshev, S., Merryfie
Seneviratne, S. I., Stanelle, T., van
M., Vitart, F. and Wood,                                                                                                                               | ld, W. J.,
den Hurk, B. J. J.                                                  |
| author 9/7/2016 1:58 PM                                                                                                                                                                                                                   |                                                                                   |
| Deleted: F.: Contribution of land
initialization to subseasonal foreca
results from a multi-model experim                                                                                                                    | l surface
st skill: First
nent, Geophys.                                    |
| author 9/7/2016 1:58 PM                                                                                                                                                                                                                   |                                                                                   |
| Deleted: 37(2), L02402,
doi:10.1029/2009GL041677, 2010                                                                                                                                                                          | )                                                                                 |
| author 9/7/2016 1:58 PM                                                                                                                                                                                                                   |                                                                                   |
| Deleted: Boisserie, M., Dirmeye
Reyes, F. J., Drewitt, G., Gordon,
Jeong, JH., Lee, WS., Li, Z., Lu                                                                                                                                 | er, P. A., Doblas-
C. T., Guo, Z.,
10,                                      |
| author 9/7/2016 1:58 PM                                                                                                                                                                                                                   |                                                                                   |
| Deleted: Malyshev, S., Merryfie
Seneviratne, S. I., Stanelle, T., van
M., Vitart, F. and Wood, E. F.: Th
the Global Land–Atmosphere Cou
Soil moisture contributions to subs
citil. I Hudramateere 12(5): 800 | den Hurk, B. J. J.
e second phase of
pling Experiment:
seasonal forecast |

doi:10.1175/2011JHM1365.1, 2011.

[revised manuscript text omitted]